# Near-Optimal Decision Trees in a SPLIT Second

**Varun Babbar** [* 1]  **Hayden McTavish** [* 1]  **Cynthia Rudin** [1]  **Margo Seltzer** [2]

## Abstract

Decision tree optimization is fundamental to interpretable machine learning. The most popular approach is to greedily search for the best feature at every decision point, which is fast but provably suboptimal. Recent approaches find the global optimum using branch and bound with dynamic programming, showing substantial improvements in accuracy and sparsity at great cost to scalability. An ideal solution would have the accuracy of an optimal method and the scalability of a greedy method. We introduce a family of algorithms called SPLIT (SParse Lookahead for Interpretable Trees) that moves us significantly forward in achieving this ideal balance. We demonstrate that not all sub-problems need to be solved to optimality to find high quality trees; greediness suffices near the leaves. Since each depth adds an exponential number of possible trees, this change makes our algorithms orders of magnitude faster than existing optimal methods, with negligible loss in performance. We extend this algorithm to allow scalable computation of sets of near-optimal trees (i.e., the Rashomon set).

## 1. Introduction

Decision tree optimization is core to interpretable machine learning (Rudin et al., 2022). Simple decision trees present the entire model reasoning process transparently, directly allowing faithful interpretations of the model (Arrieta et al., 2020). This helps users choose whether to trust the model and to critically examine any perceived flaws.

---

[*]Equal contribution [1]Department of Computer Science, Duke University, Durham, USA [2]Department of Computer Science, University of British Columbia, Vancouver, Canada. Correspondence to: Varun <varun.babbar@duke.edu>, Hayden <hayden.mctavish@duke.edu>.

*Proceedings of the $42^{nd}$ International Conference on Machine Learning*, Vancouver, Canada. PMLR 267, 2025. Copyright 2025 by the author(s).

Code for our algorithms and experiments can be found at https://github.com/VarunBabbar/SPLIT-ICML.

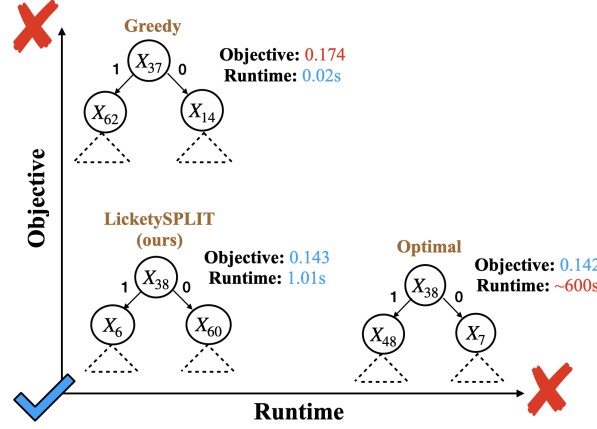

*Figure 1.* An illustration of the power of our optimization algorithm. We train 3 decision trees on the Bike dataset, with the aim of predicting bike rentals in Washington DC in a given time period. A greedy tree is fast but suboptimal. An optimal tree is well performing but *very* slow. Our algorithm strikes the perfect balance, providing well performing trees in a *SPLIT* second, orders of magnitude faster than optimal approaches seen in literature.

Optimizing the performance of decision trees while preserving their simplicity presents a significant challenge. Traditional greedy methods scale linearly with both dataset size and the number of features (Breiman, 1984; Quinlan, 2014).

However, these methods tend to yield suboptimal results, lacking general guarantees on either sparsity or accuracy. Recent advances in decision tree algorithms use dynamic programming techniques combined with branch-and-bound strategies, offering solutions that are faster than brute-force approaches and provably optimal (Lin et al., 2020; Aglin et al., 2020; Demirović et al., 2022; McTavish et al., 2022). In fact, Demirović et al. (2022) and van der Linden et al. (2024) reveal an average gap of 1-2 percentage points between greedy and optimal trees, with Demirović et al. (2022) showing that some datasets can exhibit gaps as large as 10 percentage points. These algorithms struggle to scale to datasets with hundreds or thousands of features or to deeper trees. It seems that we should return to greedy methods for larger-scale problems, but this would come at a loss of performance. Ideally, we should leverage greed only when it does not significantly deviate from optimality and use dynamic programming otherwise. Dynamic programming

approaches build trees recursively, downward from the root. Problems farther from the root contain fewer samples and produce fewer splits. As we show, *greedy splits near the root sacrifice performance*, while *greedy splits near the leaves produce performance close to the optimal*. This suggests that we can tolerate less precision on problems close to leaves than on problems closer to the root – and that full optimization on those problems closer to the leaves yields only marginal returns relative to greedy, since we only have a few splits remaining. This has enormous implications, since the number of candidate trees increases exponentially with increases in depth; using greedy splitting closer to the leaves of the tree massively reduces the search space.

We leverage this observation to construct SPLIT (SParse Lookahead for Interpretable Trees), a family of decision tree algorithms that are over **100× faster than state of the art optimal decision tree algorithms, with negligible sacrifice in performance**. They can also be tuned to a user-defined level of sparsity. Instead of searching through the entire space of decision trees up to a given depth, our algorithm performs dynamic programming with branch and bound up to only a shallow "lookahead" depth, conditioned on all splits henceforth being chosen greedily.

Our contributions are as follows.

- We develop a family of decision tree algorithms that scale with the dataset size and number of features comparably to standard greedy algorithms but produce trees that are as accurate and sparse as optimal ones (e.g., Lin et al., 2020).

- We extend our decision tree algorithms to allow scalable, accurate approximations of the Rashomon set of decision trees (Breiman, 2001; Xin et al., 2022).

- We theoretically prove that our algorithms scale exponentially faster in the number of features than optimal decision tree methods and are capable of performing arbitrarily better than a purely greedy approach.

## 2. Related Work

We are interested in accurate, interpretable decision tree classifiers that we can find efficiently. We discuss these three goals as they pertain to existing work.

Consistent with recommendations from Rudin et al. (2022); Costa & Pedreira (2023), we emphasize sparsity, expressed in terms of the number of leaves, as the primary mechanism for tree interpretability. Sparsity has a strong correlation with user comprehension (Piltaver et al., 2016). Zhou et al. (2018) fit a regression model to user-reported interpretability for decision trees, also finding that trees with fewer leaves were more interpretable. They also found that deep, sparse trees were more interpretable than shallow trees with the same sparsity. Izza et al. (2022) provides a way to use a

sparse decision tree to provide succinct individual explanations. However, finding deep, sparse trees with existing methods can be computationally infeasible. We bridge this gap – our algorithms are capable of finding sparse trees without constraining them to be shallow.

**Greedy Decision Trees** A long line of work explores greedy algorithms such as CART (Breiman, 1984) and C4.5 (Quinlan, 2014). These methods first define a heuristic feature quality metric such as the Gini impurity score (Breiman, 1984) or the information gain (Quinlan, 2014) rather than choosing a global objective function. At every decision node, the feature with the highest quality is chosen as the splitting feature. This process is repeated until a termination criteria is reached. One such criteria often used is the minimum support of each leaf. Trees can then be postprocessed with pruning methods.

**Branch and Bound Optimization** Among the many methods for globally optimizing trees, Branch-and-bound approaches with dynamic programming are state of the art for scalability, because they exploit the structure of decision trees (Costa & Pedreira, 2023; Lin et al., 2020; Demirović et al., 2022; McTavish et al., 2022; Aglin et al., 2020). While many other methods exist for optimizing trees, such as MIP solvers (Bertsimas & Dunn, 2017; Verwer & Zhang, 2019), we focus our discussion and comparison of globally optimal decision tree methods on the currently fastest types of approaches – dynamic programming with branch and bound (DPBnB). These approaches search through the space of decision trees while tracking lower and upper bounds of the overall objective at each split to reduce the search space. They can find optimal trees on medium-sized datasets with tens of features and shallow maximum tree depths (Sullivan et al., 2024; Aglin et al., 2020; Lin et al., 2020; Demirović et al., 2022). Aglin et al. (2020) uses a DPBnB method with advanced caching techniques to find optimal decision trees, though it does not explicitly optimize for sparsity. In contrast, Lin et al. (2020); Hu et al. (2019) use a DPBnB approach to find a tree that optimizes a weighted combination of empirical risk and sparsity, defined by the number of leaves in the tree. McTavish et al. (2022) further enhances this approach by incorporating smart guessing strategies to construct tighter lower bounds for DPBnB, resulting in computational speedups. Demirović et al. (2022) extends the work of Aglin et al. (2020) by focusing on finding the optimal tree with a hard constraint on the number of permissible nodes, using advanced caching techniques and an optimized depth-2 decision tree solver. Mazumder et al. (2022) addresses continuous features by defining lower and upper bounds based on quantiles of feature distributions. However, their method is applicable only to shallow optimal trees with depth $\leq 3$, limiting its utility in scenarios with higher-order feature interactions.

**Lookahead Trees**   Some older approaches to greedy decision tree optimization consider multiple levels of splits before selecting the best split at a given iteration (Norton, 1989). That is, unlike the other greedy approaches, these approaches do not pick the split that optimizes a heuristic immediately. Instead, they pick a split that sets up the best possible heuristic value on the following split.

These approaches still focus on locally optimizing a heuristic measure that is not necessarily aligned with a global objective. By contrast, our method selects splits to directly optimize the sparse misclassification rate of the final tree. We globally optimize the search up to the specified lookahead depth, switching to heuristics only when deciding splits past our lookahead depth. In so doing, our method largely avoids the pathology noted in Murthy & Salzberg (1995), who note cases where their own lookahead approach results in a substantially worse tree than one constructed with a standard greedy approach. For our method, it is provably impossible for a fully greedy entropy-based method with the same constraints as our approach to achieve a better training set objective than our approach. (See Theorem A.1)

**Other Hybrid Methods**   Several other approaches are compatible with branch and bound techniques. Blanc et al. (2024) seek to bridge the gap between greedy and optimal decision trees by selecting a fixed subset of the top $k$ feature splits for each sub-problem. However, this framework does not explicitly account for sparsity. Further, the method is limited by using a *global* setting for search precision: the approach considers the same number of candidate splits at each subproblem. As we show in our experiments, there is merit to tailoring the level of search precision to parts of the search space where it is most needed. The Blossom algorithm (Demirović et al., 2023) traverses a branch and bound dependency graph structure while using greedy heuristics to guide the search order. Relative to our approach, this algorithm optimizes from the bottom up, starting with greedy splits at each level, then optimizing the splits furthest from the root first. This choice guarantees eventual optimality while giving anytime behavior, but misses out on leveraging the property motivating this work – that greedy splits are most detrimental near the top of the tree. Like the approach of Blanc et al. (2024), Blossom also does not account for sparsity.

There are a few methods that use probabilistic search techniques to optimize trees. Sullivan et al. (2024) take a Bayesian approach, finding the maximum-a-posteriori tree by optimizing over an AND/OR graph, akin to the graph used in earlier branch-and-bound methods like that of Lin et al. (2020). Although their method demonstrates strong performance, their experimental results reveal that it is not responsive to sparsity-inducing hyperparameters – accordingly, we found in our experiments that the method struggles

to optimize for sparsity.

Recent work by Chaouki et al. (2024) devises a Monte Carlo Tree Search algorithm using Thompson sampling to enable online, adaptive learning of sparse decision trees. We show that our method achieves superior performance and sparsity on all datasets tested.

## 3. Preliminaries

We consider a typical supervised machine learning setup, with a dataset $D = \{(\mathbf{x}_i, y_i)\}_{i=1}^N$ sampled from a distribution $\mathcal{D}$, where $\mathbf{x}_i \in \{0, 1\}^K$ is a binary feature vector and $y_i \in \{0, 1\}$ is a binary label.[1] Let $\mathcal{F}$ be the set of features. Define $D(f)$ as the subset of $D$ consisting of all samples where feature $f \in \mathcal{F}$ is 1 (and $D(\bar{f})$ as the subset where feature $f$ is 0). Let $D^+$ and $D^-$ denote the set of examples with positive and negative labels, respectively.

**Node specific notation**   Let $D_t$ be the support set of node $t$ in a tree (i.e., the set of training examples assigned to this node); we call each $D_t$ a *subproblem*. Let $f_t \in \mathcal{F}$ be the feature we split on at $t$. Let $D_t(f_t)$ and $D_t(\bar{f}_t)$ be the support sets of the children of $t$. Unless stated otherwise, a greedy split at node $t$ chooses the feature $f$ that maximizes the information gain, which is equivalent to solving:

$$f_t = \min_{f \in \mathcal{F}} \frac{|D_t(f)|}{|D_t|} H\left(\frac{|D_t^+(f)|}{|D_t(f)|}\right) + \frac{|D_t(\bar{f})|}{|D_t|} H\left(\frac{|D_t^+(\bar{f})|}{|D_t(\bar{f})|}\right)$$

with entropy $H(p) = -p \log p - (1 - p) \log(1 - p)$.

**Tree specific notation**   We now briefly discuss sparse greedy and optimal trees. We define $T_g(D, d, \lambda)$ to be a decision tree of depth at most $d$ trained greedily on $D$ with sparsity penalty $\lambda$. Intuitively, this sparse greedy algorithm will make a split at a node only when the gain in overall accuracy is greater than $\lambda$. Algorithm 4 in the Appendix illustrates this procedure. Modern methods such as Lin et al. (2020); McTavish et al. (2022), on the other hand, find a tree $T$ in the space of decision trees $\mathcal{T}$ that solves the following optimization problem:

$$\mathcal{L}^*(D, d, \lambda) = \min_{T \in \mathcal{T}} L(T, D, \lambda) \text{ s.t. depth}(T) \le d \qquad (1)$$

$$= \min_{T \in \mathcal{T}} \sum_{i=1}^{|D|} \frac{1}{N} \Big( l\big(T(\mathbf{x}_i), y_i\big) + \lambda S(T) \Big) \text{s.t. depth}(T) \le d$$

where $L(T, D, \lambda)$ is the regularized loss of tree $T$ on dataset (or data subset) $D$, $S(T)$ is the number of leaves in $T$, $\ell(T(\mathbf{x}), y)$ is the loss incurred by $T$ in its prediction on $\mathbf{x}$

---

[1]The discussions and methods in this paper can trivially be extended to multiclass problems; we focus our discussion and evaluation of the methodology on binary labels.

(for this paper, we set $\ell$ to be the 0-1 loss), and $N$ is the global dataset size. As discussed in Section 2, the fastest contemporary methods solve this problem using a branch-and-bound approach (Costa & Pedreira, 2023; Lin et al., 2020; Demirović et al., 2022; McTavish et al., 2022).

**Rashomon Sets**   Our work is motivated by the properties of near-optimal decision trees and allows for scalable approximation of that set. Xin et al. (2022) define the Rashomon set, denoted by $\mathcal{R}(D, \lambda, \epsilon, d)$, as the collection of all trees whose objective is within $\epsilon$ of the minimum value in Equation 1. Formally:

$$\mathcal{R}(D, \lambda, \epsilon, d) = \{T \in \mathcal{T} : L(T, D, \lambda) \leq \mathcal{L}^*(D, d, \lambda) + \epsilon \\ \wedge \text{depth}(T) \leq d\}. \tag{2}$$

In Section 4, we use Rashomon sets to investigate properties of near-optimal trees.

Rashomon sets can be used for a range of downstream tasks (Rudin et al., 2024); one crucial task is the measurement of variable importance over a set of near-optimal models instead of only for a single model (Donnelly et al., 2023; Fisher et al., 2019). Reliable variable importance measures in this setting rely on minimal feature selection prior to computing the Rashomon set and minimal constraints on the tree's depth to allow high-order interactions. Our approach can be used to accelerate the computation of a Rashomon set, supporting the feasibility of these approaches.

**Branch and Bound**   Given a depth budget $d$, branch and bound with a sparsity penalty (Lin et al., 2020; McTavish et al., 2022) finds the optimal loss $\mathcal{L}^*(D, d, \lambda)$ that minimizes Equation 1.

The key insight behind branch and bound is that the optimal solution for dataset $D$ at depth $d'$ has a dependency on the optimal solution for datasets $D(f)$ and $D(\bar{f})$ at depth $d' - 1$, for each $f \in \mathcal{F}$. Starting from the root, branch and bound algorithms consider different candidate features, $f$, on which to split in the process of determining the objective. As candidates are considered, we identify the subproblems we encounter by the subset of data they relate to and their remaining depth. We track current upper and lower bounds of subproblems in order to prune parts of the search space as we explore it. In particular, if our lower bounds on $\mathcal{L}^*(D_t(f_1), d' - 1, \lambda)$ and $\mathcal{L}^*(D_t(\bar{f}_1), d' - 1, \lambda)$ sum to a larger value than the sum of upper bounds on $\mathcal{L}^*(D_t(f_2), d' - 1, \lambda)$ and $\mathcal{L}^*(D_t(\bar{f}_2), d' - 1, \lambda)$, for example, then we have proven that $f_1$ is not the minimizing split for dataset $D$.

$\mathcal{L}(D_t, d', \lambda)$ can always start with an upper bound of $ub = \lambda + \min\left(\frac{|D_t^-|}{|D_t|}, \frac{|D_t^+|}{|D_t|}\right)$. A universal lower bound is $\lambda$. To get a tighter lower bound, if $d' > 0$, the lower bound can

start at $\min(ub, 2\lambda)$, since either $\mathcal{L}(D_t, d', \lambda) = ub$, or the objective will be the sum of two other $\mathcal{L}$ calls, both of which must necessarily have cost at least $\lambda$. These upper and lower bounds are then updated as we explore a graph structure containing these subproblems. Once these bounds have converged, and we know the value of $\mathcal{L}(D, d', \lambda)$ for the whole dataset $D$, we can extract the optimal tree by simply tracking the feature $f$ that leads to the optimal score for $D$ and then successively track the splits for the optimal value with respect to $D(f)$ and $D(\bar{f})$, and so on.

**Discretization**   Our algorithm will assume feature vectors to be binary, i.e., $\mathbf{x}_i \in \{0, 1\}^K$. Real-world datasets often have features that require discretization to fit our setting. While some methods preserve optimality (e.g., splitting at the mean between unique values in the training set), others such as bucketization (described and proven to be suboptimal in Lin et al., 2020), binning into quantiles, and feature engineering reduce the search space at the cost of optimality. In our experiments, we use threshold guessing (McTavish et al., 2022), which sacrifices optimality with respect to a real-valued dataset but maintains theoretical and empirical guarantees relative to a reference decision tree ensemble.

## 4. Algorithm Motivation

A key motivating property of SPLIT is that we can find high quality trees even when splitting greedily far from the root of the tree. To support this intuition, we empirically investigate how frequently near optimal trees behave greedily far from the root. To do so, we first generate the Rashomon set of decision trees for various values of sparsity penalty $\lambda$ and Rashomon bound $\epsilon$. Let $T \in \mathcal{R}(D, \lambda, \epsilon, d)$ be a tree in the Rashomon set, and let $n \in T$ be any node in $T$. Then, we compute the fraction of all nodes at a given level $\ell \leq d$ (where level 0 corresponds to the root) that were greedy (by which we mean that the split at this node in the tree is optimal with respect to information gain). This corresponds to the following proportion:

$$\frac{\displaystyle\sum_{T \in \mathcal{R}(D, \lambda, \epsilon, d)} \sum_{n \in T} \mathbb{1}[n \text{ is greedy} \wedge \text{level}(n) = \ell]}{\displaystyle\sum_{T \in \mathcal{R}(D, \lambda, \epsilon, d)} \sum_{n \in T} \mathbb{1}[\text{level}(n) = \ell]}. \tag{3}$$

 Figure 2 shows the results of this investigation for 6 different datasets for different values of $\epsilon$ and $\lambda$. We note that there is a general increase in percentage of greedy splits as one goes deeper in the tree.

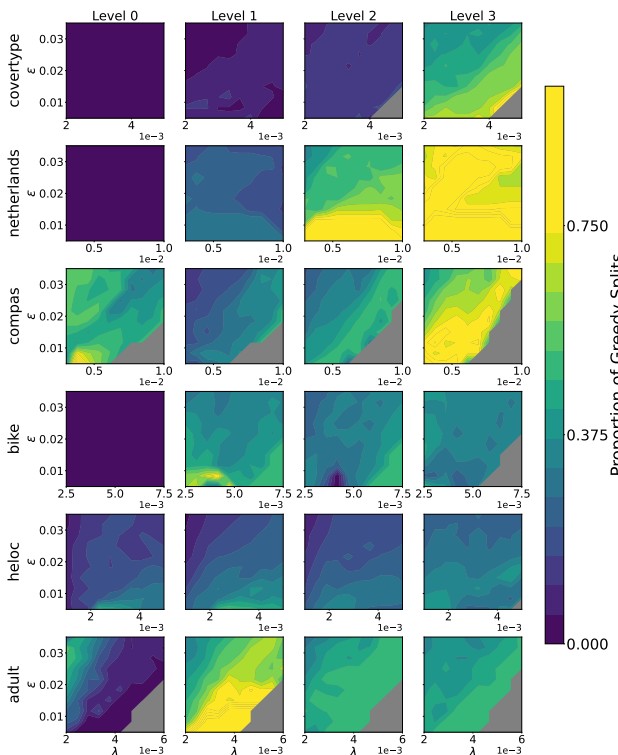

Figure 2. A heatmap of the proportion of splits of trees in the Rashomon set that are greedy, stratified by level, for different $(\lambda, \epsilon)$ combinations. Only 4 levels are shown as the $5^{th}$ level corresponds to the leaf. The greyed out regions in the bottom right of a plot represent $(\lambda, \epsilon)$ for which the Rashomon set did not contain any trees of that depth. Generally, as we approach the leaves, the proportion of splits appearing in $\epsilon$-optimal trees become increasingly greedy. This is especially noticeable for the Netherlands, Covertype, and COMPAS datasets.

Additional motivating empirical results for using greedy splits far from the root of the tree are provided in Appendix A.2.

## 5. Algorithm Details

### 5.1. SParse Lookahead for Interpretable Trees (SPLIT)

We now formalize our main algorithm, SPLIT, which takes as input a *lookahead depth* parameter. This is the depth up to which a search algorithm optimizes over all combinations of feature splits, conditioned on splits beyond this depth behaving greedily. Our algorithm exploits the fact that sub-problems closer to the leaves exhibit smaller optimality gaps than those at the root, providing a mechanism to trade off among runtime, accuracy, and sparsity.

**Formulating the optimization problem** Concretely, for a given depth budget $d$, lookahead depth $d_l < d$, and feature

set $\mathcal{F}$, we **first** solve the following recursive equation:

$$\mathcal{L}(D, d', \lambda) =$$

$$
\begin{cases}
\min \left\{ \lambda + \frac{|D^-|}{N}, \lambda + \frac{|D^+|}{N}, \right. \\
\quad \left. \min_{f \in \mathcal{F}} \left\{ L\Big(T_g\big(D(f), d', \lambda\big)\Big) + L\Big(T_g\big(D(\bar{f}), d', \lambda\big)\Big) \right\} \right\} \\
\hspace{8em} \text{if } d' = d - d_l \\[1em]
\min \left\{ \lambda + \frac{|D^-|}{N}, \lambda + \frac{|D^+|}{N}, \right. \\
\quad \left. \min_{f \in \mathcal{F}} \left\{ \mathcal{L}\Big(D(f), d' - 1, \lambda\Big) + \mathcal{L}\Big(D(\bar{f}), d' - 1, \lambda\Big) \right\} \right\} \\
\hspace{8em} \text{if } d' > d - d_l.
\end{cases}
$$

$$(4)$$

Where N is the size of the dataset at the root. We can con-

---

**Algorithm 1** get_bounds($D, d_l, d, d', N$) $\rightarrow$ lb, ub

---

**Require:** $D, d_l, d, d', N$ {support, lookahead depth, current search depth, maximum search depth, size of full dataset in GOSDT call}

1: **if** $d' = d_l$ **then**
2: $\quad T_g = \text{Greedy}(D, d - d_l, \lambda)$ {Find greedy tree rooted at $D$ (Alg 4 in the Appendix)}
3: $\quad S(T_g) = \#$ Leaves in $T_g$
4: $\quad \alpha \leftarrow \frac{1}{N} \sum_{(x,y) \in D} \mathbf{1}[y \neq T_g(x)] + \lambda S(T_g)$
5: $\quad lb \leftarrow \alpha$
6: $\quad ub \leftarrow \alpha$ {subproblem solved because ub = lb}
7: **else** {use basic initial bounds}
8: $\quad lb \leftarrow 2\lambda$
9: $\quad ub \leftarrow \lambda + \min \left\{ \frac{|D^-|}{N}, \frac{|D^+|}{N} \right\}$
10: **end if**
11: **return** lb,ub {Return Lower and Upper Bounds}

---

strain the search space to include only greedy trees past the lookahead depth by modifying the lower and upper bounds used in branch and bound (see Algorithm 1). In particular, sub-problem nodes initialized at depths up to the lookahead depth are assigned initial lower and upper bounds equivalent to that in GOSDT (Lin et al., 2020) (see Section 2). At the lookahead depth, however, the lower and upper bounds for a subproblem are fixed to be the loss of a greedy subtree trained on that subproblem. After these bound assignments, our algorithm uses the GOSDT algorithm with these new bounds to solve Equation 4 – this is summarized by Lines 1-2 in Algorithm 2. We defer more details of the GOSDT algorithm to Section A.12 in the Appendix.

**Postprocessing with Optimal Subtrees** Once we have solved Equation 4, we do not need to use greedy sub-trees past the lookahead depth. We can improve our approach by

---

**Algorithm 2** SPLIT($\ell$, D, $\lambda$, $d_l$, $d$, $p$)

---

**Require:** $\ell, D, \lambda, d_l, d, p$ {loss function, samples, regularizer, lookahead depth, depth budget, postprocess flag}

1: ModifiedGOSDT = GOSDT reconfigured to use **get_bounds** (Algorithm 1) whenever it encounters a new subproblem
1: $t_{lookahead}$ = ModifiedGOSDT($\ell, D, \lambda, d_l$) {Call ModifiedGOSDT with depth budget $d_l$}
2: **if** $p$ **then** {Fill in the leaves of this prefix}
3:    **for** leaf $u \in t_{lookahead}$ **do**
4:      $d_u$ = depth of leaf
5:      $D(u)$ = subproblem associated with $u$
6:      $\lambda_u = \lambda \frac{|D|}{|D(u)|}$ {Renormalize $\lambda$ for the subproblem in question}
7:      $t_u$ = GOSDT($D(u), d-d_u, \lambda_u$) {Find the optimal subtree for $D(u)$}
8:      **if** $t_u$ is not a leaf **then**
9:        Replace leaf $u$ with sub-tree $t_u$
10:      **end if**
11:    **end for**
12: **end if**
13: **return** $t_{lookahead}$

---

**Algorithm 3** LicketySPLIT($\ell$, $D$, $\lambda$, $d$)

---

**Require:** $\ell, D, \lambda, d$ {loss function, samples, regularizer, full depth}

0: $t_{lookahead}$ = SPLIT($\ell, D, \lambda, 1, d, 0$) {Call SPLIT with lookahead depth 1 and no post-processing}
1: **if** $t_{lookahead}$ is not a leaf **then**
2:    **for** child $u \in t_{lookahead}$ **do**
3:      $D(u)$ = subproblem associated with $u$
4:      $\lambda_u = \lambda \frac{|D|}{|D(u)|}$ {Renormalize $\lambda$ for the subproblem in question}
5:      $t_u$ = LicketySPLIT($\ell, D(u), \lambda_u, d-1$)
6:      Replace $u$ with subtree $t_u$
7:    **end for**
8: **end if**
9: **return** $t_{lookahead}$

---

replacing these subtrees with fully optimal decision trees. Lines 3-9 in Algorithm 2 illustrate this. Thus, the performance of the lookahead tree with the aforementioned greedy subtrees is just an upper bound on the objective of the tree our method ultimately finds.

Note that the renormalization in line 6 of Algorithm 2 ensures that the $\lambda$ penalty stays proportional to the penalty for each misclassified point. Our objective (Equation 1) assigns a $\frac{1}{N}$ penalty for each misclassification, where $N$ is the size of the full dataset with which GOSDT was called. If the original dataset is $D$, then when we call GOSDT on any descendent subproblem $D(u)$, our penalty per misclassification goes up by a factor of $\frac{|D|}{|D(u)|}$. We need to scale $\lambda$ appropriately to stay proportional to the original dataset $D$.

### 5.2. LicketySPLIT: Polynomial-time SPLIT

We present a polynomial-time variant of SPLIT, called LicketySPLIT, in Algorithm 3. This method works by recursively applying SPLIT with lookahead depth 1. That is, we first find the optimal initial split for the dataset, given that we are fully greedy henceforth. Then, during postprocessing, instead of doing what SPLIT would do —running a fully optimal decision tree algorithm on the root's left and right subproblems —we run LicketySPLIT recursively on these two subproblems. We stop considering further calls to LicketySPLIT for a subproblem if SPLIT returns a leaf instead of making splits (either due to the depth limit or $\lambda$).

### 5.3. RESPLIT: Rashomon set Estimation with SPLIT

At the cutting edge of compute requirements for decision tree optimization is the computation of Rashomon sets of decision trees. Xin et al. (2022) compute a Rashomon set of all near-optimal trees, based on the GOSDT algorithm (Lin et al., 2020). This task generates an extraordinary number of trees and has high memory and runtime costs. To make this tractable, Xin et al. (2022) leverage depth constraints and feature selection from prior work to reduce the depth and set of features considered (McTavish et al., 2022). While necessary for scalability, this can prevent exploration of near-optimal models across all features or at greater decision tree depths. Both factors are relevant for work on variable importance based on Rashomon sets (Fisher et al., 2019; Dong & Rudin, 2020; Donnelly et al., 2023). We leverage SPLIT as a way to dramatically improve scalability of Rashomon set computation, reliably approximating the full Rashomon set and allowing feasible exploration while relaxing or removing depth and feature constraints.

Our algorithm, RESPLIT, is described in Appendix A.10; it first leverages SPLIT as a subroutine to obtain a set of prefix trees such that completing them greedily up to the depth budget would result in an $\epsilon$ approximation of the optimal solution to Equation 4. At each leaf of each prefix tree, it calls TreeFARMS (Xin et al., 2022) to find a large set of shallow subtrees that are at least as good as being greedy, yielding an approximate Rashomon set computed much faster than state of the art. We also show a novel indexing mechanism to query RESPLIT trees in Appendix A.11.

## 6. Theoretical Analysis of Runtime and Optimality

We present theoretical results establishing the performance and scalability of our algorithms. All proofs, including

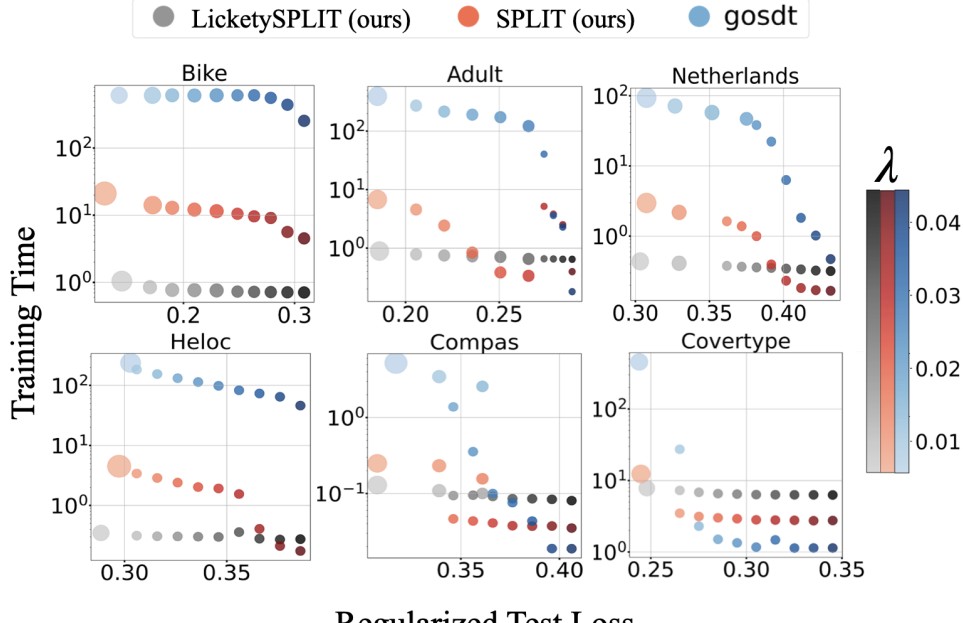

*Figure 3.* Regularized test loss vs training time (in seconds) for GOSDT (McTavish et al., 2022) vs our algorithms. The size of the points indicates the number of leaves in the resulting tree. Both SPLIT and LicketySPLIT are much faster for most values of sparsity penalty $\lambda$, with the only potential slowdown being in the sub-second regime due to overhead costs.

additional lemmas not described below, are in Appendix Section A.8. Even without the speedups discussed in Section 5.2, Algorithm 2 is quite scalable. Theorem 6.1 shows the asymptotic analysis of the algorithm, with and without caching. Note that the default behaviour of Algorithm 2 is to cache repeated sub-problems.

**Theorem 6.1** (Runtime Complexity of SPLIT). *For a dataset $D$ with $k$ features and $n$ samples, depth constraint $d$ such that $d \ll k$, and lookahead depth $0 \leq d_l < d$, Algorithm 2 has runtime $\mathcal{O}\big(n(d - d_l)k^{d_l+1} + nk^{d-d_l}\big)$. If we cache repeated subproblems, the runtime reduces to $\mathcal{O}\Big(\frac{n(d-d_l)k^{d_l+1}}{d_l!} + \frac{nk^{d-d_l}}{(d-d_l)!}\Big)$.*

This algorithm is linear in sample size and, because $d_l < d$ and $d - d_l < d$, is exponentially faster than a globally optimal approach, which searches through $\mathcal{O}((2k)^d)$ sub-problems in the worst case.

Corollaries 6.2 and 6.3 show that, compared to globally optimal approaches, we see substantial improvements in runtime when lookahead depth is around half the global search depth.

**Corollary 6.2** (Optimal Lookahead Depth for Minimal Runtime). *The optimal lookahead depth that minimizes the asymptotic runtime of Algorithm 2 is $d_l = \frac{(d-1)}{2}$ for large $k$, regardless of whether subproblems are cached.*

**Corollary 6.3** (Runtime Savings of SPLIT Relative to Globally Optimal Approaches). *Asymptotically, under the same*

*conditions as Theorem 6.1 and with caching repeated subproblems, Algorithm 2 saves a factor of $\mathcal{O}\Big(k^{\frac{d-1}{2}}\big(\frac{d}{2}\big)!\Big)$ in runtime relative to globally optimal approaches (e.g., GOSDT).*

Theorem 6.4 describes the runtime complexity of our LicketySPLIT method from Section 5.2, showing that it can be even faster than Algorithm 2 (indeed, achieving low-order polynomial runtime).

**Theorem 6.4** (Runtime Complexity of LicketySPLIT). *For a dataset $D$ with $k$ features and $n$ samples, and for depth constraint $d$, Algorithm 3 has runtime $O(nk^2d^2)$.*

We can thus use Algorithm 3 to leverage a recursive search while remaining comfortably polynomial. This is a dramatic improvement to asymptotic scalability relative to globally optimal decision tree construction methods, which solve an NP-hard problem.

**Theorem 6.5** (SPLIT Can be Arbitrarily Better than Greedy). *For every $\epsilon > 0$ and depth budget $d$, there exists a data distribution $\mathcal{D}$ and sample size $n$ for which, with high probability over a random sample $S \sim \mathcal{D}^n$, Algorithm 2 with $d_l = \frac{d-1}{2}$ achieves accuracy at least $1 - \epsilon$ but a pure greedy approach achieves accuracy at most $\frac{1}{2} + \epsilon$.*

Theorem 6.5 shows that Algorithm 2 can arbitrarily outperform greedy methods in accuracy, even when we choose its minimum runtime configuration of $d_l = \frac{d-1}{2}$. We prove

a similar claims for LicketySPLIT and RESPLIT in the appendix (see Theorems A.7 and A.6).

# 7. Experiments

Our experiments provide an evaluation of decision trees, considering aspects of performance, interpretability, and training budget. To this end, our evaluation addresses the following questions:

1. How fast are SPLIT and LicketySPLIT compared to unmodified GOSDT?

2. Are SPLIT and LicketySPLIT able to produce trees that lie on the frontier of sparsity, test loss performance, and training time?

3. How good is the Rashomon set approximation produced by RESPLIT?

For all experiments below we set the depth budget of our algorithms to 5. The lookahead depth for Algorithm 2 is set to 2 since, from Corollary 6.2, this produces the lowest runtime for the chosen depth budget. We defer more details of our experimental setup and datasets to Appendix A.7.4. Appendix A.3 has additional evaluations of our methods.

## 7.1. How do our algorithms compare to GOSDT?

Our first experiments support the claim that our method is significantly faster than GOSDT whilst achieving similar regularized test losses. This is shown in Figure 3. Here, we vary the sparsity penalty, $\lambda$, which is a common input to all algorithms in this figure, and compute the regularized test objective from Equation 1 for each value of $\lambda$. We set a timeout limit of 1200 seconds for GOSDT, after which it gives the best solution found so far. We note two regimes:

- When all methods have lower regularized objective values (left side of each plot), **our methods are orders of magnitude faster than GOSDT.** For instance, on the Bike dataset, SPLIT has training times of $\sim$10 seconds, while GOSDT runs for $\sim 10^3$ seconds. LicketySPLIT takes merely a second in most cases. This is the regime most relevant to our algorithms.

- When the optimal objective is high and the tree is super-sparse (right side of each plot), SPLIT and Lickety-SPLIT have small overhead costs and can be slower, because we need to train a greedy tree for each sub-problem encountered at the lookahead depth in order to initialize bounds via Algorithm 1. However, in this regime, all methods already have runtimes of $\sim$1 second, so the extra overhead cost is insignificant. This is especially seen in the COMPAS and Netherlands datasets.

## 7.2. Characterising the Frontier of Test Loss, Sparsity, and Runtime

Figure 4 characterises the frontier of training time, sparsity, and test loss for several algorithms. Here, we vary hyperparameters associated with each algorithm to produce trees of varying sparsity levels (where sparsity is the number of leaves). We see that there exists a frontier between test loss and sparsity, and different methods lie on different parts of the frontier. To maximize interpretability and accuracy, we want a tree to lie in the bottom left corner of the frontier, within the highlighted red rectangle. Out of all algorithms tested, ours consistently lie on the frontier and in the red rectangle. Alongside state of the art performance, our algorithms are often **over 100× faster** than their contemporaries. For more datasets, see Figure 5 in the Appendix.

## 7.3. Rashomon Set Approximation

We now show that RESPLIT enables fast, accurate approximation of the Rashomon set of near-optimal trees, while scaling much more favorably than state-of-the-art method TreeFARMS (Xin et al., 2022). We demonstrate that variable importance conclusions using RID (Donnelly et al., 2023) remain almost identical under RESPLIT, relative to the full Rashomon set. That is, RESPLIT allows accurate summary statistics of the full Rashomon set to be computed at greater depths and over more binary features while enhancing scalability. Table 1 shows computation of RID with and without RESPLIT. RESPLIT enables $10 - 20\times$ faster variable importance computation. Furthermore, the correlation between variable importances is very close to 1, suggesting that RESPLIT trees serve as good proxies for estimating importances derived from the complete Rashomon set. Table 2 also shows that most of the trees output by RESPLIT lie in the true Rashomon set or very close to it.

| Dataset | Full (s) | RESPLIT (s) | $\tau$ |
|---|---|---|---|
| COMPAS | 152 | **18** | 1.0 |
| Spambase | 2659 | **154** | 0.930 |
| Netherlands | 4255 | **216** | 0.932 |
| HELOC | 5564 | **337** | 0.979 |
| HIV | 9273 | **388** | 0.959 |
| Bike | 14330 | **194** | 0.999 |

*Table 1.* Table summarizing the advantages of RESPLIT. The first 2 columns show the time taken to compute all bootstrapped Rashomon sets for the Rashomon Importance Distribution (RID) (Donnelly et al., 2023) with and without RESPLIT. # of bootstrapped datasets = 10, $\lambda = 0.02$, $\epsilon = 0.01$, depth budget 5, lookahead depth 3. The last column shows the Pearson correlation between variable importances computed by RID and RID + RE-SPLIT. There is nearly perfect correlation seen in every case.

**Test Loss vs # Leaves for Different Decision Tree Algorithms**

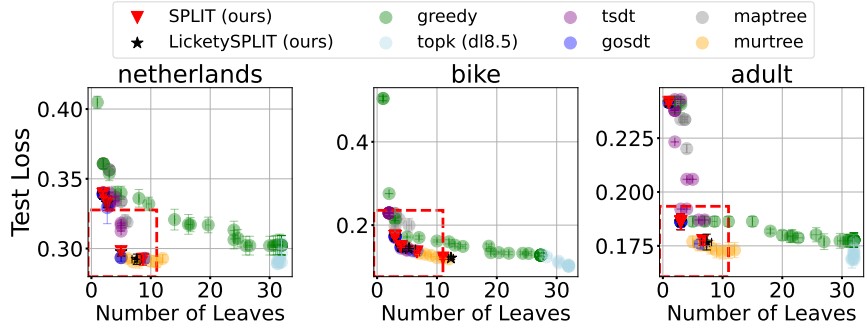

**Test Loss vs Training Time for Models in the Red Box**

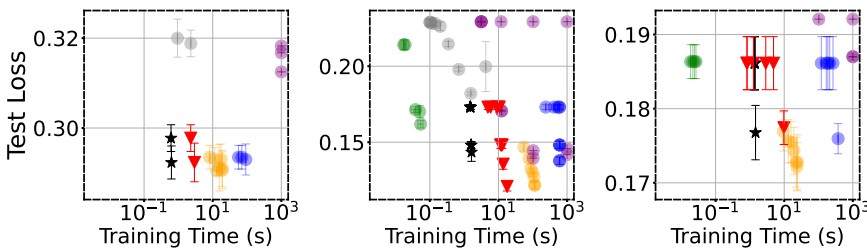

*Figure 4.* A comparison between the performance of our algorithms and competitors (depth budget 5, lookahead depth 2). The red box in the upper plot illustrates the region containing sparse and accurate models. The lower plots show the test loss vs training time for models in the red box. SPLIT and LicketySPLIT consistently lie on the bottom left of the test loss-sparsity frontier, with runtimes orders of magnitude faster than many competitors. Our algorithms also offer the ideal compromise between runtime and loss. All metrics are averaged over 3 test-train splits.

| Dataset | Precision | Precision (Slack .01) |
|---|---|---|
| Bike | 0.974 (370/380) | 1.000 |
| COMPAS | 1.000 (27/27) | 1.000 |
| HELOC | 0.974 (528/542) | 1.000 |
| HIV | 0.528 (243/460) | 0.984 |
| Netherlands | 0.911 (102/112) | 1.000 |
| Spambase | 0.597 (850/1422) | 0.933 |

*Table 2.* Proportion of RESPLIT Trees in the true Rashomon set (precision) and within at most .01 loss of being in the set. Most of the trees output by RESPLIT end up being in the Rashomon set. Trees which are not in the Rashomon set are almost always very close to being in it. We employ the same parameters as Table 1.

## 8. Conclusion

We introduced SPLIT, LicketySPLIT, and RESPLIT, a novel family of decision tree optimization algorithms. At their core, these algorithms perform branch and bound search up to a lookahead depth, beyond which they switch to greedy splitting. Our experimental results show dramatic improvements in runtime compared to state of the art algorithms, with negligible loss in accuracy or sparsity. RESPLIT also scalably finds a set of near-optimal trees without adversely impacting downstream variable importance tasks. Future work could explore conditions under which subproblems exhibit large optimality gaps, offering new insights for effi-

cient decision tree and Rashomon set optimization.

## Acknowledgements

We acknowledge funding from the National Institutes of Health under 5R01-DA054994, the National Science Foundation under award NSF 2147061, and through the Department of Energy under grant DE-SC0023194. We thank Srikar Katta, Jon Donnelly, Zachery Boner, Yixiao Wang, and Zakk Heile for helpful discussions and feedback throughout this project.

## Impact Statement

This paper presents work whose goal is to advance the field of Machine Learning. There are many potential societal consequences of our work, none which we feel must be specifically highlighted here.

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

# A. Appendix

## A.1. Further Comparisons With Other Methods

### A.1.1. MORE DATASETS WITH DEPTH 5 TREES

In Section 7 of the paper, we showed results for three datasets. Here, we evaluate SPLIT, LicketySPLIT, and its contemporaries on 6 additional datasets. All datasets were evaluated on three random 80-20 train-test splits of the data, with the average and standard error reported. Results are in Figure 5. Note that Covertype has smaller error bars because the dataset size is much larger – it has $\sim 5 \times 10^6$ examples, while COMPAS and HELOC have only $\sim 10^4$ examples.

**Test Loss vs # Leaves for Different Decision Tree Algorithms**

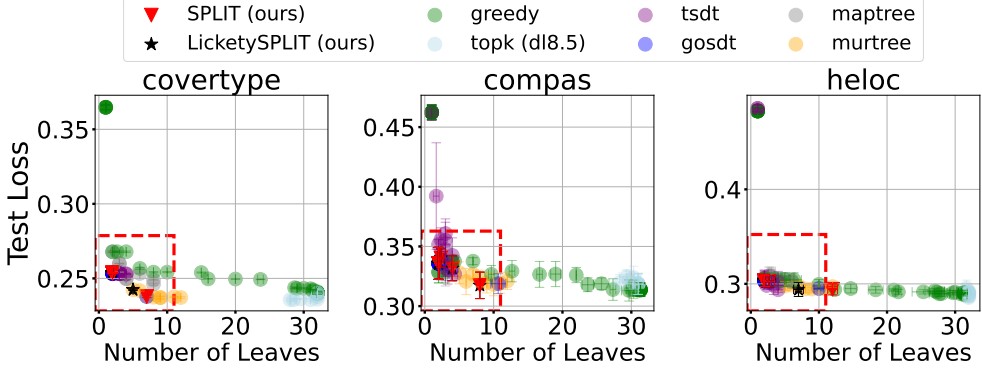

**Test Loss vs Training Time for Models in the Red Box**

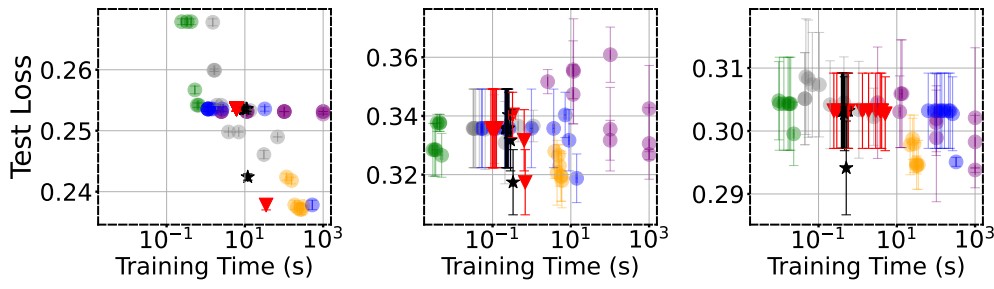

**Test Loss vs # Leaves for Different Decision Tree Algorithms**

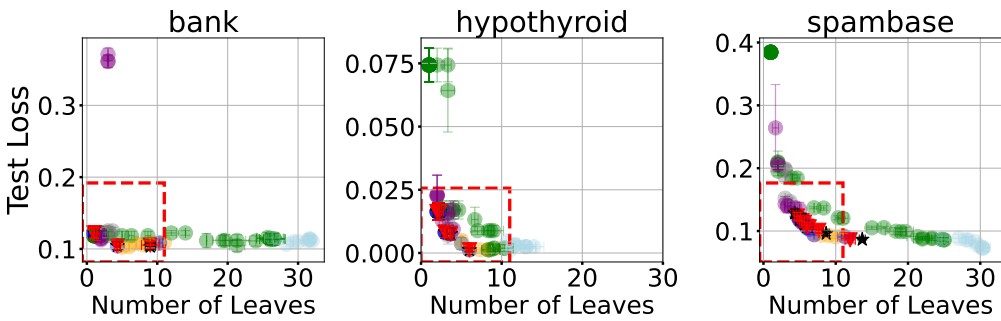

**Test Loss vs Training Time for Models in the Red Box**

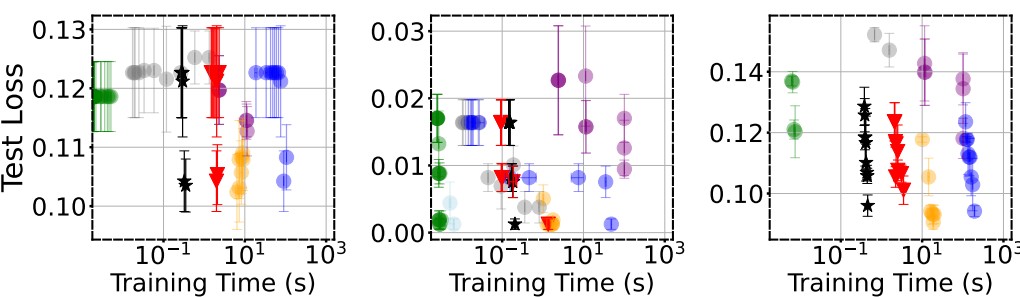

*Figure 5.* A performance comparison between our algorithm and those in literature. The lower row are zoomed in versions of the red boxes in the upper row. This is complementary to Figure 4 and shows more datasets for completeness. The depth budget for all algorithms whose depth budget can be specified is 5.

A.1.2. WHAT ABOUT DEPTH 4 TREES?

In this section, we perform the same evaluation as above, but with depth 4 trees. We set the lookahead depth as 2.

**Test Loss vs # Leaves for Different Decision Tree Algorithms**

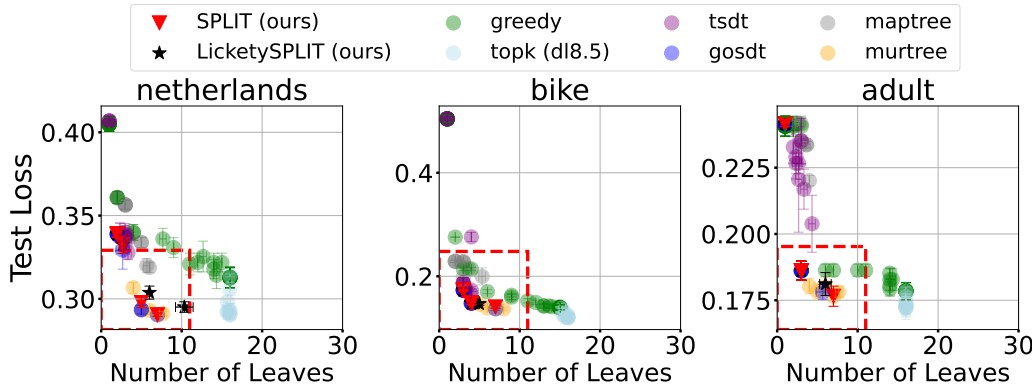

**Test Loss vs Training Time for Models in the Red Box**

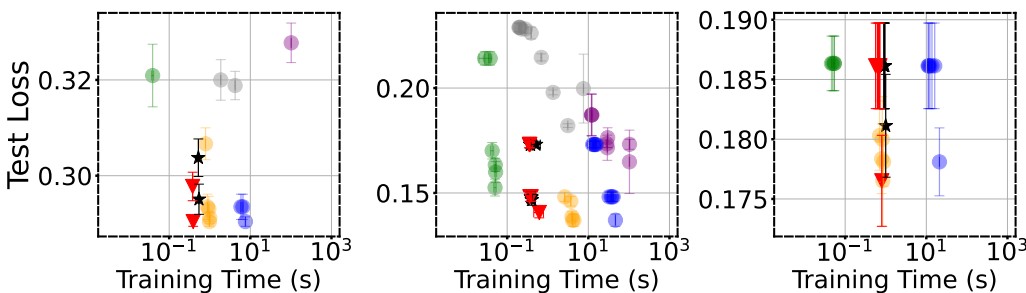

**Test Loss vs # Leaves for Different Decision Tree Algorithms**

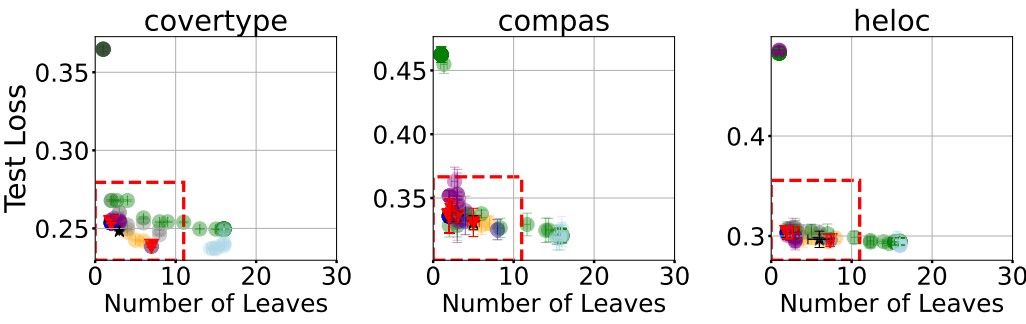

**Test Loss vs Training Time for Models in the Red Box**

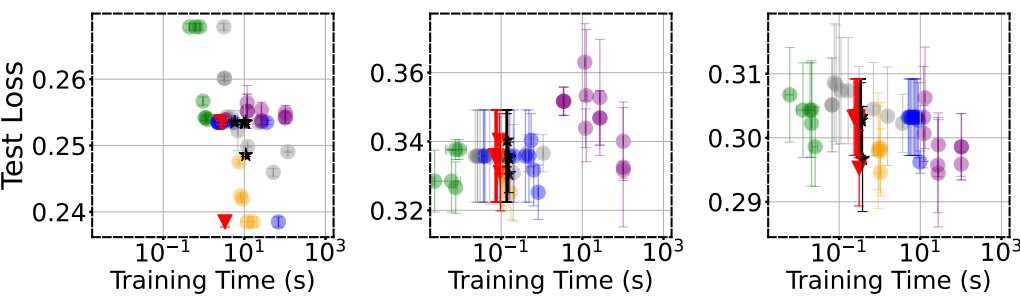

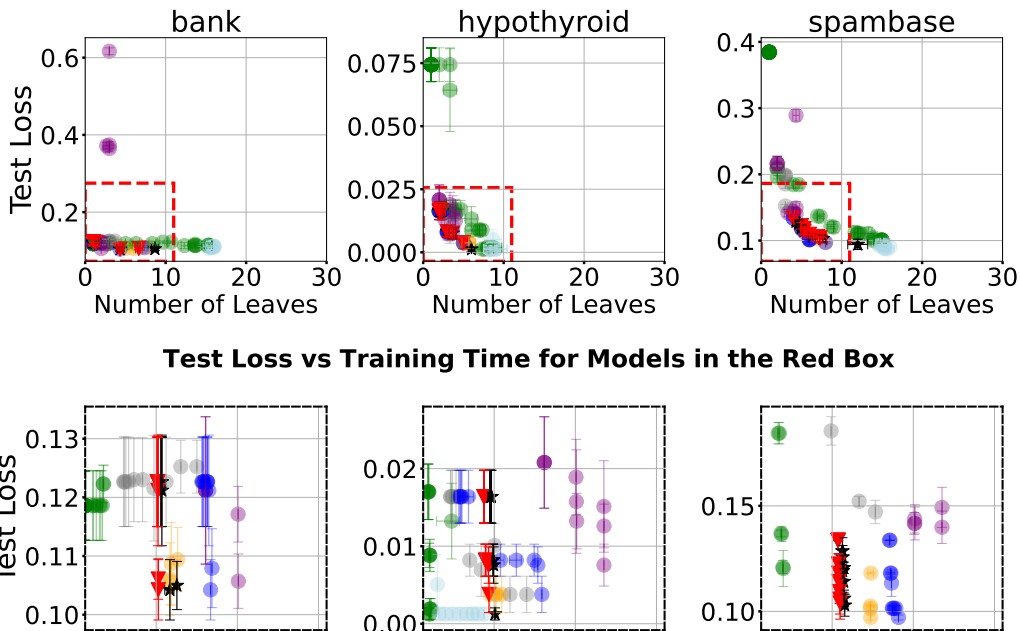

Figure 6. A performance comparison between our algorithm and those in literature. Depth 4 – Lookahead depth 2.

A.1.3. WHAT ABOUT DEPTH 6 TREES?

In this section, we perform the same evaluation as above, but with depth 6 trees. We set the lookahead depth as 2. Note that Murtree and GOSDT are not included in the comparison as they take much longer to run for deeper trees.

**Test Loss vs # Leaves for Different Decision Tree Algorithms**

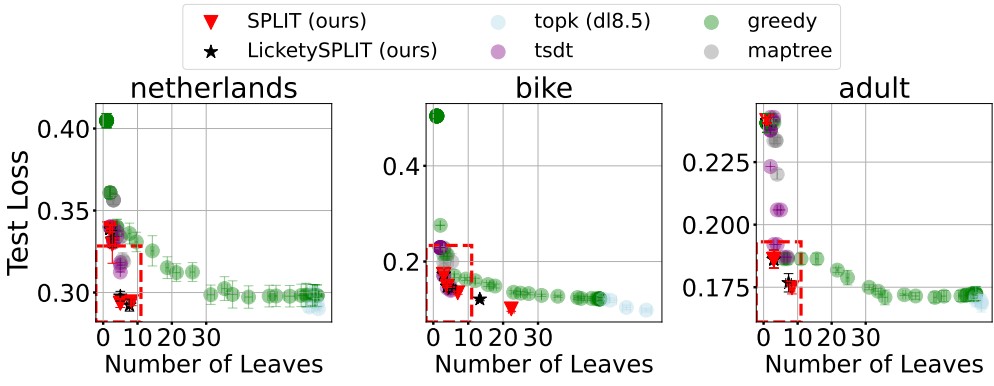

**Test Loss vs Training Time for Models in the Red Box**

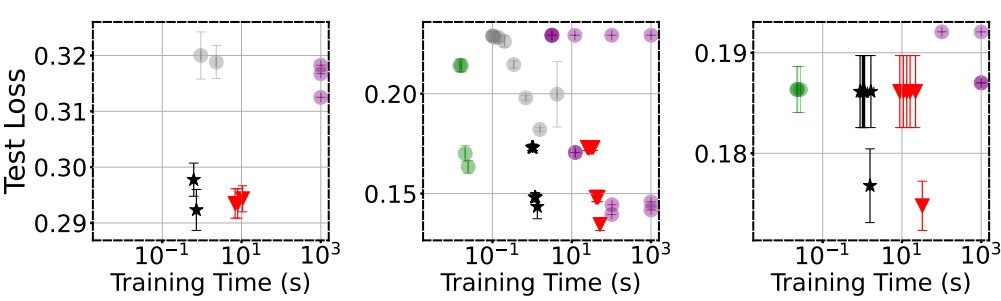

**Test Loss vs # Leaves for Different Decision Tree Algorithms**

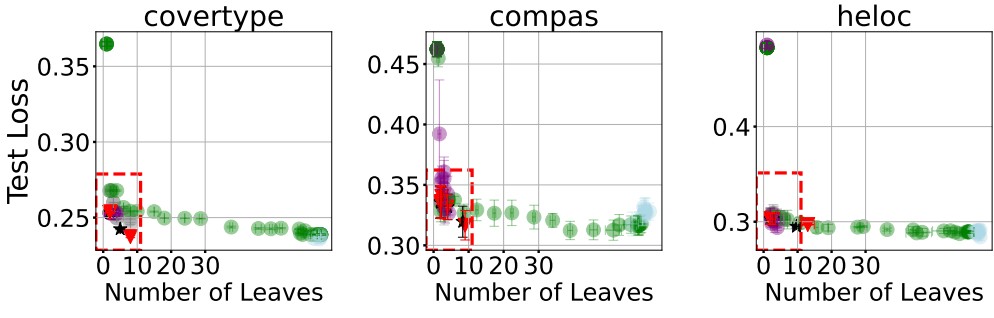

**Test Loss vs Training Time for Models in the Red Box**

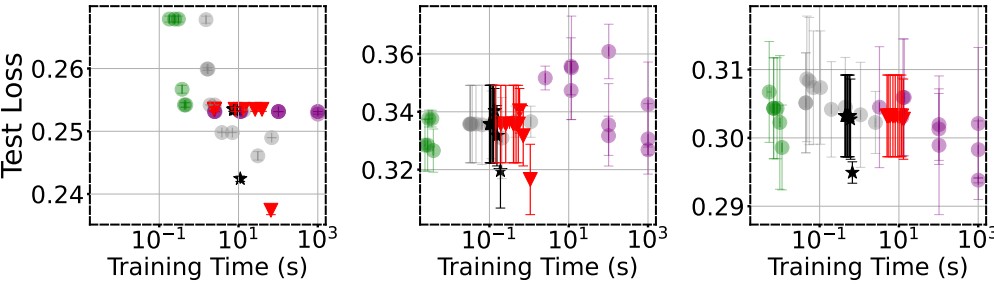

**Test Loss vs # Leaves for Different Decision Tree Algorithms**

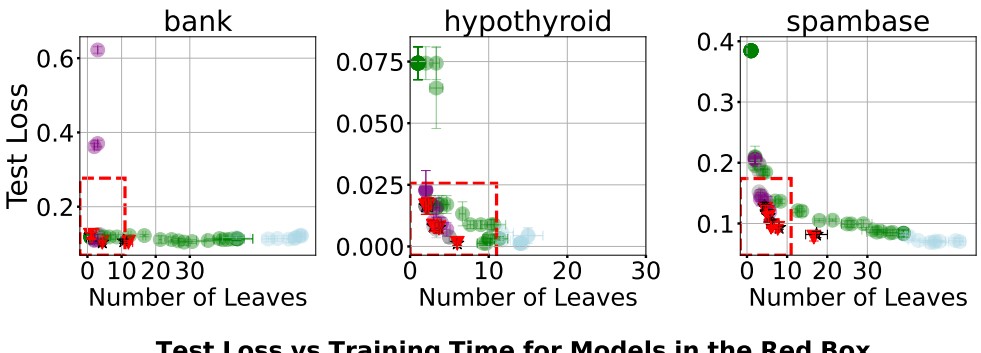

**Test Loss vs Training Time for Models in the Red Box**

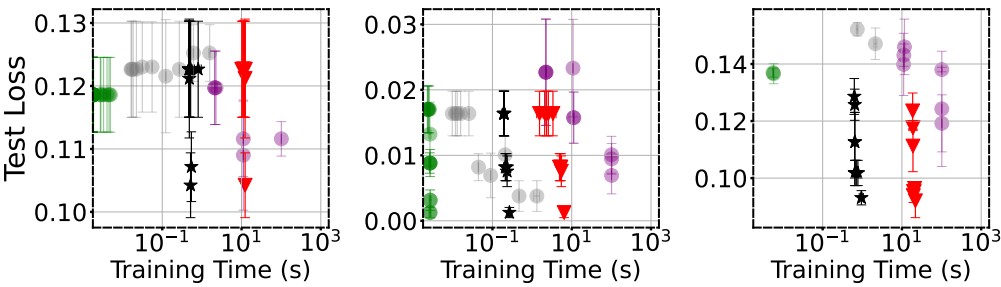

*Figure 7.* A performance comparison between our algorithm and those in literature. Depth 6 – Lookahead depth 2.

## A.2. Many Near-Optimal Trees Exhibit Monotonically Decreasing Optimality Gaps Closer to Leaves

Consider an $\epsilon$-optimal tree $T \in \mathcal{R}(D, \lambda, \epsilon, d)$. For a subtree $t$ of $T$, define $\lambda_t$ as the value of $\lambda$ that results in the greedy tree, $T_g$, having the same number of leaves as $t$. We now define the *optimality gap* $\delta(D_t, t)$ as the difference between the loss of $t$ and the loss of an equally sparse greedy tree on the sub-problem associated with $t$. This enables a fair performance comparison between greedy and optimal trees, as the training loss of any given tree will otherwise monotonically decrease with the number of leaves.

$$\delta(D_t, t) = L(t, D_t, \lambda) - L(T_g(D_t, \text{depth}(t), \lambda_t), D_t, \lambda_t). \tag{5}$$

For a tree $T \in \mathcal{R}$, we then compute the average optimality gap associated with subtrees at each level. That is, given a level $\ell$, we compute:

$$\beta(T, D, \ell) = \frac{\sum\limits_{t \in T} \delta(D_t, t) \mathbb{1}[t \text{ is rooted at level } \ell]}{\sum\limits_{t \in T} \mathbb{1}[t \text{ is rooted at level } \ell]}. \tag{6}$$

We want to determine if $\beta(T, D, l)$ is monotonically decreasing with $\ell$ for a given tree $T$ – if this is true, then being greedy closer to the leaf does not incur much loss in performance. Our intuition is as follows: if there are many such near optimal trees, then a semi-greedy search strategy could potentially uncover at least one of them. The following statistic computes the proportion of all trees in the Rashomon set that have monotonically decreasing optimality gaps as $\ell$ increases (i.e., moves from root towards leaves):

$$m(D, \lambda, \epsilon, d) = \frac{\sum_{T \in \mathcal{R}(D, \lambda, \epsilon, d)} \mathbb{1}\left[\beta(T, D, \ell) \text{ is monotonically decreasing with } \ell\right]}{|\mathcal{R}(D, \lambda, \epsilon, d)|}. \tag{7}$$

Figure 8 shows this statistic for Rashomon sets with varying values of the sparsity penalty $\lambda$. We fix $\epsilon = 0.025$. The sparser a near-optimal tree, the more likely that it will be greedy, however, for all datasets, there exist near-optimal trees with monotonically decreasing optimality gaps even for low sparsity penalties. This has important algorithmic implications for developing interpretable models, because it means that a search strategy that is increasingly greedy near the leaves can produce a near-optimal tree.

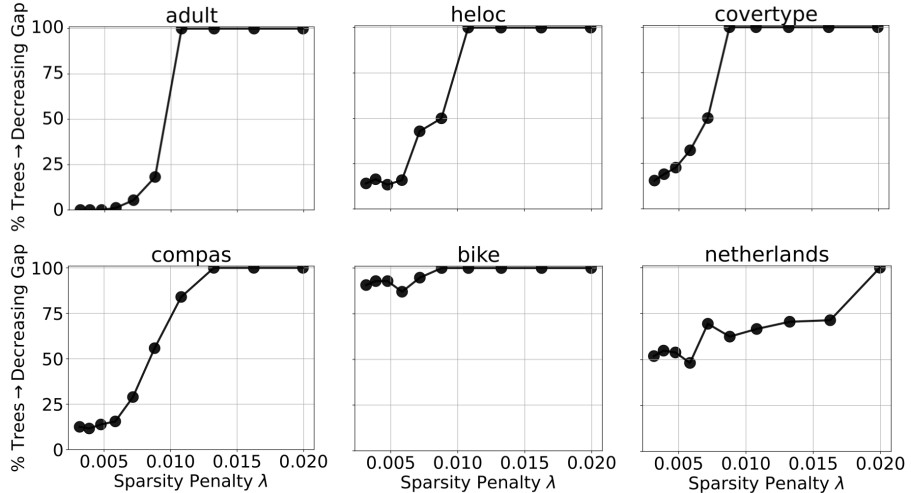

*Figure 8.* Percentage of trees in the Rashomon set that exhibit monotonically decreasing optimality gaps. For sparse trees (i.e., where $\lambda$ is larger), we are more likely to find a tree whose optimality gap is consistently decreasing at each level. This suggests that behaving greedily only near the leaves can produce a well-performing tree.

## A.3. Miscellaneous Properties of SPLIT

### A.3.1. WHICH LOOKAHEAD DEPTH SHOULD I USE?

In this section, we explore the effect of the lookahead depth on the runtime and regularised test and train losses. We use the aggressively binarized versions of the datasets, as elaborated in Section A.7.

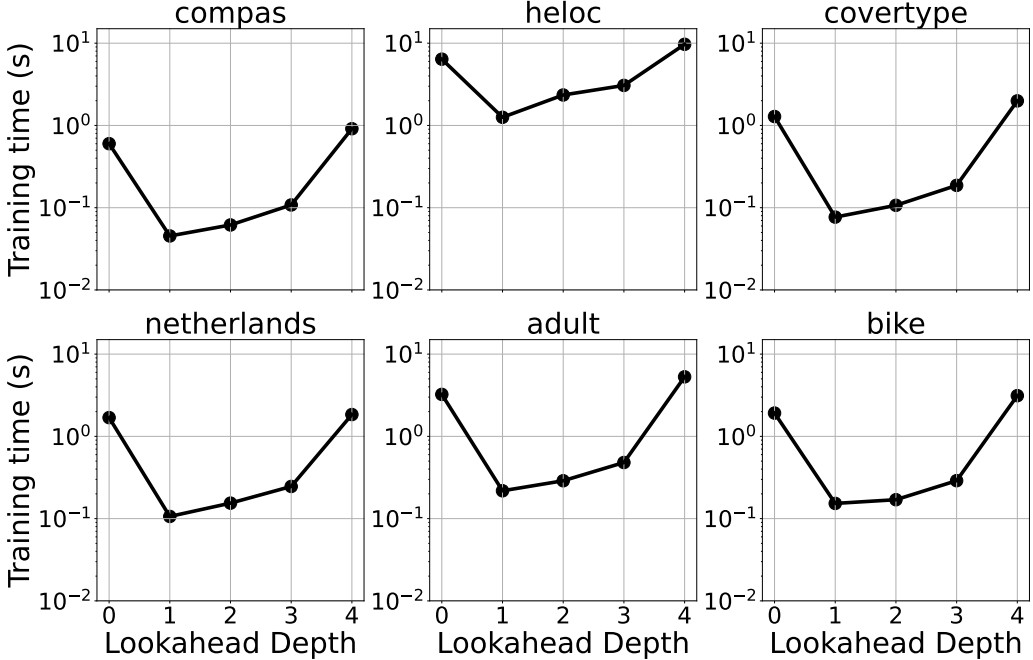

*Figure 9.* Runtime as a function of the lookahead depth. $\lambda = 0.001$

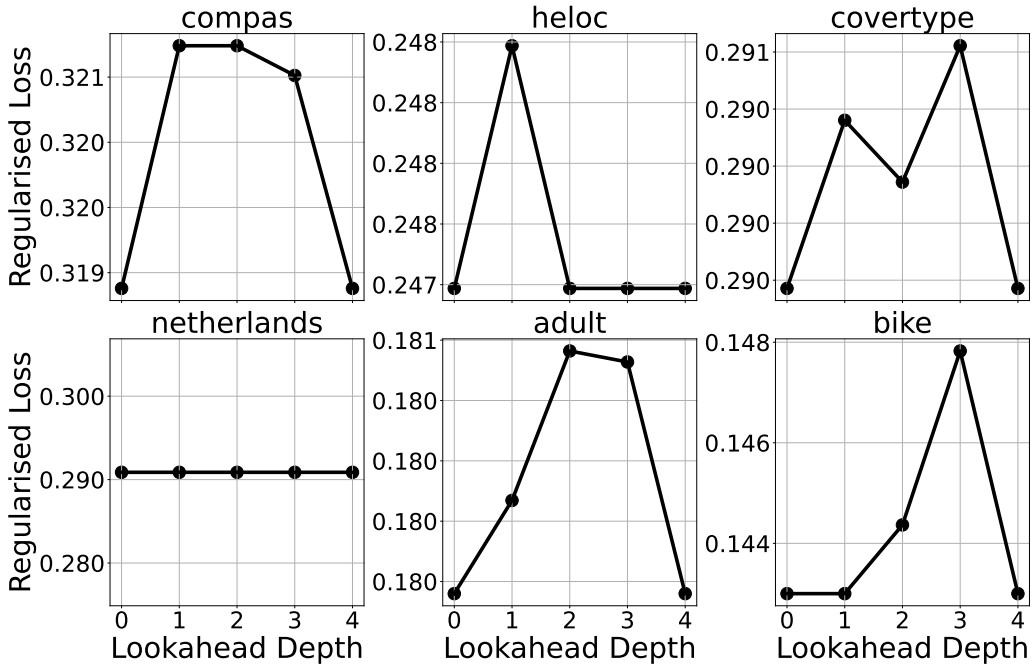

*Figure 10.* Regularised **training loss** as a function of the lookahead depth. $\lambda = 0.001$

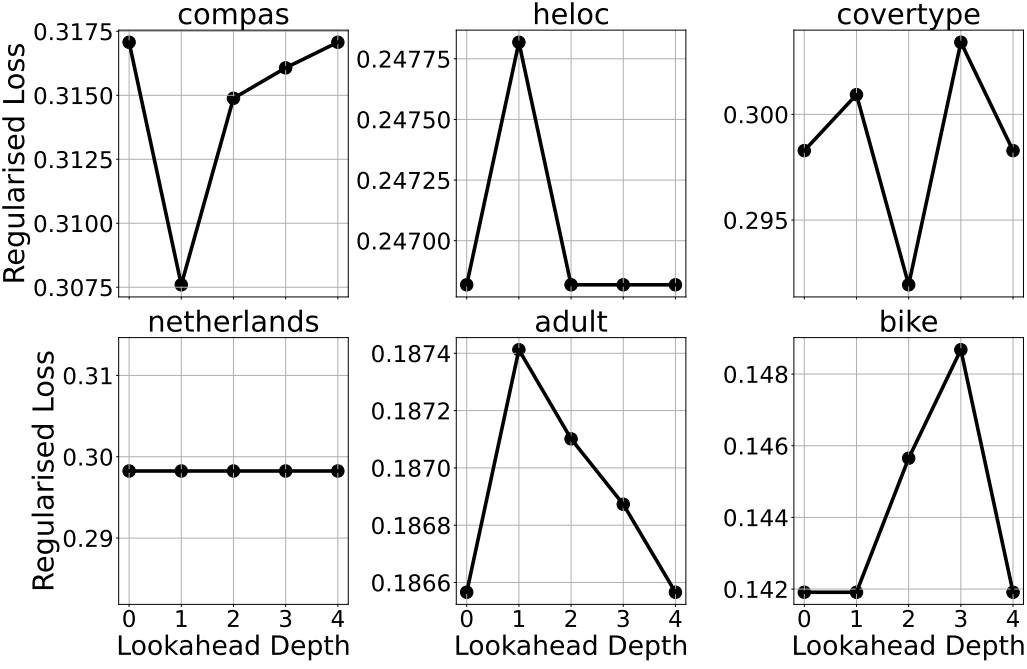

*Figure 11.* Regularised **test loss** as a function of the lookahead depth. $\lambda = 0.001$

From the figures, we see that there indeed exists an optimal lookahead depth that minimizes the runtime of SPLIT. At this depth, however, there is only a small increase in regularised training loss. Surprisingly, the test loss can also be lower at the runtime minimizing depth.

### A.3.2. ARE SPLIT TREES IN THE RASHOMON SET?

This evaluation characterises the near optimal behaviour of trees produced by our algorithms. In particular, we're interested in understanding how often trees produced by our algorithms lie in the Rashomon set. To do this, we sweep over values of $\lambda$. For each $\lambda$, we first generate SPLIT and LicketySPLIT trees and compute the minimum value of $\epsilon$ needed such that they are in the corresponding Rashomon set of decision trees with depth budget $5$ – this is denoted by the respective frontiers of both algorithms.

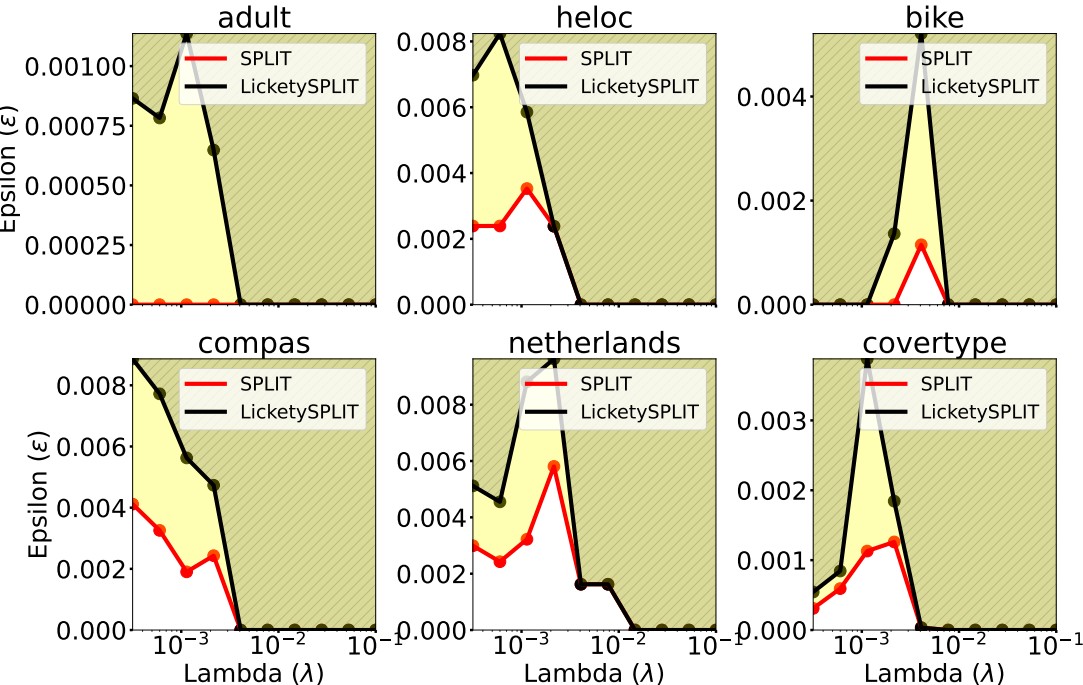

*Figure 12.* An illustration of near-optimality of our algorithms for depth budget 5. The light yellow region represents the $(\lambda, \epsilon)$ configurations for which only SPLIT produces trees in the Rashomon set, while the darker region represents $(\lambda, \epsilon)$ values for which both SPLIT and LicketySPLIT produce trees in the Rashomon set. The figure shows that our trees are almost always in the Rashomon set even for small values of $(\epsilon, \lambda)$.

Figure 12 shows that this minimum $\epsilon$ is small regardless of the value of $\lambda$. While SPLIT has a smaller minimum $\epsilon$, implying a lower optimality gap, particularly noteworthy is the performance of LicketySPLIT. Despite admitting a polynomial runtime, it manages to lie in the Rashomon set even for $\epsilon$ as small as $10^{-3}$.

### A.3.3. SPLIT WITH OPTIMALITY PRESERVING DISCRETIZATION

In this section, we briefly consider how SPLIT performs under full binarization of the dataset. For a given dataset, we perform full binarization by collecting every possible threshold (i.e. split point) present in every feature. We then compare the resulting regularised test loss and runtimes to that of threshold guessing.

- For this experiment, we first randomly choose 2000 examples from the Netherlands, Covertype, HELOC, and Bike datasets. Larger dataset sizes would produce around $10^5$ features for the fully binarized dataset, which would make optimization extremely expensive computationally.

- We then produce two versions of the dataset – a fully binarized version (which contains around 3000-5000 features for each dataset), and a threshold-guessed version (McTavish et al., 2022) with `num_estimators` = 200. The latter ensures that the number of features in the resulting datasets is between 40-60.

For a given dataset, let $D^*$ and $D_{tg}$ its the fully binarized and threshold guessed version. We then run SPLIT and

LicketySPLIT on these datasets and compute the difference in regularised training loss between $D^*$ and $D_{tg}$. Figure 13 shows the resulting difference and the corresponding runtimes.

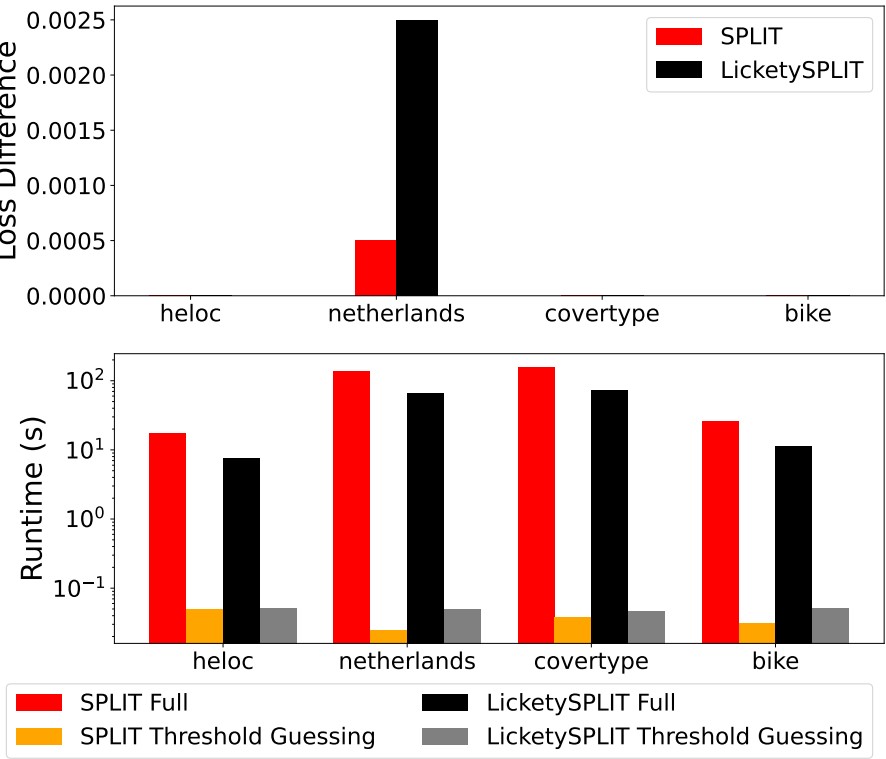

*Figure 13.* Difference in regularised training loss between SPLIT / LicketySPLIT trained on a fully binarized dataset vs the same dataset binarized using threshold guessing. We set $\lambda = 0.01$.

We see that there is almost no difference in loss between the fully binarized dataset and the threshold guessed dataset, suggesting that there is minimial sacrifice in performance when using SPLIT / LicketySPLIT with threshold guessing. Furthermore, using threshold guessing results in runtimes that are orders of magnitude faster. These observations have also been corroborated by McTavish et al. (2022), though in the context of vanilla GOSDT.

### A.3.4. WHAT IS THE PERFORMANCE GAP BETWEEN GOSDT POST-PROCESSING FOR SPLIT / LICKETYSPLIT AND PURELY GREEDY POST-PROCESSING?

We now examine the the additional improvement brought about by the GOSDT post-processing scheme for SPLIT and the recursive post-processing. We next illustrate the gap between SPLIT / LicketySPLIT trees and a tree that is trained purely using a lookahead strategy and behaving purely greedily subsequently. Concretely, we first solve Equation 8, i.e:

$$
\mathcal{L}(D, d', \lambda) = \begin{cases} \lambda + \min\left\{\frac{|D^-|}{|D|}, \frac{|D^+|}{|D|}\right\} & \text{if } d' = 0 \\ \lambda + \min\left\{\frac{|D^-|}{|D|}, \frac{|D^+|}{|D|}, \min_{f \in \mathcal{F}}\left\{L\Big(T_g\big(D(f), d', \lambda\big)\Big) + L\Big(T_g\big(D(\bar{f}), d', \lambda\big)\Big)\right\}\right\} & \text{if } d' = d - d_l \\ \lambda + \min\left\{\frac{|D^-|}{|D|}, \frac{|D^+|}{|D|}, \min_{f \in \mathcal{F}}\left\{\mathcal{L}\Big(D(f), d' - 1, \lambda\Big) + \mathcal{L}\Big(D(\bar{f}), d' - 1, \lambda\Big)\right\}\right\} & \text{if } d' > d - d_l. \end{cases}
$$
(8)

Let $T_{\mathcal{L},g}$ be the tree representing the solution to this equation - this is a lookahead prefix tree with greedy splits after depth $d_l$. Let $T_{SPLIT}$ be the tree that replaces the greedy subtree after depth $d_l$ with optimal GOSDT splits - this refers to lines 3-9 in Algorithm 2. Let $T_{LSPLIT}$ be the tree that replaces the greedy subtree after depth $d_l$ with recursive LicketySPLIT subtrees

(this refers to lines 3-7 in Algorithm 3). We then vary the value of the sparsity penalty $\lambda \in [10^{-3}, 10^{-1}]$ and compute the post-processing gaps on the training dataset $D$:

$$L(T_{\mathcal{L},g}, D, \lambda) - L(T_{SPLIT}, D, \lambda) \tag{9}$$

$$L(T_{\mathcal{L},g}, D, \lambda) - L(T_{LSPLIT}, D, \lambda) \tag{10}$$

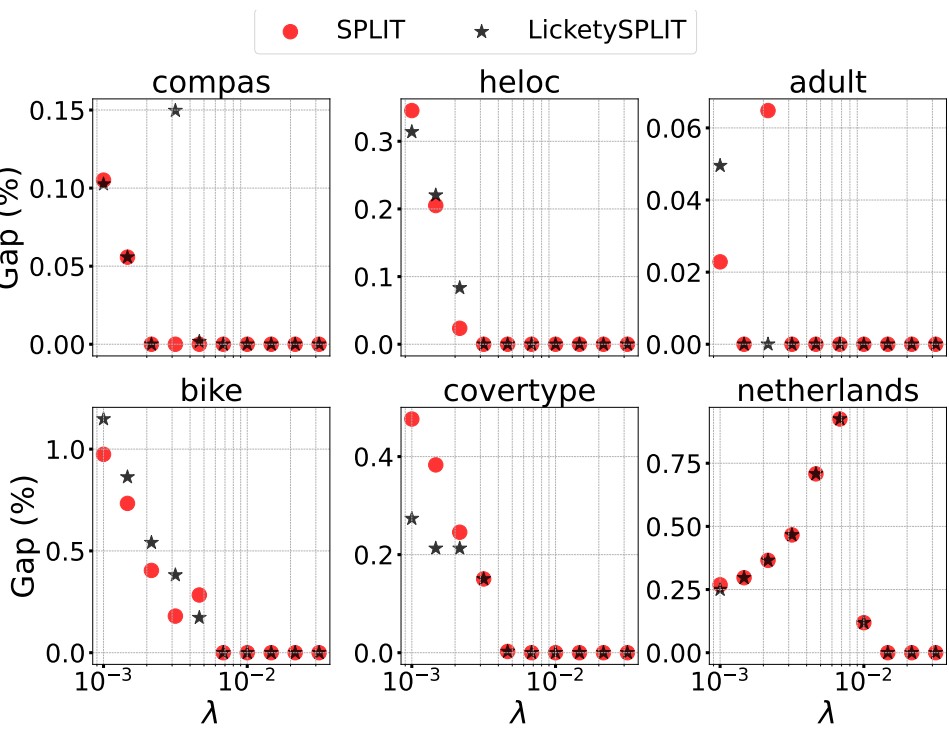

*Figure 14.* Gap (in % points) in accuracy between SPLIT / LicketySPLIT and a lookahead prefix tree followed by a purely greedy approach. Depth budget = 5.

### A.4. SPLIT and LicketySPLIT Scaling Experiments

We now evaluate the scalability of SPLIT and its variants as the number of features increases. For each dataset evaluated, we use the threshold guessing mechanism from (McTavish et al., 2022) to binarize the dataset. In particular:

- We first train a gradient boosted classifier with a specified number of estimators $n_{est}$. Each estimator is a single decision tree stump with an associated threshold.

- We then collect all the thresholds generated during the boosting process, order them by Gini variable importance, and remove the least important thresholds (i.e., any thresholds which result in any performance drop)

In this experiment, we choose $n_{est}$ in a logarithmically spaced interval between 20 and $10^4$, to obtain binary datasets with 10-1000 features. We set a conservative value of $\lambda = 0.001$ for SPLIT / LicketySPLIT, as from Figure 12, this ensures that the optimality gap for our method is around $\sim 10^{-3}$.

Figure 15 shows the results of this experiment.

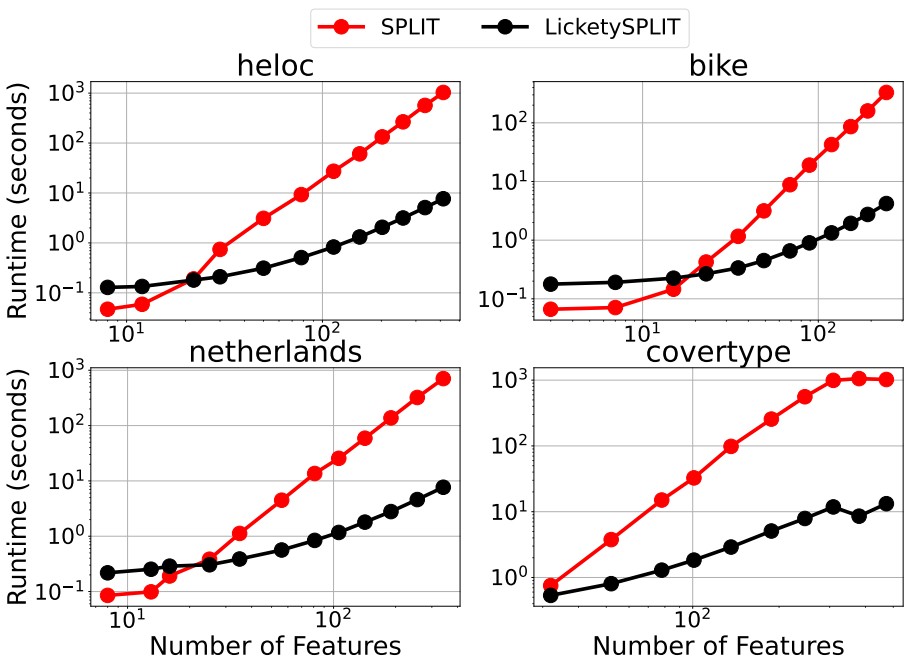

*Figure 15.* Runtime of SPLIT and LicketySPLIT as the number of features increases. $\lambda = 0.001$

## A.5. Rashomon Importance Distribution Under RESPLIT vs TreeFARMS: Threshold Guessing

In this section, we compare RESPLIT and TreeFARMS in terms of their ability to generate meaningful variable importances under the Rashomon Importance Distribution (Donnelly et al., 2023). This analysis is a more complete representation of that in Table 1. The variable importance metric considered is Model Reliance (MR) - the precise details of how this is computed are in (Donnelly et al., 2023).

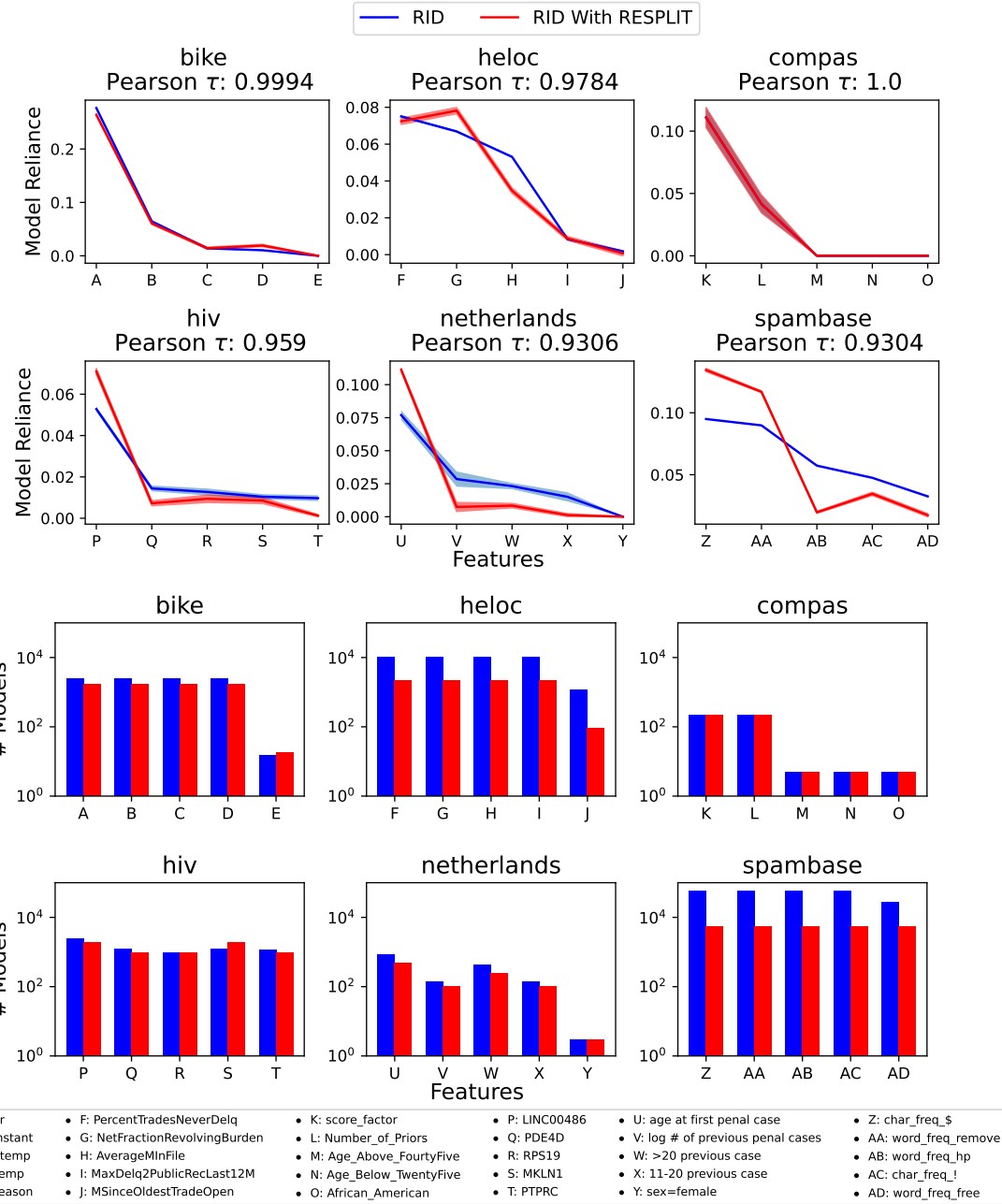

*Figure 16.* (top) Model Reliance for the top 5 features when the Rashomon Importance Distribution is computed in its original form (with TreeFARMS), and when RESPLIT is used as the Rashomon set generating algorithm. The reported Pearson correlation is computed between the top 20 features. We see that it is very close to 1, i.e. features that are important under RID will also remain important when RESPLIT is used. (bottom) The number of models across the bootstrapped Rashomon sets which split on a given feature. We note from the bar plots that RESPLIT is also able to generate a large number of trees - often times as many as TreeFARMS.

**Parameters**: $\lambda = 0.02, \epsilon = 0.01$, # bootstrapped datasets = 10, depth budget = 5, lookahead depth = 3.

## A.6. Rashomon Importance Distribution Under RESPLIT: Quantile Binarization

In this section, we show similar results as in the previous section, but when datasets are binarized using feature quantiles. We chose 3 quantiles per feature (corresponding to each 3rd of the distribution), resulting in datasets with $3\times$ the number of features. For most of these datasets, RID with TreeFARMS failed to run in reasonable time.

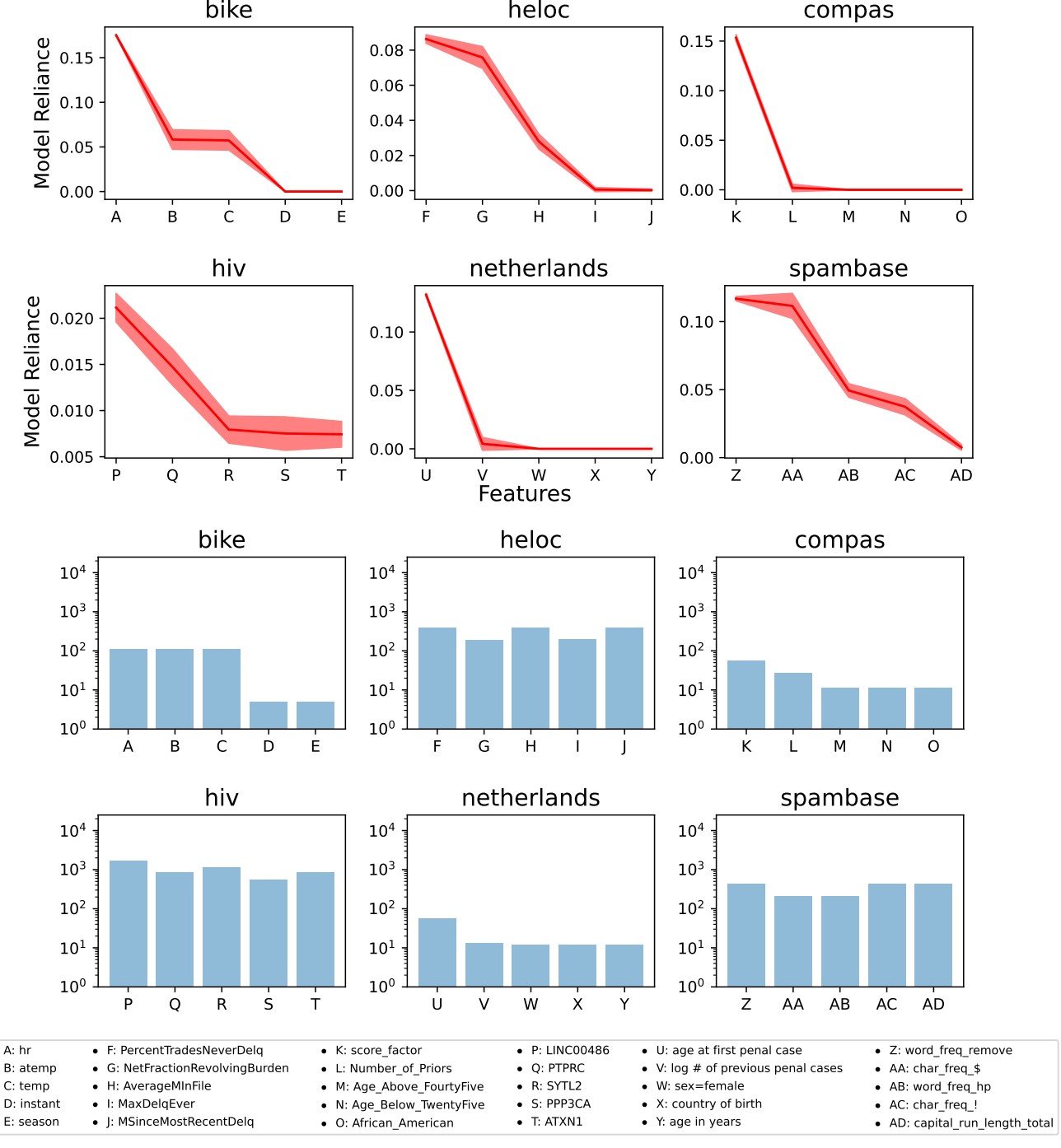

*Figure 17.* (top) Model Reliance under RESPLIT for the top 5 features. (bottom) The number of models across the bootstrapped Rashomon sets which split on a given feature. We note that the features which are important for RESPLIT under threshold guessing are also similarly important under quantile binarization, suggesting that our approach can generalize to different binarization schemes.
**Parameters**: $\lambda = 0.02, \epsilon = 0.01$, # bootstrapped datasets = 10, depth budget = 5, lookahead depth = 3.

## A.7. Experimental Setup

### A.7.1. DATASETS

In this paper, we performed experiments with 10 datasets:

- The **Home Equity Line of Credit (HELOC)** (FICO, 2018) dataset used for the Explainable ML Challenge. This dataset aims to predict the risk of loan default given the credit history of an individual. It consists of 23 features related to financial history, including FICO (credit) score, loan amount, number of delinquent accounts, credit inquiries, and other credit performance indicators. The dataset contains approximately 10,000 instances.

- Two recidivism datasets **(COMPAS and Netherlands)**. COMPAS aims to predict the likelihood of recidivism (reoffending) for individuals who have been arrested. The dataset consists of approximately 6000 instances and includes 11 features including demographic attributes, criminal history, risk of general recidivism, and chargesheet information. The Netherlands dataset is a similar recidivism dataset containing demographic and prior offense features for individuals, used to predict reoffending risk.

- The **Covertype** dataset, which aims to predict the forest cover (one of 7 types) for areas of the Roosevelt National Forest in northern Colorado, based on cartographic data. It contains 54 attributes derived from US Geological Survey data. These include continuous variables like elevation, aspect, slope, and others related to soil type and climate. The dataset has over 580,000 instances, each corresponding to a 30m × 30m patch of the forest.

- The **Adult** dataset, which aims to predict whether an individual's income exceeds \$50000 per year based on demographic and occupational information. It contains around 50000 train and test examples, with 14 features.

- The **Bike** dataset (Fanaee-T & Gama, 2013), which contains a two-year historical log of bikeshare counts from 2011-2012 in Washington D.C., USA. It contains features relating to the weather at every hour – with the aim being to predict the number of bike rentals in the city in that given time period.

- The **Hypothyroid** dataset, which contains medical records used to predict whether a patient has hypothyroidism based on thyroid function test results and other medical attributes. It includes categorical and continuous variables such as TSH (thyroid-stimulating hormone) levels, age, and presence of goiter, with thousands of instances.

- The **Spambase** dataset, which consists of email data used to classify messages as spam or not spam. The dataset contains 57 features extracted from email text, such as word frequencies, capital letter usage, and special character counts, with around 4600 instances.

- The **Bank** dataset, which is used to predict whether a customer will subscribe to a bank term deposit based on features like age, job type, marital status, education level, and past marketing campaign success. It consists of approximately 4500 instances with 16 attributes.

- The **HIV** dataset contains RNA samples from 2 patients. The labels correspond to whether the observed HIV viral load is high or not.

One reason for our choice of datasets was that we wanted to stress-test our methods in scenarios where the dataset has $\mathcal{O}(10^3 - 10^5)$ examples - our smallest dataset has 2623 examples and the largest almost has almost 600000 examples. There are a number of datasets from prior work (e.g. Monk1, Monk2, Monk3, Iris, Moons, Breast Cancer) which only have $\mathcal{O}(10^2)$ examples - for these, many optimal decision tree algorithms are fast enough (i.e. operating in the sub-second regime) that limits any practical scalability improvements. Our aim was to go from the $\mathcal{O}$(hours) regime to the sub-1 second regime, hence, we chose datasets whose size would best reflect the performance improvements we were hoping to showcase.

### A.7.2. PREPROCESSING

- We first exclude all examples with missing values

- We correct for class imbalances by appropriately resampling the majority class. This was the most prevalent in the HIV dataset, where we observed a 90 : 10 class imbalance. We corrected this by randomly undersampling the majority class.

- All datasets have a combination of categorical and continuous features, while SPLIT / LicketySPLIT / RESPLIT and many other decision tree algorithms require binarization of features. We therefore use the threshold guessing mechanism of binarization from McTavish et al. (2022), which can handle both these feature types. In particular:

  - We first train a gradient boosted classifier with a specified number of estimators $n_{est}$. Each estimator is a single decision tree stump with an associated threshold.
  - We then collect all the thresholds generated during the boosting process, order them by Gini variable importance, and remove the least important thresholds (i.e., any thresholds which result in any performance drop)

We store three binarized versions of each dataset for experiments with SPLIT and LicketySPLIT:

  - For version 1, we chose $n_{est}$ for each dataset such that the resulting binarized dataset has between 40-100 features. This is the version used for experiments in Figures 4, 5, 6, 7 when we compare SPLIT / LicketySPLIT with other datasets.
  - For version 2, we chose $n_{est}$ for each dataset such that the resulting dataset has around 20-25 features. This is the version used when we use the TreeFARMS algorithm (Xin et al., 2022) to generate Rashomon sets to explore the properties of near optimal decision trees, as TreeFARMS can be very slow otherwise. Figures 2 and 8 use this version of the datasets.
  - We additionally store another version of the datasets which is fully binarized, i.e., every possible split point is considered. Section A.3.3 uses this version of the dataset is to justify the use of threshold guessing in the context of our algorithm.

- Additionally, for aggressively binarized version of the dataset (i.e., version 2), we subsample Covertype so that it has $\approx 20000$ examples. This is again to ensure that the TreeFARMS algorithm runs in a reasonable amount of time.

We also show scaling experiments for our algorithms, which are described in Section A.4.

| Data Set | Samples | # Features | # Features After Binarization | # Features After Aggressive Binarization |
|---|---|---|---|---|
| HELOC | 10459 | 24 | 62 | 23 |
| COMPAS | 6172 | 12 | 39 | 24 |
| Adult | 32561 | 15 | 65 | 23 |
| Netherlands | 20000 | 10 | 52 | 23 |
| Covertype | 581012 | 55 | 41 | 21 |
| Bike | 17379 | 17 | 99 | 23 |
| Spambase | 4600 | 57 | 78 | 23 |
| Hypothyroid | 2643 | 30 | 72 | 23 |
| Bank | 4521 | 16 | 67 | 23 |

*Table 3.* Characteristics of the 9 datasets tested in this paper for LicketySPLIT and SPLIT experiments. We generate two binarized versions of each dataset using the threshold guessing mechanism in (McTavish et al., 2022) which are used for different sets of experiments.

| Data Set | Samples | # Features | # Features After Binarization |
|---|---|---|---|
| HELOC | 10459 | 24 | 47 |
| COMPAS | 6172 | 12 | 39 |
| Netherlands | 20000 | 10 | 52 |
| Bike | 17379 | 17 | 99 |
| Spambase | 4600 | 57 | 78 |
| HIV | 4521 | 100 | 57 |

*Table 4.* Characteristics of the 6 datasets tested in this paper for RESPLIT experiments. As in Table 3, we use the threshold guessing mechanism for binarization.

A.7.3. DETAILS OF COMPARATIVE EXPERIMENTS FOR SPLIT AND LICKETYSPLIT

- **Greedy**: This is the standard scikit-learn DecisionTreeClassifier class that implements CART. We vary the sparsity of this algorithm by changing the `min_samples_leaves` argument. This is the minimum number of examples required to be in a leaf in order for CART to make further a split at that point.

- **GOSDT** (Lin et al., 2020): We vary the sparsity parameter $\lambda$, choosing equispaced values from $0.001$ to $0.02$.

- **SPLIT / LicketySPLIT**. We search over the same $\lambda$ values as GOSDT. For SPLIT, additionally, we set the lookahead depth to be 1.

- **Thompson Sampled Decision Trees (TSDT)** (Chaouki et al., 2024): Following the practices described in the Appendix Section $B$ of their paper, we fix the following parameters:

    - $\gamma = 0.75$
    - Number of iterations $= 10000$

  Additionally, we also fix the following parameters, based on the Jupyter notebooks in the Github repository of TSDT.

    - `thresh_tree` $= -1e - 6$
    - `thresh_leaf` $= 1e - 6$
    - `thresh_mu` $= 0.8$
    - `thresh_sigma` $= 0.1$

  To obtain different levels of sparsity, we vary the $\lambda$ parameter. We experiment with 3 values of $\lambda : \{0.0001, 0.001, 0.01\}$. For each value of $\lambda$, we also experiment with different time limits for the algorithm: $\{1, 10, 100, 1000\}$. Lastly, we use the FAST-TSDT version of their code, as according to the paper, it strikes a good balance between speed and performance (which is consistent with our paper's motivation).

- **Murtree** (Demirović et al., 2022): For this method, we vary the `max_num_nodes`, which is the hard sparsity constraint imposed by Murtree on the number of leaves.

    - For depth 5 trees, we choose `max_num_nodes` in the set $\{4, 5, 6, 7, 8, 9, 10, 11\}$.
    - For depth 4 trees, we choose `max_num_nodes` in the set $\{3, 4, 5, 6, 7\}$.

- **MAPTree** (Sullivan et al., 2024): The paper has two hyperparameters, $\alpha$ and $\beta$, which in theory control for sparsity in theory by adjusting the prior. However, the authors show that MAPTree does not exhibit significant sensitivity to $\alpha$ and $\beta$ across any metric. Therefore, the only parameter we choose to vary for this experiment is `num_expansions`. We chose 10 values of this parameter in a logarithmically spaced interval from $[10^0, 10^{3.5}]$.

- **Top-k (DL8.5)** (Blanc et al., 2024): The paper also does not specify how to vary the sparsity parameter - hence, we vary $k$ from 1-10.

Note that there is no depth budget hyperparameter for MAPTree and TSDT, but we still show these algorithms across all experiments for comparative purposes.

**Our method vs another bespoke-greedy approach**    We briefly discuss another decision tree optimization algorithm from (Balcan & Sharma, 2024) that demonstrates good performance on a tabular dataset. This method first proposes a novel greediness criterion called the $(\alpha, \beta)$-Tsallis entropy, defined as:

$$g(\alpha, \beta) = \frac{C}{\alpha - 1} \left( \left( 1 - \sum_{i=1}^{c} p_i^{\alpha} \right)^{\beta} \right) \tag{11}$$

where $P = \{p_i\}$ is a discrete probability distribution. Then a decision tree is trained in CART-like fashion, but with this greediness criteria instead. Note that

For the COMPAS dataset, which is one of the smaller datasets in our experiments, we conducted a brief evaluation by averaging results over 3 trials for 3 different values of the hyperparameters $\alpha$ and $\beta$, arranged in a grid-based configuration as defined in (Balcan & Sharma, 2024). These hyperparameters influence the functional form of the above greedy heuristic. Below, we summarize the key settings and observations:

- **Values of** $\alpha$: [0.5, 1, 1.5]

- **Values of** $\beta$: [1, 2, 3]

**Observations**:

1. The method in (Balcan & Sharma, 2024) achieves approximately **31.6% test error** with around **10 leaves**, requiring an average of **10 minutes** to train for a single hyperparameter setting. Another thing to note is that it isn't clear a-priori which hyperparameter will lead to the best performance (in terms of the desired objective in Equation 1), so many different combinations of hyper-parameters might need to be tested in order to find a well-performing tree.

2. **SPLIT** achieves approximately **31.9% test error** with fewer than **10 leaves** in approximately **1 second**.

3. **LicketySPLIT** achieves approximately **31.9% test error** with fewer than **10 leaves** in under **1 second**.

In summary, our proposed methods are over **600x faster** than (Balcan & Sharma, 2024), with a negligible difference in test performance.

**A Note on Comparative Experiments with Blossom (Demirović et al., 2023)**  We briefly compare SPLIT with Blossom, an anytime decision tree algorithm incorporates greedy heuristics to guide search order, albeit in a bottom up manner. To our understanding, Blossom has no hyperparameters we can tune (except depth budget, min size, and min depth), which limits its flexibility in adapting to various datasets.

| Dataset | Runtime (s) | | Test Loss | | # Leaves | |
|---|---|---|---|---|---|---|
| | Blossom | LicketySPLIT | Blossom | LicketySPLIT | Blossom | LicketySPLIT |
| compas | 0.442 [0.381, 0.476] | **0.334** [0.332, 0.336] | 0.314 [0.303, 0.323] | 0.317 [0.305, 0.329] | 32.0 [32.0, 32.0] | **8.0** [8.0, 8.0] |
| adult | 16.223 [15.556, 16.744] | **1.459** [1.453, 1.465] | 0.177 [0.175, 0.179] | **0.177** [0.173, 0.181] | 32.0 [32.0, 32.0] | **7.3** [6.4, 8.3] |
| netherlands | 6.161 [6.128, 6.194] | **0.627** [0.622, 0.632] | 0.287 [0.282, 0.291] | 0.292 [0.288, 0.296] | 32.0 [32.0, 32.0] | **7.7** [6.7, 8.6] |
| heloc | 27.179 [26.810, 27.548] | **0.510** [0.502, 0.518] | 0.286 [0.281, 0.291] | **0.294** [0.286, 0.303] | 32.0 [32.0, 32.0] | **7.0** [6.2, 7.8] |
| spambase | 18.334 [18.200, 18.478] | **0.487** [0.482, 0.492] | 0.090 [0.085, 0.094] | **0.087** [0.085, 0.088] | 32.0 [32.0, 32.0] | **13.7** [13.2, 14.1] |
| bike | 158.744 [157.138, 159.964] | **1.679** [1.633, 1.725] | 0.112 [0.108, 0.115] | **0.121** [0.117, 0.125] | 32.0 [32.0, 32.0] | **12.3** [11.9, 12.8] |
| bank | 11.452 [11.053, 11.673] | **0.353** [0.346, 0.360] | 0.105 [0.097, 0.112] | **0.103** [0.098, 0.108] | 32.0 [32.0, 32.0] | **9.0** [8.2, 9.8] |
| hypothyroid | 2.799 [2.699, 2.909] | **0.206** [0.198, 0.214] | 0.004 [0.002, 0.006] | **0.001** [0.000, 0.002] | 17.8 [15.4, 20.2] | **6.0** [6.0, 6.0] |
| covertype | 13.617 [13.244, 13.861] | **11.864** [11.577, 12.151] | 0.237 [0.236, 0.238] | 0.242 [0.241, 0.243] | 32.0 [32.0, 32.0] | **5.0** [5.0, 5.0] |

*Table 5.* Comparison of Blossom and LicketySPLIT across datasets when Blossom is allowed to finish. We show the LicketySPLIT configuration (after grid-search across $\lambda$) that yielded the best test loss on that dataset (averaged across 5 trials with depth budget 5). Only the mean is bolded when LicketySPLIT performs better. Values are reported as mean [lower, upper] (indicating 95% confidence intervals).

From Table 5, we see that, despite being allowed to run to completion (taking over $10\times$ longer in many cases), Blossom often underperforms LicketySPLIT in test loss. Furthermore, it is much less sparse than LicketySPLIT, having over $4\times$ as many leaves for similar test performances.

We ran another experiment to examine Blossom's anytime performance. In order to facilitate a fair comparison, we made Blossom run for approximately the same amount of time as LicketySPLIT (i.e. generally $\sim 1$ second) on a given dataset. Table 6 shows the best performing tree found by LicketySPLIT (found by varying $\lambda$ and computing the resulting test loss) for each dataset compared with a tree found by Blossom (depth 5). We see that, given comparable runtimes, LicketySPLIT often achieves lower test loss with much fewer leaves compared to Blossom (which mostly branches out to 32 leaves).

| Dataset | Runtime (s) | | Test Loss | | # Leaves | |
|---|---|---|---|---|---|---|
| | Blossom | LicketySPLIT | Blossom | LicketySPLIT | Blossom | LicketySPLIT |
| compas | 0.436 [0.404, 0.454] | **0.334** [0.332, 0.336] | 0.314 [0.303, 0.323] | 0.317 [0.305, 0.329] | 32.000 [32.000, 32.000] | **8.000** [8.000, 8.000] |
| heloc | 0.996 [0.992, 1.000] | **0.510** [0.502, 0.518] | 0.285 [0.281, 0.289] | 0.294 [0.286, 0.303] | 32.000 [32.000, 32.000] | **7.000** [6.184, 7.816] |
| bike | 1.896 [1.882, 1.908] | **1.679** [1.633, 1.725] | 0.128 [0.125, 0.133] | **0.121** [0.117, 0.125] | 30.000 [30.000, 30.000] | **12.333** [11.862, 12.805] |
| covertype | 14.297 [13.846, 14.604] | **11.864** [11.577, 12.151] | 0.237 [0.236, 0.238] | 0.242 [0.241, 0.243] | 32.000 [32.000, 32.000] | **5.000** [5.000, 5.000] |
| adult | 1.722 [1.717, 1.727] | **1.459** [1.453, 1.465] | 0.176 [0.174, 0.179] | **0.176** [0.173, 0.181] | 32.000 [32.000, 32.000] | **7.333** [6.390, 8.276] |
| netherlands | 1.238 [1.230, 1.246] | **0.627** [0.622, 0.632] | 0.284 [0.280, 0.287] | 0.292 [0.288, 0.296] | 32.000 [32.000, 32.000] | **7.667** [6.724, 8.610] |
| bank | 0.771 [0.765, 0.777] | **0.353** [0.346, 0.360] | 0.107 [0.099, 0.112] | **0.103** [0.098, 0.108] | 32.000 [32.000, 32.000] | **9.000** [8.184, 9.816] |
| hypothyroid | 0.649 [0.635, 0.663] | **0.206** [0.198, 0.214] | 0.004 [0.002, 0.006] | **0.001** [0.000, 0.002] | 18.800 [16.400, 21.200] | **6.000** [6.000, 6.000] |
| spambase | 0.832 [0.811, 0.856] | **0.487** [0.482, 0.492] | 0.091 [0.085, 0.097] | **0.087** [0.085, 0.088] | 32.000 [32.000, 32.000] | **13.667** [13.196, 14.138] |

*Table 6.* Comparison of Blossom and LicketySPLIT in an anytime setting (i.e. Blossom execution is stopped around the same time as LicketySPLIT) . We show the LicketySPLIT configuration (after grid-search across $\lambda$) that yielded the best test loss on that dataset (averaged across 5 trials with depth budget 5). Only the mean is bolded where LicketySPLIT outperforms Blossom. Values are reported as mean [lower, upper] (indicating $95\%$ confidence intervals). Note that the Blossom algorithm is not able to explicitly account for sparsity, hence it always returns fully grown trees up to a given depth.

### A.7.4. DESCRIPTION OF MACHINES USED

All experiments were performed on an institutional computing cluster. This was a single Intel(R) Xeon(R) Gold 6226 machine with a 2.70GHz CPU. It has 300GB RAM and 24 cores.

## A.8. Appendix Proofs

**Theorem A.1.** *Consider $T$, a tree output by LicketySPLIT, and $T'$, a tree output by a method which is constrained to only make an information-gain-maximizing split at each node (or not to split at all). Then, considering the training set objective from Equation 1 for training set $D$ and given depth constraint $d$, we have: $L(T, D, \lambda) \leq L(T', D, \lambda)$.*

*Proof.* The proof will proceed by induction.

**Base Case**:

When there is insufficient remaining depth to split, or no split improves the objective, then $T'$ and $T$ both return a leaf with equivalent performance.

**Inductive Step:**

$T$ considers the split that $T'$ would make (a greedy split), and evaluates the resulting performance of a greedy tree after that split. It also considers all other splits, and evaluates the performance of a greedy tree after that split. It either picks the split that $T'$ would make, or it picks one that will correspond to a tree better than $T'$, assuming that the objective after the first split is at least as good as a greedy tree past that first split (which, by the inductive hypothesis, we know is true).

Thus, by induction LicketySPLIT will do at least as well as $T'$.

We can, of course, extend this to SPLIT fairly trivially, since SPLIT is more rigorous than LicketySPLIT. The splits up to the lookahead depth are optimal assuming the continuation past the lookahead depth is at least as good as a greedy method (so SPLIT will either start with greedy splits up to the lookahead, matching $T'$, or it will find some better prefix with respect to the training objective). The post-processing further improves the performance of SPLIT relative to $T'$. $\qquad\square$

### A.8.1. PROOF OF THEOREM 6.1

**Theorem 6.1** (Runtime Complexity of SPLIT)**.** *For a dataset $D$ with $k$ features and $n$ samples, depth constraint $d$ such that $d \ll k$, and lookahead depth $0 \leq d_l < d$, Algorithm 2 has runtime $\mathcal{O}\big(n(d - d_l)k^{d_l+1} + nk^{d-d_l}\big)$. If we cache repeated subproblems, the runtime reduces to $\mathcal{O}\Big(\frac{n(d-d_l)k^{d_l+1}}{d_l!} + \frac{nk^{d-d_l}}{(d-d_l)!}\Big)$.*

*Proof.* We divide the computation process into two stages:

- **Stage** 1 involves computing the lookahead tree prefix. There are $k$ choices to split on at each level, yielding $2k$ nodes at the next level and hence $(2k)^{d_l}$ nodes (sub-problems) at level $d_l$. For each of the $(2k)^{d_l}$ sub-problems at depth $d_l$, we will compute a greedy subtree of depth $d - d_l$. Let $S_i$ be the $i^{th}$ sub-problem at depth $d_l$ (with corresponding size $|S_i|$). The runtime of a greedy decision tree algorithm with depth $d_g$ for a sub-problem of size $n$ and $k$ features is $\mathcal{O}(nkd_g)$ (where $d_g = d - d_l$ in our algorithm). The runtime complexity for this phase is therefore:

$$\mathcal{O}\Big( \sum_{i=1}^{(2k)^{d_l}} |S_i|(k - d_l)(d - d_l) \Big) = \mathcal{O}\left( (k - d_l)(d - d_l) \sum_{i=1}^{(2k)^{d_l}} |S_i| \right) \tag{12}$$

where we have $(k - d_l)$ features remaining to be split on at the end of lookahead. Now,

$$\sum_{i=1}^{(2k)^{d_l}} |S_i| = \mathcal{O}\Big( nk^{d_l} \Big), \tag{13}$$

because at each level, we split on $\mathcal{O}(k)$ features and route $n$ examples down each path. Thus, the runtime for this stage simplifies to:

$$\mathcal{O}\Big( \sum_{i=1}^{(2k)^{d_l}} |S_i|(k - d_l)(d - d_l) \Big) = \mathcal{O}\big( nk^{d_l}(k - d_l)(d - d_l) \big) \tag{14}$$

$$= \mathcal{O}\big( n(d - d_l)k^{d_l+1} \big) \tag{15}$$

where the second equality stems from the fact that $k - d_l = O(k)$, because $d \ll k$ and $d_l < d$. *However*, there is redundancy here, because this expression assumes that all sub-problems at level $d_l$ are unique - this is not the case. Consider a subproblem identified by the sequence of splits $f_1 = 0 \to f_2 = 1 \to f_3 = 0$. The exact order of the splits does not matter in identifying the subproblem. This implies that multiple sequences of splits correspond to the same subproblem, leading to an overestimation of the runtime. At level $d_l$, there are therefore $d_l!$ redundant subproblems (corresponding to the different ways of arranging the sequence of splits). We only need to solve, i.e. compute a greedy tree, for one of them and store the solution for the other identical subproblems. If we cache subproblems in this manner, the final runtime for this stage becomes:

$$\mathcal{O}\Big(\frac{n(d - d_l)k^{d_l+1}}{d_l!}\Big) \tag{16}$$

- **Stage** 2 involves replacing the leaves of the learned prefix tree with an optimal tree of depth $d - d_l$ so that the resulting tree has depth $\leq d$. Let $u$ be a leaf node in this prefix tree and $n_u$ be its corresponding sub-problem size. As before, we will search over all trees of size $d - d_l$, which requires evaluation of $(2k)^{d-d_l}$ nodes in the search tree. This time, however, the evaluation at the last node will be linear in the sub-problem size (as we are not considering any splits beyond depth $d$). By the same argument as Stage 1, the runtime of this phase is therefore $\mathcal{O}\big(k^{d-d_l}n_u\big)$. Summing this across all subproblems $u$, we get $\sum_u \mathcal{O}\big(k^{d-d_l}n_u\big)$. As the total sum of sub-problem sizes across all leaves is equal to the original dataset size, this sum is equal to $\mathcal{O}\big(k^{d-d_l}n\big)$. By the same subproblem redundancy argument as in Stage 1, the final runtime complexity of this stage upon caching redundant subproblems becomes:

$$\mathcal{O}\Big(\frac{nk^{d-d_l}}{(d - d_l)!}\Big) \tag{17}$$

Combining Stages 1 and 2, we get that the total runtime of SPLIT is:

$$\begin{cases} \mathcal{O}\big(n(d - d_l)k^{d_l+1} + nk^{d-d_l}\big) & \text{Without Caching} \\[2ex] \mathcal{O}\Big(\frac{n(d-d_l)k^{d_l+1}}{d_l!} + \frac{nk^{d-d_l}}{(d-d_l)!}\Big) & \text{With Caching.} \end{cases} \tag{18}$$

$\square$

### A.8.2. PROOF OF COROLLARY 6.2

**Corollary 6.2** (Optimal Lookahead Depth for Minimal Runtime). *The optimal lookahead depth that minimizes the asymptotic runtime of Algorithm 2 is $d_l = \frac{(d-1)}{2}$ for large $k$, regardless of whether subproblems are cached.*

*Proof.* We evaluate the optimal lookahead depth in both scenarios, caching and no-caching.

## Case 1: Lookahead Without Caching

In this case, the runtime expression is $\mathcal{O}\Big(n(d - d_l)k^{d_l+1} + nk^{d-d_l}\Big)$. We divide the proof into 6 parts:

**Part 1: Finding the stationary point of the runtime**

Consider the runtime expression from Theorem 6.1. We now minimize this with respect to $d_l$:

$$\frac{\partial}{\partial d_l}\left(n(d-d_l)k^{d_l+1} + nk^{d-d_l}\right) = 0 \tag{19}$$

$$\iff \frac{\partial}{\partial d_l}\left((d-d_l)k^{d_l+1} + k^{d-d_l}\right) = 0 \tag{20}$$

$$\iff \frac{\partial}{\partial d_l}\left(dk^{d_l+1} - d_l k^{d_l+1} + k^{d-d_l}\right) = 0 \tag{21}$$

$$\iff d(\log k)k^{d_l+1} - (k^{d_l+1} + d_l(\log k)k^{d_l+1}) - (\log k)k^{d-d_l} = 0 \tag{22}$$

$$\iff ((d-d_l)\log k - 1)k^{d_l+1} - (\log k)k^{d-d_l} = 0 \tag{23}$$

$$\implies \left((d-d_l) - \frac{1}{\log k}\right)k^{2d_l+1} = k^d. \tag{24}$$

We can now simplify this equation to analytically express the lookahead depth $d_l$ as a function of $k$. To do so, we define a new variable $u$ such that:

$$d_l = \frac{u - 2 + 2d\log k}{2\log k}. \tag{25}$$

Under this definition of $d_l$, we can now rewrite Equation 24 in terms of $u$:

$$\left(\left(d - \frac{u-2+2d\log k}{2\log k}\right) - \frac{1}{\log k}\right)e^{\log k\left(2\left(\frac{u-2+2d\log k}{2\log k}\right)+1\right)} = k^d \tag{26}$$

$$\left(d - \frac{u+2d\log k}{2\log k}\right)e^{u-2+2d\log k+\log k} = k^d \tag{27}$$

$$\Rightarrow \frac{-u}{2\log k}e^{-2}k^{2d+1}e^u = k^d \tag{28}$$

$$ue^u = -2e^2 k^{-(d+1)}\log k. \tag{29}$$

As the solution to this equation is known to be analytically intractable, we express $u$ in terms of the Lambert $W$ function, which is a well known function that cannot be expressed in terms of elementary functions. Denoted by $W(z)$, the Lambert $W$ function satisfies the following equation:

$$W(z)e^{W(z)} = z. \tag{30}$$

From Equation 29, we can express $u$ in terms of $W(.)$, giving us:

$$u = W(-2e^2 k^{-(d+1)}\log k). \tag{31}$$

Substituting this back into the expression for $d_l$ in Equation 25, we get:

$$d_l = \frac{W(-2e^2 k^{-(d+1)}\log k) - 2 + 2d\log k}{2\log k}. \tag{32}$$

**Part 2: Bounding the Lambert $W$ function**

Let $z = -2e^2 k^{-(d+1)}\log k$. For sufficiently large $k$, $z \in [-\frac{1}{e}, 0]$. In this domain, there are two possible values of $W(z)$, $W_0(z)$ and $W_{-1}(z)$, such that $W_0(z) \geq W_{-1}(z)$. Figure 18 shows these two branches of the $W$ function.

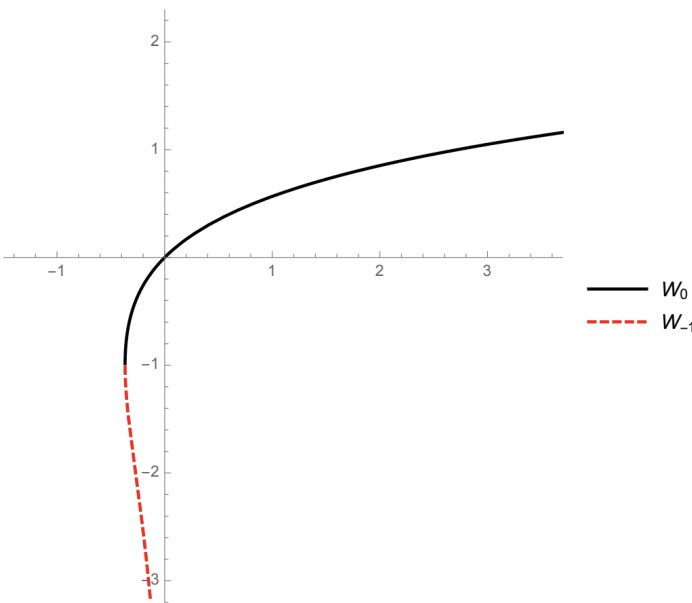

*Figure 18.* The Lambert W function, which has two branches in the real plane, $W_0(z)$ and $W_{-1}(z)$. Figure from (Lóczi, 2021).

For now, consider the function $W_{-1}(z)$. We will show later that choosing this branch of the $W$ function results in the value of $d_l$ that minimizes the runtime.
(Chatzigeorgiou, 2013) show the following lower bound for $W_{-1}(z)$:

$$W_{-1}(z) \geq \log(-z) - \sqrt{2(-1 - \log(-z))}. \tag{33}$$

(Lóczi, 2021) show the following upper bound for $W_{-1}(z)$:

$$W_{-1}(z) \leq \log(-z) - \log(-\log(-z)). \tag{34}$$

Denote the lower bound as $W_{-1}^{lb}(z)$ and the upper bound as $W_{-1}^{ub}(z)$. We can now write upper and lower bounds for the optimal $d_l$ (call this $d_l^*$) in Equation 32.

$$\frac{W_{-1}^{lb}(z) - 2 + 2d\log k}{2\log k} \leq d_l^* \leq \frac{W_{-1}^{ub}(z) - 2 + 2d\log k}{2\log k} \tag{35}$$

where $z = -2e^2 k^{-(d+1)} \log k$ from above.

**Part 3: Lower Bound for $d_l^*$**

We now evaluate the lower bound for $d_l^*$, substituting $z = -2e^2 k^{-(d+1)} \log k$ into the left side of Equation 35:

$$d_l^* \geq \frac{W_{-1}^{lb}(z) - 2 + 2d\log k}{2\log k} \tag{36}$$

$$= \frac{W_{-1}^{lb}\left(-2e^2 k^{-(d+1)} \log k\right) - 2 + 2d\log k}{2\log k} \tag{37}$$

$$= \frac{\log\left(2e^2 k^{-(d+1)} \log k\right) - \sqrt{2\left(-1 - \log\left(2e^2 k^{-(d+1)} \log k\right)\right)} - 2 + 2d\log k}{2\log k} \tag{38}$$

$$= \frac{\log 2 - (d+1)\log k + \log\log k + 2d\log k - \sqrt{-6 - 2\log 2 + 2(d+1)\log k - 2\log\log k}}{2\log k}. \tag{39}$$

Consider the term:

$$\sqrt{-6 - 2\log 2 + 2(d+1)\log k - 2\log\log k}. \tag{40}$$

As $k$ becomes large, we can ignore the constants. Asymptotically, $\log k \gg \log\log k$, and hence this term approaches $\sqrt{2(d+1)\log k}$. Thus, we can write:

$$d_l^* \geq \frac{\log 2 - (d+1)\log k + \log\log k + 2d\log k - \sqrt{2(d+1)\log k}}{2\log k} \tag{41}$$

$$= d - \frac{d+1}{2} + \frac{\log 2}{2\log k} + \frac{\log\log k}{\log k} - \frac{\sqrt{2(d+1)\log k}}{\log k} \tag{42}$$

$$= \frac{d-1}{2} - \mathcal{O}\Big(\frac{1}{\sqrt{\log k}}\Big) \tag{43}$$

as $d$ is a constant and $k \gg d$.

**Part 4: Upper Bound for $d_l^*$**

We can similarly evaluate the upper bound for $d_l^*$:

$$d_l^* \leq \frac{W_{-1}^{ub}(-2e^2 k^{-(d+1)}\log k) - 2 + 2d\log k}{2\log k} \tag{44}$$

$$= \frac{\log\big(2e^2 k^{-(d+1)}\log k\big) - \log\Big(-\log\big(-2e^2 k^{-(d+1)}\log k\big)\Big) - 2 + 2d\log k}{2\log k} \tag{45}$$

$$= \frac{\log 2 - (d+1)\log k + \log\log k - \log\Big(-\big(\log 2 + 2 - (d+1)\log k + \log\log k\big)\Big) + 2d\log k}{2\log k}. \tag{46}$$

Consider the term:

$$\log\Big(-\big(\log 2 + 2 - (d+1)\log k + \log\log k\big)\Big). \tag{47}$$

As $k$ becomes large, we can ignore the constants. We can also consider the asymptotic lower bound of the subsequent expression:

$$\log\Big((d+1)\log k - \log\log k\Big) \geq 1 - \frac{1}{(d+1)\log k - \log\log k} \tag{48}$$

If we plug in this lower bound in Equation 46, the resulting expression is still a valid upper bound.

$$d_l^* \leq \frac{\log 2 - (d+1)\log k + \log\log k - 1 + \frac{1}{(d+1)\log k - \log\log k} + 2d\log k}{2\log k} \tag{49}$$

$$= d - \frac{d+1}{2} + \frac{\log 2 - 1}{2\log k} + \frac{\log\log k}{2\log k} + \frac{1}{2(d+1)\log^2 k - 2\log k \log\log k} \tag{50}$$

$$= \frac{d-1}{2} + \mathcal{O}\Big(\frac{\log\log k}{\log k}\Big). \tag{51}$$

**Part 5: Putting it all together**

Finally, we get the following lower and upper bounds on the optimal lookahead depth $d_l^*$:

$$\frac{d-1}{2} - \mathcal{O}\Big(\frac{1}{\sqrt{\log k}}\Big) \leq d_l^* \leq \frac{d-1}{2} + \mathcal{O}\Big(\frac{\log\log k}{\log k}\Big). \tag{52}$$

For large $k$, these bounds will converge, and hence, in this limit, $d_l^* = \frac{d-1}{2}$.

**Part 6: Verifying that $d_l^* = \frac{d-1}{2}$ is the minimum**

We show that the computed value of $d_l^*$ is indeed the minimum by evaluating the second derivative of the runtime, i.e. $\frac{\partial^2}{\partial d_l^2}\left(n(d-d_l)k^{d_l+1} + nk^{d-d_l}\right)|_{d_l=d_l^*}$.

$$\frac{\partial^2}{\partial d_l^2}\left(n(d-d_l)k^{d_l+1} + nk^{d-d_l}\right) = \frac{\partial}{\partial d_l}\left(n\big((d-d_l)\log k - 1\big)k^{d_l+1} - n(\log k)k^{d-d_l}\right) \tag{53}$$

where we use the the derivative expression from Part 1 of this proof. Simplifying further:

$$\frac{\partial}{\partial d_l}\left(n\big((d-d_l)\log k - 1\big)k^{d_l+1} - n(\log k)k^{d-d_l}\right) \tag{54}$$

$$= \frac{\partial}{\partial d_l}\left(ndk^{d_l+1}\log k - nd_lk^{d_l+1}\log k - nk^{d_l+1} - nk^{d-d_l}\log k\right) \tag{55}$$

$$= ndk^{d_l+1}\log^2 k - nk^{d_l+1}\log k - nd_lk^{d_l+1}\log^2 k - nk^{d_1+1}\log k + nk^{d-d_l}\log^2 k. \tag{56}$$

We now substitute $d_l = \frac{d-1}{2}$ and simplify the result:

$$\frac{\partial^2}{\partial d_l^2}\left(n(d-d_l)k^{d_l+1} + nk^{d-d_l}\right)\bigg|_{d_l=d_l^*} \tag{57}$$

$$= ndk^{\frac{d+1}{2}}\log^2 k - nk^{\frac{d+1}{2}}\log k - n\Big(\frac{d-1}{2}\Big)k^{\frac{d+1}{2}}\log^2 k - nk^{\frac{d+1}{2}}\log k + nk^{\frac{d+1}{2}}\log^2 k \tag{58}$$

$$= n\Big(\frac{d+1}{2}\Big)k^{\frac{d+1}{2}}\log^2 k + nk^{\frac{d+1}{2}}\log^2 k - 2nk^{\frac{d+1}{2}}\log k \tag{59}$$

$$= n\Big(\frac{d+3}{2}\Big)k^{\frac{d+1}{2}}\log^2 k - 2nk^{\frac{d+1}{2}}\log k. \tag{60}$$

This is clearly $> 0$ as the $\log^2 k$ terms dominate $\log k$. Thus, the value $d_l^* = \frac{d-1}{2}$ corresponds to the minimum of the runtime.

## Case 2: Lookahead with Caching

In this case, the runtime expression is $\mathcal{O}\left(\frac{n(d-d_l)k^{d_l+1}}{d_l!} + \frac{nk^{d-d_l}}{(d-d_l)!}\right)$. We divide the proof into 3 parts:

### Part 1: Finding the stationary point of the runtime

We can replace the factorial in the runtime expression with the Gamma function, i.e.:

$$d_l! = \Gamma(d_l + 1) = \int_0^\infty t^{d_l}e^{-t}\, dt \tag{61}$$

as this allows us to apply the derivative operator. Further employing the definition of the Digamma function $\psi(x) = \frac{\Gamma'(x)}{\Gamma(x)}$, we now minimize the runtime with respect to $d_l$:

$$\frac{\partial}{\partial d_l}\left(\frac{n(d-d_l)k^{d_l+1}}{\Gamma(d_l+1)} + \frac{nk^{d-d_l}}{\Gamma(d-d_l+1)}\right) = 0 \tag{62}$$

$$\Rightarrow n\frac{\partial}{\partial d_l}\left(\frac{(d-d_l)k^{d_l+1}}{\Gamma(d_l+1)}\right) + n\frac{\partial}{\partial d_l}\left(\frac{k^{d-d_l}}{\Gamma(d-d_l+1)}\right) = 0 \tag{63}$$

$$\Rightarrow n\left[\frac{\partial}{\partial d_l}\left((d-d_l)k^{d_l+1}\right)\cdot\frac{1}{\Gamma(d_l+1)} - \frac{(d-d_l)k^{d_l+1}}{\Gamma(d_l+1)^2}\cdot\frac{\partial}{\partial d_l}\Gamma(d_l+1)\right] + \tag{64}$$

$$n\left[\frac{\partial}{\partial d_l}\left(k^{d-d_l}\right)\cdot\frac{1}{\Gamma(d-d_l+1)} - \frac{k^{d-d_l}}{\Gamma(d-d_l+1)^2}\cdot\frac{\partial}{\partial d_l}\Gamma(d-d_l+1)\right] = 0 \tag{65}$$

$$\Rightarrow n\left[-k^{d_l+1} + (d-d_l)k^{d_l+1}\log k\right]\cdot\frac{1}{\Gamma(d_l+1)} - n\frac{(d-d_l)k^{d_l+1}}{\Gamma(d_l+1)^2}\cdot\Gamma(d_l+1)\psi(d_l+1) + \tag{66}$$

$$n\left[-k^{d-d_l}\log k\right]\cdot\frac{1}{\Gamma(d-d_l+1)} - n\frac{k^{d-d_l}}{\Gamma(d-d_l+1)^2}\cdot\Gamma(d-d_l+1)\psi(d-d_l+1) = 0 \tag{67}$$

$$\Rightarrow \frac{(-k^{d_l+1} + (d-d_l)k^{d_l+1}\log k)\Gamma(d_l+1) - (d-d_l)k^{d_l+1}\Gamma(d_l+1)\psi(d_l+1)}{\Gamma(d_l+1)^2} + \tag{68}$$

$$\frac{k^{d-d_l}\left(\Gamma(d-d_l+1)\psi(d-d_l+1) - (\log k)\Gamma(d-d_l+1)\right)}{\Gamma(d-d_l+1)^2} = 0 \tag{69}$$

Simplifying this expression, we get:

$$\Rightarrow \frac{(-k^{d_l+1} + (d-d_l)k^{d_l+1}\log k) - (d-d_l)k^{d_l+1}\psi(d_l+1)}{\Gamma(d_l+1)} + \frac{k^{d-d_l}\left(\psi(d-d_l+1) - \log k\right)}{\Gamma(d-d_l+1)} = 0 \tag{70}$$

$$\Rightarrow \frac{k^{d_l+1}\left(-1 + (d-d_l)(\log k - \psi(d_l+1))\right)}{\Gamma(d_l+1)} = \frac{k^{d-d_l}\left(\log k - \psi(d-d_l+1)\right)}{\Gamma(d-d_l+1)} \tag{71}$$

$$\Rightarrow \frac{k^{2d_l-d+1}\left(-1 + (d-d_l)(\log k - \psi(d_l+1))\right)}{\Gamma(d_l+1)} = \frac{\log k - \psi(d-d_l+1)}{\Gamma(d-d_l+1)} \tag{72}$$

$$\Rightarrow k^{2d_l-d+1} = \frac{\left(\log k - \psi(d-d_l+1)\right)\Gamma(d_l+1)}{\Gamma(d-d_l+1)\left(-1 + (d-d_l)(\log k - \psi(d_l+1))\right)}. \tag{73}$$

**Part 2: Bounding the Optimal Lookahead Depth**

Unlike the previous case, it is not possible to derive a closed functional form for the optimal lookahead depth for any given value of $k$ (although we can simulate it numerically). Instead, we need to analyze how this expression behaves as in the limit as $k \to \infty$. Because $k \gg d, d_l$, $\log k \gg \psi(d-d_l+1)$. Similarly, $\log k \gg \psi(d_l+1)$. Furthermore, we can ignore all expressions which are not functions of $k$ as they are insignificant when $k$ is large. Thus, in this limit:

$$k^{2d_l-d+1} \to \frac{\Gamma(d_l+1)\log k}{\Gamma(d-d_l+1)(d-d_l)\log k} \tag{74}$$

$$\Rightarrow k^{2d_l-d+1} = \frac{\Gamma(d_l+1)}{\Gamma(d-d_l+1)(d-d_l)} \tag{75}$$

$$\Rightarrow (2d_l - d + 1)\log k = \log\Gamma(d_l+1) - \log\Gamma(d-d_l+1) - \log(d-d_l) \tag{76}$$

$$\Rightarrow 2d_l - d + 1 = \frac{\log\Gamma(d_l+1) - \log\Gamma(d-d_l+1) - \log(d-d_l)}{\log k}. \tag{77}$$

Observe the term $\log \Gamma(d_l + 1) - \log \Gamma(d - d_l + 1) - \log(d - d_l)$. We can write the factorial form of this expression to understand it better:

$$\log \Gamma(d_l + 1) - \log \Gamma(d - d_l + 1) - \log(d - d_l) = \log \left( \frac{d_l!}{(d - d_l)!(d - d_l)} \right). \tag{78}$$

Notice that for any $d_l$ between $0$ and $\lfloor \frac{d}{2} \rfloor$ (inclusive), the RHS is always less than $0$. Similarly, for $\lfloor \frac{d}{2} \rfloor < d_l \leq d - 1$, the term is always greater than $0$. Given that these are constant as $k$ increases:

$$\log \Gamma(d_l + 1) - \log \Gamma(d - d_l + 1) - \log(d - d_l) = \begin{cases} -\mathcal{O}(1) & 0 \leq d_l \leq \lfloor \frac{d}{2} \rfloor \\ \mathcal{O}(1) & \lfloor \frac{d}{2} \rfloor < d_l \leq d - 1. \end{cases} \tag{79}$$

This implies:

$$-\mathcal{O}\left( \frac{1}{\log k} \right) \leq \frac{\log \Gamma(d_l + 1) - \log \Gamma(d - d_l + 1) - \log(d - d_l)}{\log k} \leq \mathcal{O}\left( \frac{1}{\log k} \right) \tag{80}$$

for all $0 \leq d_l \leq d - 1$ (which are the constraints in our setup). Hence, we conclude that, for large $k$:

$$\mathcal{O}\left( \frac{1}{\log k} \right) \leq 2d_l - d + 1 \leq -\mathcal{O}\left( \frac{1}{\log k} \right) \tag{81}$$

$$\Rightarrow \frac{d - 1}{2} - \mathcal{O}\left( \frac{1}{\log k} \right) \leq d_l \leq \frac{d - 1}{2} + \mathcal{O}\left( \frac{1}{\log k} \right) \tag{82}$$

which approaches $d_l = \frac{d-1}{2}$ as $k \to \infty$. Henceforth, we denote this asymptotically optimal value as $d_l^*$.

**Part 3: Verifying that $d_l^* = \frac{d-1}{2}$ is the minimum for large $k$**

We show that the computed value of $d_l^*$ is indeed the minimum for large $k$ by evaluating the second derivative of the runtime, i.e. $\frac{\partial^2}{\partial d_l^2} \left( \frac{n(d-d_l)k^{d_l+1}}{d_l!} + \frac{nk^{d-d_l}}{(d-d_l)!} \right) \big|_{d_l = d_l^*}$.

$$\frac{\partial^2}{\partial d_l^2} \left( \frac{n(d - d_l)k^{d_l+1}}{d_l!} + \frac{nk^{d-d_l}}{(d-d_l)!} \right) \tag{83}$$

$$= n \frac{\partial}{\partial d_l} \left( \frac{(-k^{d_l+1} + (d-d_l)k^{d_l+1} \log k)\Gamma(d_l + 1) - (d-d_l)k^{d_l+1}\Gamma(d_l+1)\psi(d_l+1)}{\Gamma(d_l+1)^2} + \right. \tag{84}$$

$$\left. \frac{k^{d-d_l}\big(\Gamma(d - d_l + 1)\psi(d - d_l + 1) - (\log k)\Gamma(d - d_l + 1)\big)}{\Gamma(d - d_l + 1)^2} \right). \tag{85}$$

We can remove the dataset size $n$ for simplicity as it doesn't affect the sign of the answer. In the limit as $k \to \infty$, we can simplify this expression and only evaluate terms that grow with $k$:

$$\frac{\partial}{\partial d_l} \left( \frac{\log k (d - d_l) k^{d_l+1} \Gamma(d_l + 1)}{\Gamma(d_l + 1)^2} - \frac{k^{d-d_l} \log k \Gamma(d - d_l + 1)}{\Gamma(d - d_l + 1)^2} \right) \tag{86}$$

$$= \left[ \frac{\partial}{\partial d_l} \left( \log k (d - d_l) k^{d_l+1} \right) \cdot \frac{\Gamma(d_l + 1)}{\Gamma(d_l + 1)^2} - \frac{\log k (d - d_l) k^{d_l+1}}{\Gamma(d_l + 1)^2} \cdot \frac{\partial}{\partial d_l} \Gamma(d_l + 1) \right] - \tag{87}$$

$$\left[ \frac{\partial}{\partial d_l} \left( k^{d-d_l} \log k \right) \cdot \frac{\Gamma(d - d_l + 1)}{\Gamma(d - d_l + 1)^2} - \frac{k^{d-d_l} \log k}{\Gamma(d - d_l + 1)^2} \cdot \frac{\partial}{\partial d_l} \Gamma(d - d_l + 1) \right] \tag{88}$$

$$= \left[ \left( - \log k k^{d_l+1} + (d - d_l) k^{d_l+1} (\log k)^2 \right) \cdot \frac{1}{\Gamma(d_l + 1)} - \frac{\log k (d - d_l) k^{d_l+1}}{\Gamma(d_l + 1)^2} \cdot \Gamma(d_l + 1) \psi(d_l + 1) \right] - \tag{89}$$

$$\left[ \left( - k^{d-d_l} (\log k)^2 \right) \cdot \frac{1}{\Gamma(d - d_l + 1)} - \frac{k^{d-d_l} \log k}{\Gamma(d - d_l + 1)^2} \cdot \Gamma(d - d_l + 1) \psi(d - d_l + 1) \right] \tag{90}$$

$$= \left[ \frac{\left( - \log k k^{d_l+1} + (d - d_l) k^{d_l+1} (\log k)^2 \right)}{\Gamma(d_l + 1)} - \frac{\log k (d - d_l) k^{d_l+1} \psi(d_l + 1)}{\Gamma(d_l + 1)} \right] - \tag{91}$$

$$\left[ \frac{-k^{d-d_l} (\log k)^2}{\Gamma(d - d_l + 1)} - \frac{k^{d-d_l} \log k \psi(d - d_l + 1)}{\Gamma(d - d_l + 1)} \right] \tag{92}$$

$$= \left[ \frac{\left( - k^{d_l+1} (\log k) + (d - d_l) k^{d_l+1} (\log k)^2 - \log k (d - d_l) k^{d_l+1} \psi(d_l + 1) \right)}{\Gamma(d_l + 1)} \right] - \tag{93}$$

$$\left[ \frac{-k^{d-d_l} (\log k)^2 + k^{d-d_l} \log k \psi(d - d_l + 1)}{\Gamma(d - d_l + 1)} \right]. \tag{94}$$

Note that the $(\log k)^2$ terms are dominant in this expression as $k \to \infty$. Hence, at $d_l^* = \frac{d-1}{2}$, the terms that will affect the sign of the expression are:

$$k^{\frac{d+1}{2}} \left( \frac{d+1}{2} \right) (\log k)^2 - k^{\frac{d+1}{2}} (\log k)^2 + \mathcal{O}(\log k). \tag{95}$$

This is clearly positive for any $d > 1$, hence the value $d_l^* = \frac{d-1}{2}$ corresponds to the minimum of the runtime. Note that in practice, if $d_l^*$ is not an integer, we can choose whichever of $\lceil d_l^* \rceil$ or $\lfloor d_l^* \rfloor$ gives us a lower runtime.

From Figure 19, we see that for a depth budget of 5, the minimum lookahead depth $d_l^*$ is slightly less than 2 for both the caching and non-caching case, which is what is predicted by Corollary 6.2. This also lines up nicely with what we observe in practice (e.g. in Figure 9). Note that our algorithms, which build on the GOSDT codebase, cache subproblems by default. $\square$

### A.8.3. PROOF OF COROLLARY 6.3

**Corollary 6.3** (Runtime Savings of SPLIT Relative to Globally Optimal Approaches). *Asymptotically, under the same conditions as Theorem 6.1 and with caching repeated subproblems, Algorithm 2 saves a factor of $\mathcal{O}\left( k^{\frac{d-1}{2}} \left( \frac{d}{2} \right)! \right)$ in runtime relative to globally optimal approaches (e.g., GOSDT).*

*Proof.* Any branch and bound algorithm for constructing a fully optimal tree will, in the worst case, involve searching through $(2k)^d$ sub-problems at depth $d$ (where we can ignore all sub-problems at shallower depths, because their cost is exponentially lower). By the same arguments as in Theorem 6.1, the runtime of brute force search without any caching is $\mathcal{O}(nk^d)$. Thus, the ratio of runtimes of brute force and Algorithm 2 is $\mathcal{O}\left( \frac{k^d}{\frac{(d-d_l)k^{d_l+1}}{d_l!} + \frac{k^{d-d_l}}{(d-d_l)!}} \right)$. From Theorem 6.1, we set

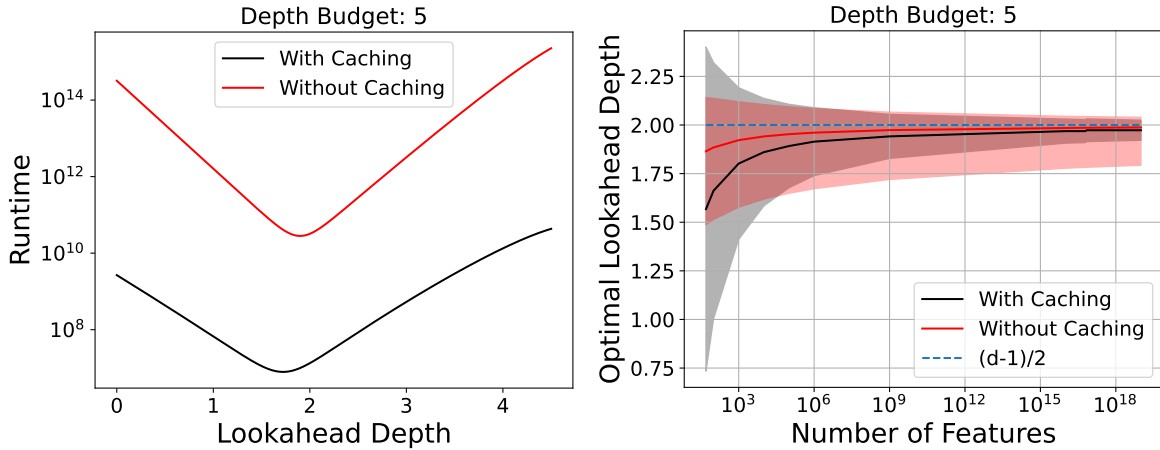

*Figure 19.* (left) The asymptotic runtime expression as a function of the lookahead depth for $k = 20$, $d = 5$, and $n = 1000$. This also lines up nicely with what we observe in practice, e.g. in Figure 9. (right) Exact value for the theoretically optimal lookahead depth as a function of the number of features (with their associated lower and upper bounds). T

$d_l = \frac{d-1}{2}$, as it minimizes the denominator of the above expression and hence gives the maximal runtime savings. Thus, the ratio of runtimes is:

$$\mathcal{O}\left( \frac{k^d}{\frac{(d-d_l)k^{d_l+1}}{d_l!} + \frac{k^{d-d_l}}{(d-d_l)!}} \right) \tag{96}$$

$$= \mathcal{O}\left( \frac{k^d \left(\frac{d+1}{2}\right)!}{\frac{d+1}{2}k^{\frac{d+1}{2}} + k^{\frac{d+1}{2}}} \right) \tag{97}$$

$$= \mathcal{O}\left( \frac{k^d \left(\frac{d+1}{2}\right)!}{\frac{d+3}{2}k^{\frac{d+1}{2}}} \right) \tag{98}$$

$$= \mathcal{O}\left( k^{\frac{d-1}{2}} \left(\frac{d}{2}\right)! \right). \tag{99}$$

$\square$

### A.8.4. Proof of Theorem 6.4

**Theorem 6.4** (Runtime Complexity of LicketySPLIT). *For a dataset $D$ with $k$ features and $n$ samples, and for depth constraint $d$, Algorithm 3 has runtime $O(nk^2d^2)$.*

*Proof.* **Sketch**: Running lookahead for a single step involves $k$ different potential splits, and a full run of a standard greedy algorithm for each sub-problem. Since a greedy algorithm's runtime for a sub-problem of size $n_s$ is $\mathcal{O}(n_s kd)$, and each split creates two sub-problems whose sub-problem sizes sum to $n$, we know that each split leads to $\mathcal{O}(nkd)$ runtime, and we have $k$ such splits to evaluate, leading to $\mathcal{O}(nk^2d)$ runtime for the first iteration.
In the recursive step, we call lookahead on two sub-problems whose sizes sum to $n$, and each of which has a similar runtime analysis.
We run at most $d$ layers of recursion.
From this, we have a total runtime bound of $\mathcal{O}(nk^2d^2)$, since we have $d$ levels which each take $\mathcal{O}(nk^2d)$ time.
**Proof via recurrence relation:**

For dataset $D$ and remaining depth $d$, and defining $i^*$ as the split selected by LicketySPLIT at the current iteration, we have the runtime recurrence relation:

$$T(D, d) = \begin{cases} T(D(i^*), d-1) + T(D(\bar{i}^*), d-1) + \sum_{i=1}^{k} \left( \mathcal{O}(|D|) + \mathcal{O}(|D(i)|kd) + \mathcal{O}(|D(\bar{i})|kd) \right) , & d > 1 \\ |D| & d = 1, \end{cases}$$

because at each level and each feature, LicketySPLIT needs to compute the split ($\mathcal{O}(|D|)$ time), then run greedy on the left and right subproblems, taking $\mathcal{O}(|D(i)|kd)$ and $\mathcal{O}(|D(\bar{i})|kd)$ time, respectively. Then it needs to recurse on the optimal of those splits.

Noting that $|D(\bar{i})| + |D(i)| = |D|$, this simplifies to:

$$T(D, d) = \begin{cases} T(D(i^*), d-1) + T(D(\bar{i}^*), d-1) + \sum_{i=1}^{k} \left( \mathcal{O}(|D|) + \mathcal{O}(|D|kd) \right) , & d > 1 \\ \mathcal{O}(|D|) & d = 1. \end{cases}$$

Given this recurrence, we can show $T(D, d) \in \mathcal{O}(nk^2d^2)$ inductively.

First, define $c_A, n_A$ be values such that the runtime of each $\mathcal{O}(|D|)$ steps in the recurrence above is below $c_A * |D|$ for $k > k_A, |D| > n_A$ (we know such values exist because of the definition of $\mathcal{O}$). Then define $c_B, n_B, d_B, k_B$ be values such that the runtime of each $\mathcal{O}(|D|kd)$ step in the recurrence above is below $c_B * |D|kd$ for $k > k_B, |D| > n_B, d > 1$ (we know such values exist because of the definition of $\mathcal{O}$). Now set:

$$c = \max(c_A, c_B, 1)$$
$$n_0 = \max(n_A, n_B, 1)$$
$$k_0 = \max(k_B, 1)$$
$$d_0 = 1$$

so that, for any $k \geq k_0, |D| = n \geq n_0, d \geq d_0$, we can bound all the $\mathcal{O}(|D|)$ steps as taking less than $c|D|$ time, and all the $\mathcal{O}(|D|kd)$ steps as taking less than $c|D|kd$ time.

$$T(D, d) \leq \begin{cases} T(D(i^*), d-1) + T(D(\bar{i}^*), d-1) + \sum_{i=1}^{k} \left( c|D| + c|D|kd \right) , & d > 1 \\ c|D| & d = 1. \end{cases}$$

We now want to show that for any $k \geq k_0, |D| = n \geq n_0, d \geq d_0$, we can bound the runtime of the recurrence $T$ as $\leq cnk^2d^2$, where $n = |D|$.

We show this by induction:

*Base Case ($d = 1$):*

Trivially, $T(D, 1) \leq c|D| \leq c|D|k^2d^2$ for any $k \geq k_0, |D| = n \geq n_0$. Note that $k^2d^2 \geq 1$ because each of $k$ and $d$ are at least 1.

*Inductive Step ($d \geq 2$):*

Now, inductively:

$$T(D, d) \leq T(D(i^*), d-1) + T(D(\bar{i}^*), d-1) + \sum_{i=1}^{k} \left( c|D| + c|D|kd \right) \tag{100}$$

$$T(D, d) = T(D(i^*), d-1) + T(D(\bar{i}^*), d-1) + c|D|k + c|D|k^2d \tag{101}$$

$$\leq ck^2(d-1)^2(|D(i*)| + c|D(\bar{i}*)|) + c|D|k + c|D|k^2d \tag{102}$$

$$\leq ck^2(d-1)^2(|D|) + c|D|k + c|D|k^2d \tag{103}$$

$$< c|D|k^2((d-1)^2 + 1 + d) \tag{104}$$

$$= c|D|k^2(d^2 - d + 2) \tag{105}$$

$$\leq c|D|k^2d^2, \text{ noting that } d \geq 2. \tag{106}$$

Thus as $|D| = n$, we have the runtime in $\mathcal{O}(nk^2d^2)$.

$\square$

A.8.5. ADDITIONAL CLAIMS, WITH PROOFS

We here prove some additional results about how our trees compare to optimal ones.

**Theorem A.2** (Optimality certificate based on lookahead depth). *Algorithm 2 will return a tree with objective no worse than a globally optimal tree with maximum depth $d_{lookahead}$.*

*Proof.* Note that Algorithm 2 considers all possible tree structures up to depth $d_{\text{lookahead}}$, with greedy completions of those structures. Those greedy completions are no worse than leaves with respect to our objective - they only expand beyond a leaf if the regularized objective is better than leaving the tree node as a leaf. So for any tree $t$ of depth at or below $d_{\text{lookahead}}$, there exists an analogous tree in the search space of Algorithm 2, with objective no worse than that of $t$.

Now, note that Algorithm 2 globally optimizes over its search space. So the tree returned by Algorithm 2 has objective no worse than any other element in the algorithm's search space.

We now have that the tree returned by Algorithm 2 has objective no worse than a globally optimal tree with maximum depth $d_{\text{lookahead}}$. For any globally optimal tree $t^*$ of that depth, we know there exists an analogous tree $t'$ in the search space of Algorithn 2, with objective no worse than that of $t^*$. And we know that the tree returned by the algorithm is no worse than tree $t'$, and thereby no worse than $t^*$.

(Note that postprocessing does not change the above, since it only ever improves the objective of the reported solution). □

**Theorem A.3** (Conditions for heuristic optimality). *If any true globally optimal tree uses greedy splits after depth $d_l$, then SPLIT will return a globally optimal tree.*

*Proof.* We prove Theorem A.3 as follows:

Our algorithm globally optimizes over the set of all trees that use greedy splits after depth $d_l$. Thus, if at least one such tree in that set is also in the set of globally optimal trees, we know we will find that tree or another equivalently good tree according to our objective. (Note that postprocessing does not change the above, since it only ever improves the objective of the reported solution). □

A.8.6. PROOF OF THEOREM 6.5

**Theorem 6.5** (SPLIT Can be Arbitrarily Better than Greedy). *For every $\epsilon > 0$ and depth budget $d$, there exists a data distribution $\mathcal{D}$ and sample size $n$ for which, with high probability over a random sample $S \sim \mathcal{D}^n$, Algorithm 2 with $d_l = \frac{d-1}{2}$ achieves accuracy at least $1 - \epsilon$ but a pure greedy approach achieves accuracy at most $\frac{1}{2} + \epsilon$.*

*Proof.* Our proof follows a similar construction as (Blanc et al., 2024). They define the function Tribes as follows:

**Definition A.4** (Tribes: from Blanc et al. 2024). For any input length $k$, let $w$ be the largest integer such that $(1 - 2^{-w})^{\ell/w} \leq \frac{1}{2}$. For $\mathbf{x} \in \{0,1\}^\ell$, let $\mathbf{x}^{(1)}$ be the first $w$ coordinates, $\mathbf{x}^{(2)}$ the second $w$, and so on. Tribes$_\ell$ is defined as:

$$\text{Tribes}_\ell(\mathbf{x}) = \left( \mathbf{x}_1^{(1)} \wedge .... \wedge \mathbf{x}_w^{(1)} \right) \vee .... \vee \left( \mathbf{x}_1^{(t)} \wedge .... \wedge \mathbf{x}_w^{(t)} \right) \tag{107}$$

where $t = \left\lfloor \frac{\ell}{w} \right\rfloor$. Blanc et al. (2019) prove the following properties of Tribes:

- Tribes$_\ell$ is monotone.

- Tribes$_\ell$ is nearly balanced:

$$\mathbb{E}_{\mathbf{x} \sim \{0,1\}^\ell}[\text{Tribes}_\ell(\mathbf{x})] = \frac{1}{2} \pm o(1)$$

  where the $o(1)$ term goes to 0 as $\ell$ goes to $\infty$.

- All variables in Tribes$_\ell$ have small correlation: For each $i \in [\ell]$,

$$\text{Cov}_{\mathbf{x} \sim \{0,1\}^\ell}[\mathbf{x}_i, \text{Tribes}_\ell(\mathbf{x})] = O\left( \frac{\log \ell}{\ell} \right).$$

Further define the majority function as follows:

**Definition A.5** (Majority). The majority function indicated by $\mathrm{Maj} : \{0,1\}^\ell \to \{0,1\}$, returns

$$\mathrm{Maj}(x) := \mathbf{1}[\text{at least half of } x\text{'s coordinates are } 1].$$

Let the number of features $k = d_l + u - 1$ for lookahead depth $d_l$ and constant $u$. Define the following data distribution over $\{0,1\}^k \times \{0,1\}$:

- Sample $\mathbf{x} \sim \mathrm{Uniform}(\{0,1\}^k)$.

- Let $\mathbf{x}(d_l)$ be the first $d_l$ elements in $\mathbf{x}$ and $x(\bar{d}_l)$ be the remaining elements. Compute:

$$y = f(\mathbf{x}) = \begin{cases} \mathrm{Tribes}_{d_l}(\mathbf{x}(d_l)) & \text{with probability } 1 - \epsilon, \\ \mathrm{Majority}(\mathbf{x}_{\bar{d}_l}) & \text{with probability } \epsilon. \end{cases} \tag{108}$$

**How does lookahead fare on this data distribution?**

Consider our lookahead heuristic. If we exhaustively search over all possible features up to depth $d_l$, we are guaranteed to perfectly classify $\mathrm{Tribes}_{d_l}(\mathbf{x}(d_l))$, as it is computed from $d_l$ features. In this scenario, the lookahead prefix tree will be a full binary tree, with $2^{d_l}$ leaves corresponding to every outcome of Tribes. When we extend this tree up to depth $d$ (with or without postprocessing), Algorithm 2 is still guaranteed to achieve at least $1 - \epsilon$ accuracy.

**How does greedy fare on this data distribution?**

We now apply Lemma 4.4 from (Blanc et al., 2024) in this context, adjusting the notation to suit our case. Let $T$ be the tree of depth $d$ returned by greedy. Consider any root-to-leaf path of $T$ that does not query any of the first $d_l$ features of $\mathbf{x}$ (i.e. the Tribes block). Only features from the Majority block are therefore queried by $T$ along this path. We can therefore write the probability of error along this path:

$$\Pr_{(\mathbf{x},y)\sim\mathcal{D}}[T(\mathbf{x}) = y \mid \mathbf{x} \text{ follows this path}]$$

$$= (1 - \epsilon) \Pr_{(\mathbf{x},y)\sim\mathcal{D}}[T(\mathbf{x}) = \mathrm{Tribes}_{d_l}(\mathbf{x}(d_l)) \mid \mathbf{x} \text{ follows this path}]$$

$$+ \epsilon \cdot \Pr_{(\mathbf{x},y)\sim\mathcal{D}}[T(\mathbf{x}) = \mathrm{Majority}(\mathbf{x}(\bar{d}_l)) \mid \mathbf{x} \text{ follows this path}]$$

$$\leq (1 - \epsilon) \cdot \left(\frac{1}{2} + o(1)\right) + \epsilon \cdot 1$$

$$\leq \frac{1 + \epsilon}{2} + o(1)$$

where the last line follows, because *Tribes* is nearly balanced. As the distribution over x is uniform, each leaf is equally likely. (Blanc et al., 2024) then show that, if only $p$-fraction of root-to-leaf paths of $T$ query at least one of the first $d_l$ coordinates, then:

$$\Pr_{(\mathbf{x},y)\sim\mathcal{D}}[T(\mathbf{x}) = y \leq (1 - p)\left(\frac{1 + \epsilon}{2} + o(1)\right) + p \cdot 1] \tag{109}$$

$$\leq \frac{1}{2} + \frac{\epsilon}{2} + \frac{p}{2} + o(1). \tag{110}$$

We now want to show that, just like in the case of (Blanc et al., 2024), $p$ is small asymptotically. If this is the case, we can claim that a greedy tree is arbitrarily bad. The only difference between (Blanc et al., 2024) and us is that their greedy tree has depth $d_l$ (adjusting for notation), but we want to construct a tree of depth $d$.

We now use Lemma 7.4 from (Blanc et al., 2019), which proves the following (again, adjusting for our notation): A random root-to-leaf path of a greedy tree $T$ satisfies the following with probability at least $1 - \mathcal{O}(u^{-2})$: *If the length of this path is less than $\mathcal{O}(\frac{u}{\log u})$, at any point along that path, all coordinates within the majority block that have not already been queried have correlation at least $\frac{1}{100\sqrt{u}}$.* Now, for a greedy tree of depth $d$:

- We need to set $u \geq \Omega(d \log d)$ so that all root-to-leaf paths have length at most $\mathcal{O}\left(\frac{u}{\log u}\right)$, so the above lemma applies.

- Remember that the size of our Tribes block is still fixed as the lookahead depth $d_l$, according to Equation 108. From the definition of tribes, all variables in this block will have correlation $\mathcal{O}\left(\frac{\log d_l}{d_l}\right)$. Because we want the correlations in the majority block to be greater than those in Tribes, we need to set $\frac{1}{100\sqrt{u}} \geq \Omega\left(\frac{\log d_l}{d_l}\right)$, implying that $u \leq \mathcal{O}\left(\frac{d_l^2}{\log^2 d_l}\right)$.

Thus, it follows that $p = \mathcal{O}(u^{-2})$ if the conditions above are satisfied. If we set $d_l = \frac{d-1}{2} = \mathcal{O}(d)$, we can say that, for any $\Omega(d \log d) \leq u \leq \mathcal{O}\left(\frac{d^2}{\log^2 d}\right)$, a greedy tree of depth $d$ will yield accuracy $\leq \frac{1}{2} + \epsilon$, as it almost never selects any variable from the Tribes block. $\qquad\square$

**Theorem A.6** (All Trees in RESPLIT Can be Arbitrarily Better Than Greedy). *For every $\epsilon, \epsilon' > 0$, depth budget $d$, and lookahead depth $d_l$, Rashomon set size $R$, there exists a data distribution $\mathcal{D}$ and sample size $n$ for which, with high probability over a random sample $S \sim \mathcal{D}^n$, all $R$ trees output by Algorithm 5 with minimum runtime lookahead depth $d_l = \frac{d-1}{2}$ achieve accuracy at least $1 - \epsilon - \epsilon' + \mathcal{O}(\epsilon\epsilon')$ but a pure greedy approach achieves accuracy at most $\frac{1}{2} + \epsilon$.*

*Proof.* We divide the proof as follows:

**Part 1: Defining the feature space**

Let the number of features $k = R + 2d$ for depth budget $d$ and a constant $R$ that is the size of the Rashomon set we want to generate. We now create a dataset of size $n$ with $k$ features in the following manner:

- Loop over $n$ iterations:
  - Sample $X_1 \ldots X_{2d}$ uniformly from $\{0, 1\}^{2d}$.
  - For each $2d < j \leq 2d + R$:
    * Choose a random index $idx(j) \sim \text{Uniform}\{1, d_l\}$
    * Define feature $X_j$ in the following manner:

$$X_j = \begin{cases} X_{idx(j)} & \text{With probability } 1 - \epsilon' \\ \bar{X}_{idx(j)} & \text{otherwise.} \end{cases} \tag{111}$$

Define the reference block of features to be $X_1, \ldots, X_d$. We break this block into 2 sub-blocks.

- Sub-block 1 corresponds to the $d_l$ features for which we will compute a parity bit. At a high level, a tree needs to know the parity of the expression in order to 'unlock' a high accuracy. This also serves to 'trick' greedy into not choosing these features, because they will have 0 correlation with the label. Let $X_1 \ldots X_{d_l}$ be the features in this sub-block.

- Sub-block 2 corresponds to the set of $d - d_l$ features over which we will take a majority vote. We will only reach this block when the parity bit is 1. Let $X_{d_l+1} \ldots X_d$ be the features in this sub-block.

**Part 2: Defining the labels**

For each example in this dataset, define the label $y$ as:

$$y = \begin{cases} (X_1 \oplus \ldots \oplus X_{d_l}) \wedge \text{Majority}(X_{d_l+1} \ldots X_d) & \text{with probability } 1 - \epsilon \\ \text{Majority}(X_{d+1} \ldots X_{2d}) & \text{with probability } \epsilon. \end{cases} \tag{112}$$

Intuitively, the label is the majority vote of the second block only when parity of the first block is even - otherwise the label is the minority vote.

**Part 3: Bounding the Error of the Rashomon Set**

We can immediately see that the best tree will achieve an error $\geq 1 - \epsilon$. The Rashomon set in this case will contain $R - 1$ trees (besides the empirical risk minimizer). In particular, each tree $T$ in the Rashomon set will split on one unique feature $X_j \; \forall 2d < j \leq 2d + R$, making a prediction on an instance $\mathbf{X} = (X_1, ... X_k)$ of the following form:

$$T(\mathbf{X}) = (X_1 \oplus \ldots X_j \ldots \oplus X_{d_l}) \wedge \text{Majority}(X_{d_l+1} \ldots X_d) \tag{113}$$

where tree $T$ employs feature $X_j$ in its path (defined in Equation 111. Whenever $X_j \neq X_{idx(j)}$ the parity of the first block will be different from that corresponding to Equation 112. However, this only happens with probability $\epsilon'$. For the $1 - \epsilon'$ proportion of cases, the error will be that of the best tree (i.e. at least $1 - \epsilon$), giving tree T an expected accuracy of least $(1 - \epsilon)(1 - \epsilon') = 1 - \epsilon - \epsilon' + \mathcal{O}(\epsilon\epsilon')$.

**Bounding the Performance of a Greedy Tree**

A greedy tree will seek to split on the feature that has the highest correlation with the label $y$. From the definition of $y$ in Equation 112, it follows that $X_{d+1}, \ldots, X_{2d}$ are the only variables that will have non 0 correlation with the label outcome. Thus, the tree will fully split only on these features up to depth $d$. However, this means that the tree does not learn the underlying parity function $(X_1 \oplus \ldots \oplus X_{d_l})$. Thus, $1 - \epsilon$ proportion of the time, the tree will achieve $\frac{1}{2}$ accuracy. Thus, the total accuracy is less than $\frac{1}{2}(1 - \epsilon) + \epsilon = \frac{1}{2} + \epsilon$. $\qquad \square$

**Theorem A.7** (LicketySPLIT Can be Arbitrarily Better than Greedy). *For every $\epsilon > 0$ and depth budget $d$, there exists a data distribution $\mathcal{D}$ and sample size $n$ for which, with high probability over a random sample $S \sim \mathcal{D}^n$, Algorithm 3 achieves accuracy at least $1 - \epsilon$ but a pure greedy approach achieves accuracy at most $\frac{1}{2} + \epsilon$.*

*Proof.* Let $x \sim \text{Uniform}(\{0, 1\}^{2d})$ and

$$y = \begin{cases} x_1 \oplus \text{Majority}(x_2, \ldots x_d) & \text{with probability } 1 - \epsilon \\ \text{Majority}(x_{d+1}, \ldots x_{2d}) & \text{with probability } \epsilon \end{cases}$$

A purely greedy, information-gain-based splitting approach will only split on features in the $x_{d+1}, \ldots x_{2d}$ block, since all have greater than zero information gain (unlike the other variables). Such a tree can improve to at most $\frac{1}{2} + \epsilon$ accuracy.

However, Algorithm 3 (LicketySPLIT), when deciding on the first split, will pick $x_1$ as the first split, after observing that being greedy from $x_1$ onwards will achieve accuracy at least $1 - \epsilon$: because once $x_1$ is known, variables $x_2, \ldots x_d$ have high information gain, and a greedy tree will pick those features for splits over $x_{d+1}, \ldots x_{2d}$. Splitting on all of the first $d$ features, then, affords performance at least $1 - \epsilon$.

$\qquad \square$

## A.9. Greedy Algorithm

---

**Algorithm 4** Greedy($D, d, \lambda$) $\rightarrow (t_{\text{greedy}}, lb)$

---

**Require:** $D, d, \lambda$ {Data subset, depth constraint, leaf regularization}
**Ensure:** $t_{\text{greedy}}, lb$ {tree grown with a greedy, CART-style method; and the objective of that tree}
 1: $t_{\text{greedy}} \leftarrow$ (Leaf predicting the majority label in $D$)
 2: $lb \leftarrow \lambda +$ (proportion of $D$ that does not have the majority label)
 3: **if** $d > 1$ **then**
 4:     let $f$ be the information gain maximizing split with respect to $D$
 5:     $t_{\text{left}}, lb_{\text{left}} \leftarrow$ Greedy($D(f), d - 1, \lambda$)
 6:     $t_{\text{right}}, lb_{\text{right}} \leftarrow$ Greedy($D(\bar{f}), d - 1, \lambda$)
 7:     **if** $lb_{\text{left}} + lb_{\text{right}} < lb$ **then**
 8:         $lb \leftarrow lb_{\text{left}} + lb_{\text{right}}$
 9:         $t_{\text{greedy}} \leftarrow$ tree corresponding to: if $f$ is True then $t_{\text{left}}$, else $t_{\text{right}}$
10:     **end if**
11: **end if**
12: **return** $t_{\text{greedy}}, lb$

---

## A.10. RESPLIT Algorithm

---

**Algorithm 5** RESPLIT($\ell$, D, $\lambda$, $d_l$, $d$)

---

**Require:** $\ell, D, \lambda, d_l, d$ {loss function, samples, regularizer, lookahead depth, depth budget}
 1: ModifiedTreeFARMS = TreeFARMS reconfigured to use **get_bounds** (Algorithm 1) whenever it encounters a new subproblem
 1: $tf =$ ModifiedTreeFARMS($\ell, D, \lambda, d_l$) {Call ModifiedTreeFARMS with depth budget $d_l$}
 2: **for** $t_{lookahead} \in tf$ **do** {Iterate through all depth $d_l$ prefixes found by ModifiedTreeFARMS}
 3:     **for** leaf $u \in t_{lookahead}$ **do**
 4:         $d_u =$ depth of leaf
 5:         $D(u) =$ subproblem associated with $u$
 6:         $\lambda_u = \lambda \frac{|D|}{|D(u)|}$ {Renormalize $\lambda$ for the subproblem in question}
 7:         $T_g, L_g =$ Greedy($D(u), d - d_u, \lambda_u$) {Objective of greedy tree trained on subproblem}
 8:         $t_u =$ TreeFARMS($D(u), d - d_u, \lambda_u, L_g$) {Find all subtrees with loss less than $L_g$}
 9:         **if** $t_u$ is not a leaf **then**
10:             Replace leaf $u$ with TreeFARMS object $t_u$
11:         **end if**
12:     **end for**
13:     $t_{lookahead} =$ Enumerate_TreeFARMS_subtrees {For each node in this prefix tree, store the number of subtrees we can generate rooted at that node. This speeds up indexing}
14: **end for**
15: **return** $tf$ {Return in-place edited ModifiedTreeFARMS object}

---

## A.11. Indexing Trees in RESPLIT

In this section, we present an algorithm that can quickly index trees output by RESPLIT. This would be especially useful if one wishes to obtain a random sample of trees from the Rashomon set. Because Algorithm 5 outputs a bespoke data structure involving TreeFARMS objects attached to a set of prefix trees, we needed to devise a method to efficiently query this structure to locate trees at a desired index.

- For each prefix found by the initial ModifiedTreeFARMS call, we additionally store the number of subtrees that can be formed with that prefix. Algorithm 6 shows how this is done.

- We also store the cumulative count of the total number of trees that can be formed by the prefixes seen so far as we iterate through the list of prefixes. Algorithm 7 called in line 2 of Algorithm 9 does this.

- Once the cumulative count is known, we start looping over the entire Rashomon set. For the $i^{th}$ index, we first obtain the corresponding prefix tree and then find the relative index of the $i^{th}$ tree within this prefix tree structure. For example, if we query the $500^{th}$ tree and our prefix contains trees indexed $400 - 600$ in the Rashomon set, the relative index of the query tree within this prefix is $100$.

- Using Algorithm 8, we proceed to recursively locate the relevant subtrees beyond the prefix. In particular, at a given node in the prefix, we have access to the number of sub-trees that can be formed with its left and right children. We use this information to create two separate indexes for the left and right child (seen in lines $9 - 10$)

- We hash all the indexes for future retrieval (line 16 in Algorithm 9).

---

**Algorithm 6** Enumerate_TreeFARMS_subtrees

---

**Require:** $t_{lookahead}$ {Lookahed prefix with TreeFARMS objects attached to leaves}
1: **if** $t_{lookahead}$ is None **then**
2:     **Return** 1
3: **else if** $t_{lookahead}$ is a TreeFARMS object **then**
4:     **Return** $len(t_{lookahead})$, $t_{lookahead}$
5: **end if**
6: left_expansions, left_subtree = enumerate_treefarms_subtrees($t_{lookahead}$.left_child)
7: $t_{lookahead}$.left_child.node = left_subtree
8: $t_{lookahead}$.left_child.subtree_count = left_expansions
9: right_expansions, right_subtree = enumerate_treefarms_subtrees($t_{lookahead}$.right_child)
10: $t_{lookahead}$.right_child.node = right_subtree
11: $t_{lookahead}$.right_child.subtree_count = right_expansions
12: **Return** left_expansions $\times$ right_expansions, $t_{lookahead}$ {Total number of subtrees = cross product of left and right subtree count}

---

**Algorithm 7** RESPLIT_Rset_Count(RESPLIT_obj)

---

**Require:** RESPLIT_obj {The RESPLIT object output by Algorithm 5}
1: $t_{count} = 0$ {Total # trees}
2: $p_{counts} = []$ {Cumulative count of # trees beginning with a given prefix}
3: **for** $t_{lookahead} \in$ RESPLIT_obj **do**
4:     $p_{count} = 1$
5:     **for** leaf $u \in t_{lookahead}$ **do**
6:         $tf_u$ = TreeFARMS object fitted on subproblem $D(u)$
7:         $s_{count} = len(tf_u)$ {Number of subtrees found for subproblem $D(u)$}
8:         $p_{count} = p_{count} \times s_{count}$
9:     **end for**
10:    $t_{count} = t_{count} + p_{count}$
11:    $p_{counts}.add(t_{count})$
12: **end for**
13: **return** $p_{counts}, t_{count}$

---

---

**Algorithm 8** get_leaf_subtree_at_idx($t_{lookahead}$, tree_idx)

---

**Require:** $t_{lookahead}$, tree_idx {A lookahead prefix tree with TreeFARMS objects attached to leaves, index to search within this tree}
 1: **if** $t_{lookahead}$ is a Leaf **then**
 2:     **return** $t_{lookahead}$ {Directly return the leaf object}
 3: **else if** $t_{lookahead}$ is a list **then**
 4:     **return** $t_{lookahead}[tree\_idx]$ {If it's a list, return the subtree at the given index}
 5: **end if**
 6: $tree \leftarrow$ Node($t_{lookahead}$.feature)) {Initialize an empty node}
 7: left_count = $t_{lookahead}$.left_child.subtree_count {The number of subtrees that can be found rooted at this node}
 8: right_count = $t_{lookahead}$.right_child.subtree_count
 9: right_idx = tree_idx % right_count
10: left_idx = tree_idx // right_count
11: $tree$.left_child = get_leaf_subtree_at_idx($t_{lookahead}$.left_child.node, left_idx)
12: $tree$.right_child = get_leaf_subtree_at_idx($t_{lookahead}$.right_child.node, right_idx)
13: **return** $tree$

---

**Algorithm 9** RESPLIT_indexing

---

**Require:** RESPLIT_obj {The RESPLIT object output by Algorithm 5}
 1: hash = $\emptyset$ {Dictionary to map global tree indices to tree objects}
 2: $t_{count}, p_{counts}$ = RESPLIT_Rset_Count(RESPLIT_obj) {Total number of trees and prefix-wise cumulative counts}
 3: start = 0
 4: **for** $i = 0$ **to** $len(p_{counts}) - 1$ **do**
 5:     **if** $i > 0$ **then**
 6:         start = p_counts[$i - 1$]+1 {Start index for prefix $i$}
 7:     **end if**
 8:     end = p_counts[$i$] {End index for prefix $i$}
 9:     $t_{lookahead}$ = RESPLIT_obj.prefix_list[$i$] {The $i$-th prefix tree}
10:     **for** local_idx = 0 **to** end $-$ start $-1$ **do**
11:         global_idx = start + local_idx {Absolute index of tree in Rashomon set}
12:         tree = get_leaf_subtree_at_idx($t_{lookahead}$, $local\_idx$) {Retrieve the corresponding subtree}
13:         hash[global_idx] = tree
14:     **end for**
15: **end for**
16: **return** hash

---

### A.12. Modifications to Existing GOSDT / TreeFARMS Code

In this section, we detail the main modifications we made to the existing GOSDT and TreeFARMS codebase in order to set up SPLIT, LicketySPLIT, and RESPLIT. The algorithm components in red are the modifications - note that GOSDT and TreeFARMS both call these functions. TreeFARMS does some additional post-processing of the search trie to find the set of near-optimal trees - the details can be seen in (Xin et al., 2022).

---

**Algorithm 10** find_lookahead_tree($\ell$, $D$, $\lambda$, $d_l$, $d$)

---

**Require:** $\ell$, $D$, $\lambda$, $d_l$, $d$ {loss function, dataset, regularizer, lookahead depth, global depth budget}

1: $Q \leftarrow \emptyset$ {priority queue}
2: $G \leftarrow \emptyset$ {dependency graph}
3: $s_0 \leftarrow \{1, \ldots, 1\}$ {bit-vector of 1's of length $n$}
4: $p_0 \leftarrow$ FIND_OR_CREATE_NODE($G, s_0, d_l, d, 0$) {root (with depth 0)}
5: $Q$.push($(s_0, 0)$)
6: $N = |D|$ {global dataset size}
7: **while** $p_0$.lb $\neq p_0$.ub **do**
8:     $s, d' \leftarrow Q$.pop() {index of problem to work on}
9:     $p \leftarrow G$.find($s$) {find problem to work on}
10:     **if** $p$.lb $= p$.ub **then**
11:         **continue** {problem already solved}
12:     **end if**
13:     $(lb', ub') \leftarrow (\infty, \infty)$ {loose starting bounds}
14:     **for** each feature $j \in [1, k]$ **do**
15:         $(s_l, s_r) \leftarrow$ split($s, j, D$) {create children}
16:         $p_l^j \leftarrow$ FIND_OR_CREATE_NODE($G, s_l, d_l, d, d' + 1, N$)
17:         $p_r^j \leftarrow$ FIND_OR_CREATE_NODE($G, s_r, d_l, d, d' + 1, N$)
18:         $lb' \leftarrow \min(lb', p_l^j.\text{lb} + p_r^j.\text{lb})$ {create bounds as if $j$ were chosen for splitting}
19:         $ub' \leftarrow \min(ub', p_l^j.\text{ub} + p_r^j.\text{ub})$
20:     **end for**
21:     **if** $p$.lb $\neq lb'$ or $p$.ub $\neq ub'$ **then** {signal the parents if an update occurred}
22:         $p$.ub $\leftarrow \min(p.\text{ub}, ub')$
23:         $p$.lb $\leftarrow \min(p.\text{ub}, \max(p.\text{lb}, lb'))$
24:         **for** $p_\pi \in G$.parent($p$) **do** {propagate information upwards}
25:             $Q$.push($(p_\pi.\text{id}, d' - 1), \text{priority} = 1$)
26:         **end for**
27:     **end if**
28:     **if** $p$.lb $\geq p$.ub **then**
29:         **continue** {problem solved just now}
30:     **end if**
31:     **if** $d' < d_l$ **then**
32:         **for** each feature $j \in [1, M]$ **do** {loop, enqueue all children}
33:             **repeat** line 14-16 {fetch $p_l^j$ and $p_r^j$ in case of update}
34:             $lb' \leftarrow p_l^j.\text{lb} + p_r^j.\text{lb}$
35:             $ub' \leftarrow p_l^j.\text{ub} + p_r^j.\text{ub}$
36:             **if** $lb' < ub'$ and $lb' \leq p$.ub **then**
37:                 $Q$.push($(s_l, d + 1), \text{priority} = 0$)
38:                 $Q$.push($(s_r, d + 1), \text{priority} = 0$)
39:             **end if**
40:         **end for**
41:     **end if**
42: **end while**
43: **return** $\mathcal{G}$

---

---

**Algorithm 11** FIND_OR_CREATE_NODE($G$, $s$, $d_l$, $d$, $d'$, N)

---

**Require:** $G, s, d_l, d, d', N$ {Graph, subproblem, lookahead depth, overall depth budget, current depth, global dataset size}
 1: **return** representation of subproblem entry for $s$, with that suboropblem being present in the graph $G$
 2: **if** $G.\text{find}(s) = \text{NULL}$ **then** {$p$ not yet in graph}
 3:     create node $p$
 4:     $p.\text{id} \leftarrow s$ {identify $p$ by $s$}
 5:     $D(s) = $ Dataset associated with subproblem $s$
 6:     $p.\text{ub}, p.\text{lb} \leftarrow \text{get\_bounds}(D(s), d_l, d', d, N)$
 7:     **if** $p.ub \leq p.lb + \lambda$ **then** {If a further split would lead to worse objective than the upper bound}
 8:         $p.\text{lb} \leftarrow p.\text{ub}$ {no more splitting needed}
 9:     **end if**
10:     $G.\text{insert}(p)$ {put $p$ in dependency graph}
11: **end if**
12: **return** $G.\text{find}(s)$

---

**Algorithm 12** get_bounds($D$, $d_l$, $d'$, $d$, $N$) $\rightarrow$ lb, ub

---

**Require:** $D, d_l, d', d, N$ {support, lookahead depth, current depth, overall depth budget, global dataset size}
 1: **return** lb, ub {Return Lower and Upper Bounds}
 2: **if** $d' = d_l$ **then**
 3:     $T_g = \text{Greedy}(D, d - d_l, \lambda)$
 4:     $H(T_g) = \#$ Leaves in $T_g$
 5:     $\alpha \leftarrow \lambda H(T_g) + \frac{1}{N} \sum_{i \in s} \mathbf{1}[y_i \neq T_g(x_i)]$
 6:     $lb \leftarrow \alpha$
 7:     $ub \leftarrow \alpha$
 8:     $lb \leftarrow$ Equivalent points bound (Lin et al., 2020)
 9:     $ub = \lambda + \min\left(\frac{1}{N} \sum_{(x,y) \in D} \mathbb{1}[y_i = 1], \frac{1}{N} \sum_{(x,y) \in D} \mathbb{1}[y_i = 0]\right)$
10: **end if**
11: **return** lb, ub

---

**Algorithm 13** extract_tree($D$, $\mathcal{G}$, $d_l$)

---

**Require:** $D, \mathcal{G}, d_l$ {Dataset, Dependency graph of search space, lookahead depth}
 1: **return** Tree $t$
 2: $t \leftarrow$ (Leaf predicting the majority label in $D$)
 3: $ub \leftarrow \lambda +$ (proportion of $D$ that has the minority label)
 4: **if** $d_l > 1$ **then**
 5:     **for** feature $f \in \mathcal{F}$ **do**
 6:         $p_f = $ subproblem associated with $D(f)$
 7:         $p_{\bar{f}} = $ subproblem associated with $D(\bar{f})$
 8:         **if** $p_f.ub + p_{\bar{f}}.ub \leq ub$ **then**
 9:             $f_{opt} = f$ {Best Feature}
10:             $ub = p_f.ub + p_{\bar{f}}.ub$
11:         **end if**
12:     **end for**
13:     $t_{left} = \text{extract\_tree}(D(f_{opt}), \mathcal{G}(f_{opt}), d_l - 1)$
14:     $t_{right} = \text{extract\_tree}(D(\bar{f}_{opt}), \mathcal{G}(\bar{f}_{opt}), d_l - 1)$
15:     $t.left = t_{left}$
16:     $t.right = t_{right}$
17: **end if**
18: **return** t

---

---

**Algorithm 14** ModifiedGOSDT($\ell$, D, $\lambda$, $d_l$, $d$)

---

**Require:** $\ell$, $D$, $\lambda$, $d_l$, $d$ {loss function, samples, regularizer, lookahead depth, depth budget}
 1: $\mathcal{G}$ = find_lookahead_tree($\ell$, D, $\lambda$, $d_l$, $d$)
 2: t = extract_tree($D$, $\mathcal{G}$, $d_l$) {Extracts the prefix of the found tree, without filling in the greedy splits}
 3: **return** t

---

### A.13. Additional Experimental Results

For thoroughness, we provide several additional experimental results in Figures 20, 21, and 22 and Tables 7, 8, 9, 10, 11, 12, 13, 14, 15, 16, and 17. These experimental results provide other perspectives on the loss/runtime/sparsity tradeoff, with particular emphasis on comparisons with a greedy approach.

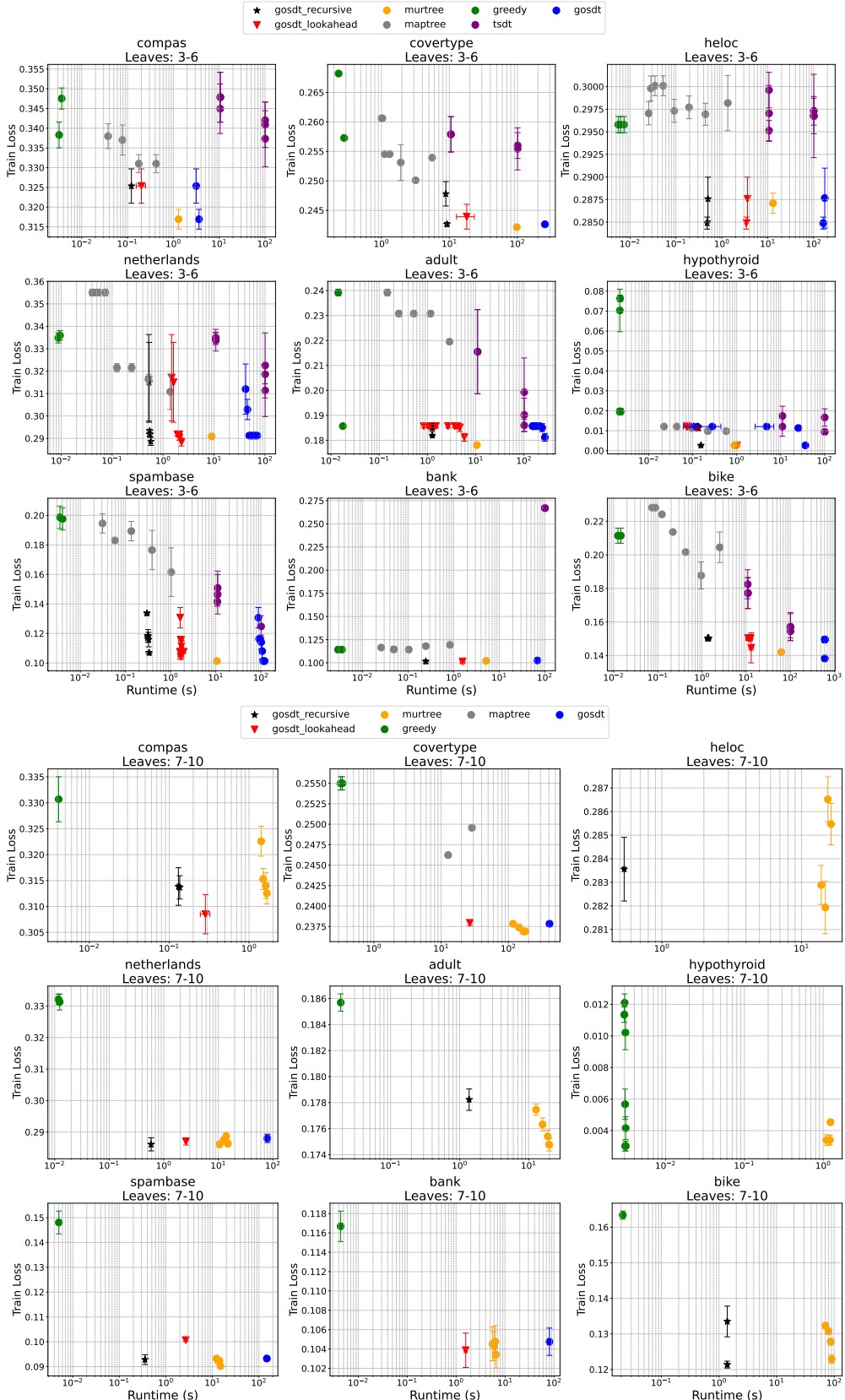

*Figure 20.* Tradeoff between **train loss and runtime** for all algorithms tested, for different sparsity levels (measured by # of leaves in the tree). Depth Budget = 5.

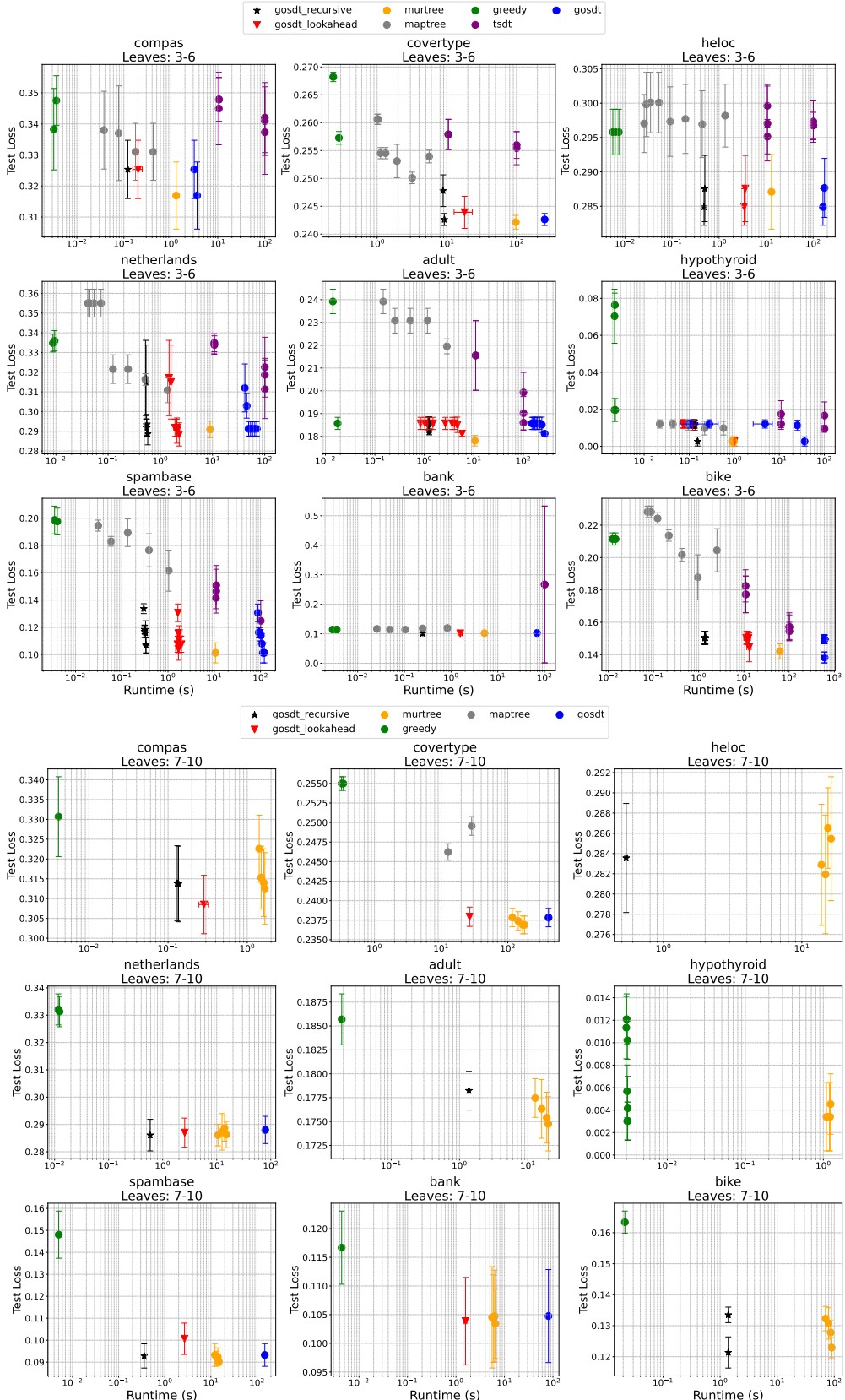

*Figure 21.* Tradeoff between **test loss and runtime** for all algorithms tested, for different sparsity levels (measured by # of leaves in the tree). Depth Budget = 5.

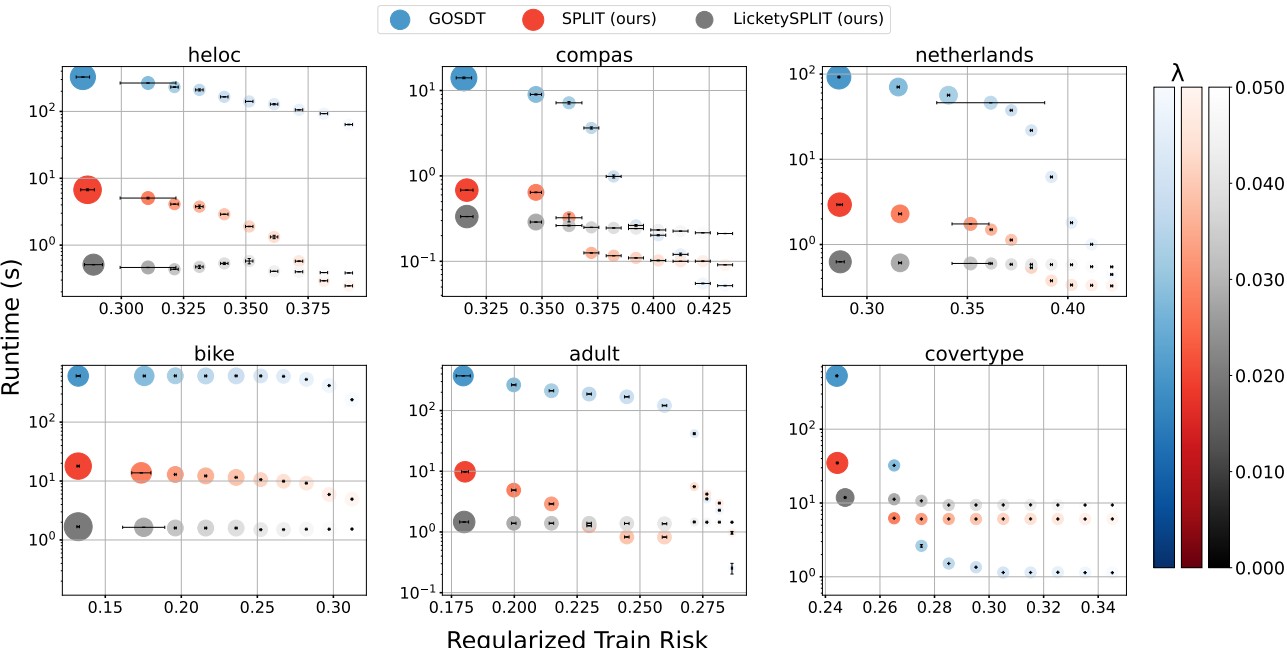

*Figure 22.* Regularized training objective vs. training time (in seconds) for GOSDT vs. our algorithms. The size of the points indicates the number of leaves in the resulting tree. Both SPLIT and LicketySPLIT are much faster for most values of sparsity penalty $\lambda$, with the only potential slowdown being in the sub-second regime due to overhead costs. Depth Budget = 5.

*Table 7.* Results for $\lambda = 0.001$

| Dataset | Algorithm | Train Objective | Runtime (s) |
|---|---|---|---|
| bike | GOSDT (SOTA) | $0.1322 \pm 0.0015$ | $606.87 \pm 2.12$ |
| | **LicketySPLIT (ours)** | $\mathbf{0.1328 \pm 0.0010}$ | $\mathbf{1.68 \pm 0.05}$ |
| | Greedy | $0.2101 \pm 0.0401$ | $0.01 \pm 0.00$ |
| adult | GOSDT (SOTA) | $0.1797 \pm 0.0032$ | $372.62 \pm 2.01$ |
| | **LicketySPLIT (ours)** | $\mathbf{0.1800 \pm 0.0020}$ | $\mathbf{1.46 \pm 0.01}$ |
| | Greedy | $0.1950 \pm 0.0350$ | $0.01 \pm 0.00$ |
| covertype | GOSDT (SOTA) | $0.2442 \pm 0.0002$ | $528.67 \pm 1.68$ |
| | **LicketySPLIT (ours)** | $\mathbf{0.2472 \pm 0.0003}$ | $\mathbf{11.86 \pm 0.29}$ |
| | Greedy | $0.2681 \pm 0.0020$ | $0.01 \pm 0.00$ |

*Table 8.* Results for $\lambda = 0.006$

| Dataset | Algorithm | Train Objective | Runtime (s) |
|---|---|---|---|
| heloc | GOSDT | $0.3107 \pm 0.0128$ | $266.15 \pm 1.81$ |
| | SPLIT (ours) | $0.3107 \pm 0.0128$ | $5.06 \pm 0.32$ |
| | LicketySPLIT (ours) | $0.3107 \pm 0.0128$ | $0.46 \pm 0.01$ |
| compas | GOSDT | $0.3473 \pm 0.0029$ | $8.99 \pm 0.31$ |
| | SPLIT (ours) | $0.3473 \pm 0.0029$ | $0.64 \pm 0.01$ |
| | LicketySPLIT (ours) | $0.3473 \pm 0.0029$ | $0.29 \pm 0.00$ |
| netherlands | GOSDT | $0.3156 \pm 0.0006$ | $70.49 \pm 0.15$ |
| | SPLIT (ours) | $0.3165 \pm 0.0008$ | $2.29 \pm 0.01$ |
| | LicketySPLIT (ours) | $0.3165 \pm 0.0008$ | $0.61 \pm 0.00$ |
| bike | GOSDT | $0.1736 \pm 0.0007$ | $607.32 \pm 0.89$ |
| | SPLIT (ours) | $0.1737 \pm 0.0072$ | $13.77 \pm 0.18$ |
| | LicketySPLIT (ours) | $0.1753 \pm 0.0158$ | $1.64 \pm 0.01$ |
| adult | GOSDT | $0.1998 \pm 0.0010$ | $264.79 \pm 0.99$ |
| | SPLIT (ours) | $0.1998 \pm 0.0010$ | $4.89 \pm 0.02$ |
| | LicketySPLIT (ours) | $0.1998 \pm 0.0010$ | $1.39 \pm 0.00$ |
| covertype | GOSDT | $0.2652 \pm 0.0001$ | $32.27 \pm 0.27$ |
| | SPLIT (ours) | $0.2652 \pm 0.0001$ | $6.21 \pm 0.02$ |
| | LicketySPLIT (ours) | $0.2652 \pm 0.0001$ | $11.24 \pm 0.01$ |

*Table 9.* Results for $\lambda = 0.011$

| Dataset | Algorithm | Train Objective | Runtime (s) |
|---|---|---|---|
| heloc | GOSDT | $0.3214 \pm 0.0017$ | $231.41 \pm 5.55$ |
| | SPLIT (ours) | $0.3214 \pm 0.0017$ | $4.10 \pm 0.03$ |
| | LicketySPLIT (ours) | $0.3214 \pm 0.0017$ | $0.44 \pm 0.00$ |
| compas | GOSDT | $0.3621 \pm 0.0066$ | $7.17 \pm 0.53$ |
| | SPLIT (ours) | $0.3621 \pm 0.0066$ | $0.32 \pm 0.08$ |
| | LicketySPLIT (ours) | $0.3621 \pm 0.0066$ | $0.26 \pm 0.01$ |
| netherlands | GOSDT | $0.3406 \pm 0.0006$ | $56.54 \pm 0.11$ |
| | SPLIT (ours) | $0.3515 \pm 0.0105$ | $1.74 \pm 0.01$ |
| | LicketySPLIT (ours) | $0.3515 \pm 0.0105$ | $0.60 \pm 0.00$ |
| bike | GOSDT | $0.1961 \pm 0.0006$ | $610.20 \pm 0.55$ |
| | SPLIT (ours) | $0.1961 \pm 0.0006$ | $12.91 \pm 0.05$ |
| | LicketySPLIT (ours) | $0.1961 \pm 0.0006$ | $1.60 \pm 0.01$ |
| adult | GOSDT | $0.2148 \pm 0.0010$ | $211.86 \pm 0.68$ |
| | SPLIT (ours) | $0.2148 \pm 0.0010$ | $2.90 \pm 0.07$ |
| | LicketySPLIT (ours) | $0.2148 \pm 0.0010$ | $1.39 \pm 0.00$ |
| covertype | GOSDT | $0.2752 \pm 0.0001$ | $2.62 \pm 0.23$ |
| | SPLIT (ours) | $0.2752 \pm 0.0001$ | $6.07 \pm 0.08$ |
| | LicketySPLIT (ours) | $0.2752 \pm 0.0001$ | $10.70 \pm 0.00$ |

| Dataset | Binarization Time (s) | Algorithm | Runtimes (s) | Test Loss |
|---------|----------------------|-----------|--------------|-----------|
| compas | [2.69, 2.91] | LicketySPLIT (ours) | [2.85, 2.97] | **[0.306, 0.328]** |
| | | SPLIT (ours) | [2.84, 3.01] | [0.314, 0.335] |
| | | CART | [0.00, 0.00] | [0.322, 0.354] |
| bank | [0.37, 0.41] | LicketySPLIT (ours) | [0.47, 0.52] | [0.103, 0.119] |
| | | SPLIT (ours) | [0.57, 0.67] | **[0.101, 0.116]** |
| | | CART | [0.00, 0.00] | [0.106, 0.123] |
| bike | [2.20, 2.32] | LicketySPLIT (ours) | [2.67, 2.81] | **[0.133, 0.147]** |
| | | SPLIT (ours) | [3.80, 4.71] | [0.139, 0.152] |
| | | CART | [0.01, 0.01] | [0.207, 0.215] |
| adult | [2.26, 2.74] | LicketySPLIT (ours) | [2.94, 3.19] | **[0.155, 0.159]** |
| | | SPLIT (ours) | [3.66, 4.26] | **[0.155, 0.159]** |
| | | CART | [0.02, 0.02] | [0.183, 0.191] |
| hypothyroid | [0.99, 1.32] | LicketySPLIT (ours) | [1.13, 1.30] | **[0.004, 0.005]** |
| | | SPLIT (ours) | [1.18, 1.36] | **[0.004, 0.005]** |
| | | CART | [0.00, 0.00] | [0.009, 0.014] |
| covertype | [19.49, 20.00] | LicketySPLIT (ours) | [25.50, 25.75] | [0.242, 0.244] |
| | | SPLIT (ours) | [25.62, 25.91] | **[0.242, 0.244]** |
| | | CART | [0.68, 0.69] | [0.266, 0.269] |
| netherlands | [1.92, 2.04] | LicketySPLIT (ours) | [2.31, 2.39] | [0.285, 0.294] |
| | | SPLIT (ours) | [2.74, 3.06] | **[0.284, 0.294]** |
| | | CART | [0.00, 0.00] | [0.314, 0.338] |
| heloc | [0.89, 1.01] | LicketySPLIT (ours) | [1.20, 1.27] | **[0.284, 0.289]** |
| | | SPLIT (ours) | [2.08, 2.53] | **[0.284, 0.289]** |
| | | CART | [0.01, 0.01] | [0.293, 0.299] |
| spambase | [0.70, 0.73] | LicketySPLIT (ours) | [0.88, 0.90] | **[0.094, 0.114]** |
| | | SPLIT (ours) | [1.16, 1.19] | [0.097, 0.111] |
| | | CART | [0.01, 0.01] | [0.164, 0.207] |

*Table 10.* Results (# leaves between 3–6). The 95% confidence interval is shown. Binarization is only applicable to LicketySPLIT/SPLIT. The runtimes for SPLIT / LicketySPLIT **include** binarization time.

| Dataset | Binarization Time (s) | Algorithm | Runtimes (s) | Test Loss |
|---|---|---|---|---|
| compas | [2.69, 2.91] | LicketySPLIT (ours) | [2.86, 2.98] | [0.303, 0.321] |
| | | SPLIT (ours) | [2.91, 3.06] | **[0.302, 0.320]** |
| | | CART | [0.00, 0.00] | [0.325, 0.338] |
| bank | [0.37, 0.41] | LicketySPLIT (ours) | [0.48, 0.53] | [0.103, 0.118] |
| | | SPLIT (ours) | [0.59, 0.68] | **[0.101, 0.117]** |
| | | CART | [0.00, 0.00] | [0.106, 0.123] |
| bike | [2.20, 2.32] | LicketySPLIT (ours) | [2.70, 2.84] | **[0.123, 0.125]** |
| | | SPLIT (ours) | [4.06, 4.78] | [0.127, 0.134] |
| | | CART | [0.01, 0.01] | [0.166, 0.238] |
| adult | [2.26, 2.74] | LicketySPLIT (ours) | [2.97, 3.22] | **[0.149, 0.154]** |
| | | SPLIT (ours) | [3.95, 4.54] | [0.149, 0.155] |
| | | CART | [0.03, 0.04] | [0.165, 0.180] |
| hypothyroid | [0.99, 1.32] | LicketySPLIT (ours) | [1.13, 1.30] | [0.003, 0.005] |
| | | SPLIT (ours) | [1.20, 1.38] | [0.003, 0.005] |
| | | CART | [0.00, 0.00] | **[0.002, 0.004]** |
| covertype | [19.49, 20.00] | LicketySPLIT (ours) | [25.50, 25.74] | [0.240, 0.243] |
| | | SPLIT (ours) | [26.57, 26.92] | **[0.239, 0.242]** |
| | | CART | [0.99, 1.00] | [0.254, 0.256] |
| netherlands | [1.92, 2.04] | LicketySPLIT (ours) | [2.31, 2.41] | **[0.282, 0.293]** |
| | | SPLIT (ours) | [2.79, 3.12] | **[0.282, 0.293]** |
| | | CART | [0.00, 0.00] | [0.297, 0.314] |
| heloc | [0.89, 1.01] | LicketySPLIT (ours) | [1.21, 1.27] | [0.284, 0.293] |
| | | SPLIT (ours) | [2.18, 2.42] | **[0.282, 0.293]** |
| | | CART | [0.00, 0.00] | [0.291, 0.327] |
| spambase | [0.70, 0.73] | LicketySPLIT (ours) | [0.89, 0.91] | **[0.085, 0.096]** |
| | | SPLIT (ours) | [1.36, 1.52] | [0.085, 0.098] |
| | | CART | [0.02, 0.02] | [0.114, 0.141] |

*Table 11.* Results (# leaves between 7–10). The 95% confidence interval is shown. Binarization is only applicable to LicketySPLIT/SPLIT. The runtimes for SPLIT / LicketySPLIT **include** binarization time.

| Dataset | Binarization Time (s) | Algorithm | Runtimes (s) | Test Loss |
|---------|------------------------|-----------|--------------|-----------|
| compas | [2.69, 2.91] | SPLIT (ours) | [2.92, 3.08] | **[0.302, 0.316]** |
| | | CART | [0.00, 0.00] | [0.318, 0.333] |
| bank | [0.37, 0.41] | LicketySPLIT (ours) | [0.49, 0.53] | [0.099, 0.116] |
| | | SPLIT (ours) | [0.59, 0.69] | **[0.100, 0.118]** |
| | | CART | [0.00, 0.00] | [0.104, 0.119] |
| bike | [2.20, 2.32] | LicketySPLIT (ours) | [2.70, 2.83] | **[0.114, 0.123]** |
| | | SPLIT (ours) | [4.40, 5.14] | [0.121, 0.129] |
| | | CART | [0.02, 0.02] | [0.130, 0.139] |
| adult | [2.26, 2.74] | LicketySPLIT (ours) | [2.97, 3.22] | [0.148, 0.155] |
| | | SPLIT (ours) | [4.09, 4.79] | **[0.148, 0.154]** |
| | | CART | [0.04, 0.04] | [0.154, 0.161] |
| netherlands | [1.92, 2.04] | LicketySPLIT (ours) | [2.32, 2.42] | [0.283, 0.291] |
| | | SPLIT (ours) | [2.89, 3.22] | **[0.282, 0.291]** |
| | | CART | [0.00, 0.00] | [0.293, 0.309] |
| heloc | [0.89, 1.01] | LicketySPLIT (ours) | [1.23, 1.29] | **[0.281, 0.292]** |
| | | SPLIT (ours) | [2.41, 2.73] | [0.286, 0.297] |
| | | CART | [0.02, 0.02] | [0.298, 0.306] |
| spambase | [0.70, 0.73] | LicketySPLIT (ours) | [0.89, 0.92] | [0.086, 0.094] |
| | | SPLIT (ours) | [1.50, 1.65] | **[0.081, 0.093]** |
| | | CART | [0.02, 0.02] | [0.114, 0.136] |

*Table 12.* Results (# leaves between 11–14). The 95% confidence interval is shown. Binarization is only applicable to LicketyS-PLIT/SPLIT. The runtimes for SPLIT / LicketySPLIT **include** binarization time.

| Dataset | Binarization Time (s) | Algorithm | # Leaves (95% CI) | Runtimes (s) | Test Loss |
|---|---|---|---|---|---|
| compas | [2.69, 2.91] | LicketySPLIT (ours) | [14.2, 17.2] | [2.99, 3.23] | [0.305, 0.325] |
| | | SPLIT (ours) | [13.6, 16.0] | [2.76, 3.32] | **[0.304, 0.314]** |
| | | CART | [28.6, 30.8] | [0.00, 0.00] | [0.307, 0.324] |
| bank | [0.37, 0.41] | LicketySPLIT (ours) | [20.2, 23.8] | [0.52, 0.57] | [0.102, 0.116] |
| | | SPLIT (ours) | [15.2, 16.0] | [0.63, 0.75] | [0.102, 0.118] |
| | | CART | [16.6, 17.8] | [0.01, 0.01] | **[0.098, 0.112]** |
| bike | [2.20, 2.32] | LicketySPLIT (ours) | [17.8, 19.6] | [2.72, 2.94] | **[0.114, 0.122]** |
| | | CART | [27.6, 28.2] | [0.02, 0.03] | [0.122, 0.130] |
| adult | [2.26, 2.74] | LicketySPLIT (ours) | [19.8, 21.4] | [2.90, 3.36] | **[0.146, 0.151]** |
| | | SPLIT (ours) | [15.2, 16.0] | [4.29, 5.45] | [0.148, 0.153] |
| | | CART | [22.6, 23.2] | [0.02, 0.03] | [0.152, 0.157] |
| netherlands | [1.92, 2.04] | LicketySPLIT (ours) | [15.4, 18.4] | [2.33, 2.72] | [0.282, 0.291] |
| | | SPLIT (ours) | [13.4, 15.2] | [2.97, 3.45] | **[0.284, 0.291]** |
| | | CART | [18.6, 20.2] | [0.01, 0.01] | [0.293, 0.306] |
| heloc | [0.89, 1.01] | LicketySPLIT (ours) | [18.0, 23.0] | [1.25, 1.44] | **[0.285, 0.295]** |
| | | SPLIT (ours) | [14.4, 16.4] | [2.31, 2.63] | [0.286, 0.301] |
| | | CART | [21.4, 23.2] | [0.02, 0.02] | [0.290, 0.299] |
| spambase | [0.70, 0.73] | LicketySPLIT (ours) | [24.4, 25.6] | [0.91, 0.96] | **[0.081, 0.088]** |
| | | SPLIT (ours) | [14.0, 15.8] | [1.46, 1.57] | [0.081, 0.093] |
| | | CART | [20.0, 23.2] | [0.02, 0.02] | [0.082, 0.103] |

*Table 13.* Comparing CART and SPLIT/LicketySPLIT for non sparse trees. The 95% confidence interval is shown. Binarization is only applicable to LicketySPLIT/SPLIT. The runtimes for SPLIT / LicketySPLIT **include** binarization time. We report the best tree with between $15 - 30$ leaves found during hyperparameter search.

| Dataset | Leaves | Runtimes | Losses |
|---|---|---|---|
| bank | [20.20, 23.80] | [0.52, 0.56] | [0.102, 0.116] |
| bike | [17.80, 19.60] | [2.80, 2.92] | [0.114, 0.122] |
| adult | [19.80, 21.40] | [3.18, 3.40] | [0.146, 0.151] |
| netherlands | [15.40, 18.40] | [2.34, 2.43] | [0.282, 0.292] |
| heloc | [18.00, 23.00] | [1.24, 1.30] | [0.286, 0.295] |
| spambase | [24.40, 25.60] | [0.88, 0.91] | [0.081, 0.088] |

*Table 14.* SPLIT/LicketySPLIT for non-sparse trees. For this variant, we set $\lambda = 1e - 5$ and ran our algorithms over 5 trials. We show the 95% confidence interval. This shows that our algorithms are capable of producing non-sparse trees. We may not prefer to do this in practice if there are interpretability constraints or if we can get a well performing model with much fewer than 20 leaves.

| Dataset | Algorithm | Runtimes (s) | Test Loss |
|---|---|---|---|
| compas | CART (with binary features) | [0.00, 0.00] | [0.325, 0.352] |
| | CART (with cont features) | [0.00, 0.00] | [0.322, 0.354] |
| bank | CART (with binary features) | [0.00, 0.00] | [0.106, 0.122] |
| | CART (with cont features) | [0.00, 0.00] | [0.106, 0.123] |
| bike | CART (with binary features) | [0.02, 0.03] | [0.208, 0.215] |
| | CART (with cont features) | [0.01, 0.01] | [0.208, 0.215] |
| adult | CART (with binary features) | [0.01, 0.01] | [0.168, 0.197] |
| | CART (with cont features) | [0.02, 0.02] | [0.183, 0.191] |
| hypothyroid | CART (with binary features) | [0.00, 0.00] | [0.009, 0.014] |
| | CART (with cont features) | [0.00, 0.00] | [0.009, 0.014] |
| covertype | CART (with binary features) | [0.06, 0.07] | [0.253, 0.256] |
| | CART (with cont features) | [0.68, 0.69] | [0.266, 0.269] |
| netherlands | CART (with binary features) | [0.01, 0.01] | [0.332, 0.342] |
| | CART (with cont features) | [0.00, 0.00] | [0.314, 0.338] |
| heloc | CART (with binary features) | [0.01, 0.01] | [0.293, 0.299] |
| | CART (with cont features) | [0.01, 0.01] | [0.293, 0.299] |
| spambase | CART (with binary features) | [0.00, 0.00] | [0.156, 0.208] |
| | CART (with cont features) | [0.01, 0.01] | [0.164, 0.207] |

*Table 15.* Comparison between CART (with binary features) and CART (with cont features) for trees with 3–6 leaves. The 95% confidence interval is shown.

| Dataset | Algorithm | Runtimes (s) | Test Loss |
|---|---|---|---|
| compas | CART (with binary features) | [0.00, 0.00] | [0.3197, 0.3388] |
| | CART (with cont features) | [0.00, 0.00] | [0.3253, 0.3385] |
| bank | CART (with binary features) | [0.00, 0.00] | [0.1109, 0.1193] |
| | CART (with cont features) | [0.00, 0.00] | [0.1061, 0.1227] |
| bike | CART (with binary features) | [0.01, 0.01] | [0.1718, 0.1911] |
| | CART (with cont features) | [0.00, 0.01] | [0.1661, 0.2380] |
| adult | CART (with binary features) | [0.02, 0.02] | [0.1647, 0.1803] |
| | CART (with cont features) | [0.03, 0.04] | [0.1647, 0.1803] |
| hypothyroid | CART (with binary features) | [0.00, 0.00] | [0.0030, 0.0053] |
| | CART (with cont features) | [0.00, 0.00] | [0.0015, 0.0042] |
| covertype | CART (with binary features) | [0.07, 0.07] | [0.2501, 0.2557] |
| | CART (with cont features) | [0.99, 1.00] | [0.2540, 0.2560] |
| netherlands | CART (with binary features) | [0.01, 0.01] | [0.3323, 0.3590] |
| | CART (with cont features) | [0.00, 0.00] | [0.2966, 0.3137] |
| heloc | CART (with binary features) | [0.00, 0.00] | [0.2895, 0.3015] |
| | CART (with cont features) | [0.01, 0.01] | [0.2926, 0.2990] |
| spambase | CART (with binary features) | [0.00, 0.00] | [0.1101, 0.1407] |
| | CART (with cont features) | [0.02, 0.02] | [0.1142, 0.1409] |

*Table 16.* Comparison between CART (with binary features) and CART (with cont features) for trees with 7–10 leaves. The 95% confidence interval is shown.

| Dataset | Algorithm | Runtimes (s) | Test Loss |
|---|---|---|---|
| compas | CART (with binary features) | [0.00, 0.00] | [0.3179, 0.3334] |
| | CART (with cont features) | [0.00, 0.00] | [0.3176, 0.3334] |
| bank | CART (with binary features) | [0.00, 0.00] | [0.1087, 0.1176] |
| | CART (with cont features) | [0.00, 0.00] | [0.1045, 0.1193] |
| bike | CART (with binary features) | [0.04, 0.05] | [0.1288, 0.1354] |
| | CART (with cont features) | [0.02, 0.02] | [0.1296, 0.1387] |
| adult | CART (with binary features) | [0.02, 0.03] | [0.1543, 0.1615] |
| | CART (with cont features) | [0.04, 0.04] | [0.1542, 0.1615] |
| hypothyroid | CART (with binary features) | [0.00, 0.00] | [0.0045, 0.0083] |
| | CART (with cont features) | [0.00, 0.00] | [0.0023, 0.0045] |
| covertype | CART (with binary features) | [0.07, 0.07] | [0.2485, 0.2537] |
| | CART (with cont features) | [1.22, 1.23] | [0.2529, 0.2554] |
| netherlands | CART (with binary features) | [0.01, 0.01] | [0.3021, 0.3115] |
| | CART (with cont features) | [0.00, 0.00] | [0.2934, 0.3090] |
| heloc | CART (with binary features) | [0.01, 0.01] | [0.2961, 0.3047] |
| | CART (with cont features) | [0.02, 0.02] | [0.2983, 0.3057] |
| spambase | CART (with binary features) | [0.01, 0.01] | [0.1064, 0.1416] |
| | CART (with cont features) | [0.02, 0.02] | [0.1144, 0.1359] |

*Table 17.* Comparison between CART (with binary features) and CART (with cont features) for trees with 11–14 leaves. The 95% confidence interval is shown.

### A.14. Predictive Multiplicity of our Rashomon Set

We illustrate another metric showing the approximation ability of RESPLIT. For each example in the training set, we computed the variance in predictions across models in the Rashomon set. The distribution of this variance over training examples is shown as a box plot for each dataset. Figure 23 and Table 18 shows that there is minimal empirical difference in the predictive multiplicity of original vs RESPLIT Rashomon sets.

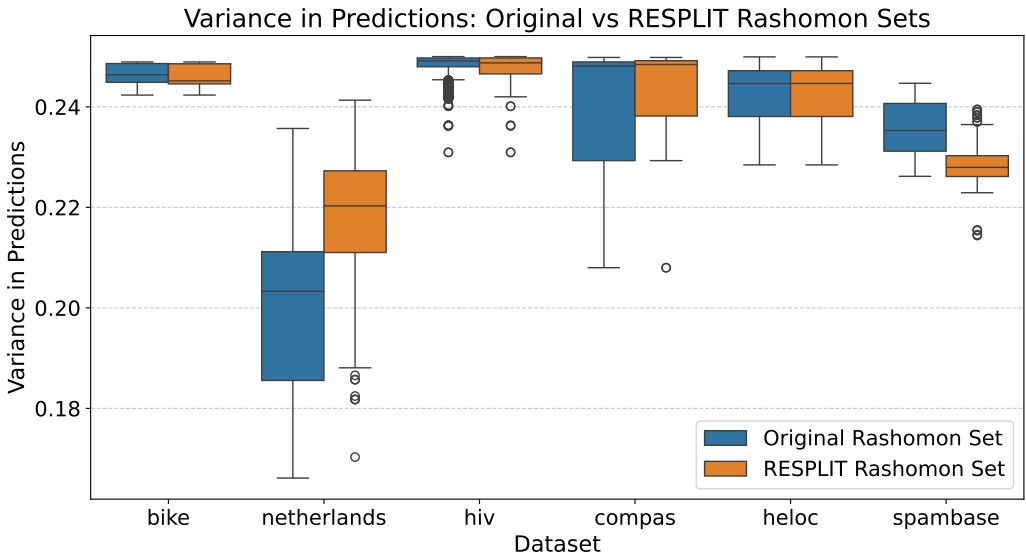

*Figure 23.* Illustration of the predictive multiplicity of the original and RESPLIT Rashomon Sets. $\lambda = 0.02$, $\epsilon = 0.01$, Depth Budget $= 5$, Lookahead depth $= 3$.

| Dataset | Mean Variance (Original Rashomon Set) $\pm$ Std Dev | Mean Variance (RESPLIT) $\pm$ Std Dev |
|---|---|---|
| bike | $0.2464 \pm 0.0023$ | $0.2458 \pm 0.0024$ |
| netherlands | $0.2017 \pm 0.0190$ | $0.2187 \pm 0.0122$ |
| hiv | $0.2485 \pm 0.0018$ | $0.2478 \pm 0.0027$ |
| compas | $0.2389 \pm 0.0138$ | $0.2407 \pm 0.0149$ |
| heloc | $0.2419 \pm 0.0081$ | $0.2419 \pm 0.0081$ |
| spambase | $0.2359 \pm 0.0055$ | $0.2284 \pm 0.0037$ |

*Table 18.* Illustration of the predictive multiplicity of the original and RESPLIT Rashomon Sets. $\lambda = 0.02$, $\epsilon = 0.01$, Depth Budget $= 5$, Lookahead depth $= 3$. This presents results in Figure 1 in tabular form.

