# OpenReview forum: "Near-Optimal Decision Trees in a SPLIT Second"
_ICML.cc/2025/Conference — ICML 2025 oral_

### Official Review · Reviewer_njLA · 2025-03-14

**Overall Recommendation:** 4

**Summary:**

The paper proposes three algorithms, SPLIT, LicketySPLIT, and RESPLIT, building from a common underlying technique to train near-optimal decision trees efficiently.

On one end of the decision tree training spectrum stands greedy algorithms, which are extremely fast but might create sub-optimal decision trees. On the other end stand methods like branch and bound, which can search for globally optimal splits and thus create optimal decision trees, but are extremely expensive computationally. The paper finds a compromise between the two methods, thus creating near-optimal decision trees almost as fast as the greedy algorithm.

Their technique is based on one critical insight, i.e., nodes near leaves can be split greedily without sacrificing significant performance. Thus, instead of performing the complete branch and bound training, the paper suggests doing optimal search only till some 'lookahead depth', and then splitting the rest of the tree greedily. This algorithm, which is called SPLIT, is able to achieve a large drop in training time while maintaining close to optimal performance.

The paper also proposed two other versions of this technique, called LicketySPLIT, to further improve the runtime, and RESPLIT, focused on finding the set of near-optimal models (i.e., the Rashomon set). Their method is compared against several SOTA techniques, showing consistently good performance and faster training, thus empirically verifying the claims.

**Claims And Evidence:**

Claims made in the paper about the runtime of their algorithms are well supported by both theoretical proofs and empirical results.

Claims made in the paper about the performance benefits of their algorithms in real-world datasets are well supported by empirical results. There are also some theoretical results to support performance benefits, however these results aren't too strong in my opinion. But then again, the paper never promises any such strong theoretical proofs, so no complaints really.

**Essential References Not Discussed:**

I'm not too familiar with related work in the field. However, I enjoyed reading the related work section of the paper, and I don't believe there were any missing references in the paper.

**Experimental Designs Or Analyses:**

The details of the experimental design are present in the paper, and I believe can be reproduced with some effort.

The experimental analysis seemed valid to me. I did not spot any issues.

**Methods And Evaluation Criteria:**

The paper promises a new algorithm to train faster yet near-optimal decision trees. All techniques are compared on two axes: training time and performance, and thus the comparisons make sense for the application at hand. The paper uses a diverse set of datasets for evaluation, with many more present in the Appendix. Overall, the evaluation setup makes sense to me.

**Other Comments Or Suggestions:**

1. There should be some intuition or support for why splitting the trees greedily near the leaves works well. The empirical results of Appendix A.2 and A.3 are quite nice, but without their presence in the main text, the algorithm SPLIT almost feels like something the paper 'stumbled' into. Throughout the main paper, I never got an answer to 'why' the paper decided to split the trees near the leaves greedily. A suggestion: Maybe a compressed version of the discussion in Appendix A.2 and Figure 7 can find a home somewhere in the main paper?
2. I wish there was a better discussion of RESPLIT and the Rashomon sets in the main paper. Again, yes, some results are present in the Appendix, but in my opinion, the creation of the Rashomon set is an extremely important application of the SPLIT algorithm, as the cost of training multiple models would explode with other complex training techniques currently present in the literature.

A purely personal take on the organization of the paper: While I really enjoyed reading the related work and preliminary sections, I believe a lot of those discussions didn't necessarily need to be in the main paper. Similarly, the theorems and theoretical results too didn't need to be in the main paper. Instead, I think the paper can have a wider appeal if the space in the main paper is given to the two things mentioned above.

However, the authors might disagree, and I won't hold that against the paper at all. It was an overall delightful read.

**Other Strengths And Weaknesses:**

The paper is very well written and I thoroughly enjoyed reading it.

I do wish there were certain changes to the organization of the paper, however, I understand the choices made given the limited space. Hence, it is not really a weakness of the paper, but simply some suggestions that I wished to see in the paper (and are thus, listed in the next section).

**Questions For Authors:**

NA

**Relation To Broader Scientific Literature:**

Decision trees are a vital part of interpretable ML research. By providing an algorithm to train decision trees significantly faster than any existing technique, while still maintaining near-optimal performance, this work can impact both (a) future work in interpretable ML, and (b) real-world training and deployment of ML models.

**Theoretical Claims:**

I checked the correctness of proofs for Theorem 5.4 (runtime for LicketySPLIT) and Theorem 5.5 (performance guarantees against greedy method). I believe they are both correct. There were many other theorems for the runtime of other versions of the algorithm (such as SPLIT or RESPLIT), however, I did not check the correctness of these claims in detail.

---

> ### Author Rebuttal · Authors · 2025-03-30
>
> Thank you for your review! We really appreciate your feedback on the organization - since ICML allows an additional page for accepted submissions, we’d be happy to move some of the intuition for greedy splits near the leaves, as well as a discussion of RESPLIT and the Rashomon Set, from the appendix into the main paper. It is really helpful to know that you found that to be important enough to include in the main paper.

---

> > ### Comment · Reviewer_njLA · 2025-04-07
> >
> > Best of luck with the submission.

---

### Official Review · Reviewer_QmXu · 2025-03-15

**Overall Recommendation:** 4

**Summary:**

The paper introduces a family of algorithms called SPLIT (SParse Lookahead for Interpretable Trees) for decision tree optimization. These algorithms aim to bridge the gap between the scalability of greedy methods and the accuracy of optimal decision tree methods. The key idea is to use dynamic programming with branch and bound up to a shallow "lookahead" depth, and then switch to greedy splitting. The authors also extend their algorithm to approximate the Rashomon set of decision trees.

**Claims And Evidence:**

The central claim is that SPLIT achieves a sweet spot between greedy and optimal methods, providing near-optimal trees with significantly improved scalability. The experimental results in the paper seem to support this claim, showing substantial speedups compared to optimal methods with minimal loss in accuracy.

**Essential References Not Discussed:**

The "Blossom: An Anytime Algorithm for Computing Optimal Decision Trees" paper by Demirović et al. (2023) is a relevant work that is not cited or discussed in the submission. Both papers address the challenge of finding optimal decision trees, with a focus on improving scalability and finite-time performance. Demirović et al. (2023) propose an anytime algorithm (Blossom) based on dynamic programming, which shares similarities with the SPLIT approach in terms of aiming for efficiency and anytime behavior. A comparison with Blossom would have strengthened the paper.

**Experimental Designs Or Analyses:**

The experimental design appears sound overall, comparing SPLIT against greedy (CART) and optimal decision tree algorithms.

**Methods And Evaluation Criteria:**

The proposed SPLIT algorithm and its variants (LicketySPLIT and RESPLIT) are clearly described. The evaluation criteria include accuracy, sparsity (number of leaves), and runtime, which are appropriate for the problem. The experiments are conducted on standard datasets, and the results are compared against greedy and optimal decision tree methods.

**Other Comments Or Suggestions:**

- It would be beneficial to discuss the limitations of the SPLIT algorithm in more detail, such as scenarios where it might underperform or when the greedy approximation is less effective.

**Other Strengths And Weaknesses:**

Strengths:
* The proposed SPLIT algorithm is novel and offers a good balance between accuracy and scalability.
* The paper is well-written and the experimental results are convincing.
* Recent developments in decision tree learning are thoroughly reviewed.

**Questions For Authors:**

Lookahead for policy improvement is a well-known technique in approximate dynamic programming and reinforcement learning. Maybe the authors should discuss the connection of this work to the DP and RL literature.

**Relation To Broader Scientific Literature:**

The paper effectively positions its contributions within the context of existing decision tree optimization methods. It clearly discusses the limitations of greedy and optimal approaches and how SPLIT aims to address them. The authors also relate their work to the concept of the Rashomon set in interpretable machine learning.

**Theoretical Claims:**

The authors theoretically prove that their algorithms scale exponentially faster in the number of features than optimal decision tree methods and can perform arbitrarily better than a purely greedy approach. I did not check the correctness of these proofs.

---

> ### Author Rebuttal · Authors · 2025-03-30
>
> > The "Blossom: An Anytime Algorithm for Computing Optimal Decision Trees" paper by Demirović et al. (2023) is a relevant work that is not cited or discussed in the submission. Both papers address the challenge of finding optimal decision trees, with a focus on improving scalability and finite-time performance. Demirović et al. (2023) propose an anytime algorithm (Blossom) based on dynamic programming, which shares similarities with the SPLIT approach in terms of aiming for efficiency and anytime behavior. A comparison with Blossom would have strengthened the paper.
>
> Thank you for mentioning Blossom. We’re glad to add a discussion of this work to the paper, and we agree it should be cited - it's an interesting and distinct approach from our own. We’re confident incorporating Blossom does not change the conclusions of our paper - the Blossom paper mentions that Murtree outperforms Blossom for depth 5 or shallower; and we compare extensively with Murtree, with our primary paper results focusing on depth 5.
>
> If you’d prefer to see a direct empirical comparison, we’re happy to look into that. Unfortunately, though we’ve made every attempt to do so, we haven't been able to run the approach by the end of the first rebuttal window - the codebase linked from the paper, https://gitlab.laas.fr/ehebrard/blossom , leads to a circular import error when we follow installation instructions on our machines (this looks to be a known issue that hasn’t yet been resolved - the latest commit to the repository adjusts the makefile/wrapper, and is titled “not quite”). Looking through the 9 works on google scholar that cite this work, none seem to have successfully run the approach and reported results, so we think this may be a larger issue than just our own environment. We're making our best effort to get an empirical comparison done in time.
>
> > Lookahead for policy improvement is a well-known technique in approximate dynamic programming and reinforcement learning. Maybe the authors should discuss the connection of this work to the DP and RL literature.
>
> Lookahead for policy improvement is not quite the same as what we’re doing, though we agree discussion of these approaches in our related work would be warranted. We’re finding a globally optimal tree when behaviour is fixed to be greedy past a specific frontier, then postprocessing to improve behaviour past that frontier, with fixed behaviour beforehand. There is certainly a similar spirit of exploration and exploitation tradeoffs in our method, which we’re happy to discuss in related work.

---

### Official Review · Reviewer_2Fk5 · 2025-03-18

**Overall Recommendation:** 4

**Summary:**

This paper proposes a decision-tree search method for producing near-optimal decision trees in an efficient way. The authors use a look-ahead mechanism to quickly evaluate tree candidates. The authors demonstrate their method in terms of loss, runtime, and Rashomon set search accuracy.

**Claims And Evidence:**

Overall, the claims are straightforward. As they discover a near-optimal model wrt accepted Rashomon criteria, and can evaluate this near optimality wrt runtime.The two main figures, and Table 1 support their efficiency claims well.

**Essential References Not Discussed:**

No suggestions

**Experimental Designs Or Analyses:**

The evaluation is straightforward. There are some qualitative results in line with prior work on Rashomon sets that would fit well for their model, i.e. to understand whether it produces a biased sampling (in the data sense, not the demographic sense) of models in the Rashomon set. This to me is frankly more interesting than runtime.

**Methods And Evaluation Criteria:**

The method is clever and very effective, empirically. The authors achieve a 100x speed-up in some applications.

**Other Comments Or Suggestions:**

Covered above.

**Other Strengths And Weaknesses:**

Overall, this work is very strong, within the specific area of optimal decision trees. One might criticize the scope of the work as niche or decision trees are too simple or outmoded. For readers who value this research area (as I do), the practical value of this paper in applications requiring maximal interpretability is very high.

One weakness is a lack of (analytically) characterizing the loss in *near* optimality (vs. optimality), and a better result for this tradeoff of optimality vs. runtime (e.g. maybe in an anytime paradigm.)

More generally, I think the authors focus too much on efficient near-optimality wrt runtime rather than the analytical (or empirical) results around the set of trained models over the search. Aspects that are typical in the Rashomon analysis are also relevant in a heuristic Rashomon search: predictive multiplicity and other variance measures, etc.

**Questions For Authors:**

1. How does the distribution of data affect the runtime of the method? One might imagine a distribution that's easy for a GLM, or some kernel function to fit, but for example, requires a deep decision tree. i.e. the fit may be near-optimal, but the optimal tree over the class of decision trees still isn't a great model. Do you have some results in these worst-cases wrt data distribution/high tree complexity?

2. Are there real applications for which random retraining for Rashomon estimation are prohibitive but this method isn't? That is, if I can use random resampling to achieve a model in the Rashomon set, the proposed method only seems to meet this rashomon threshold (empirically) more efficiently. It doesn't, for example, achieve a stricter Rashomon threshold than retraining is likely to achieve (i.e. "nearer-optimal" than RETRAIN)?.

3. Do you have results for the predictive multiplicity in the set of models trained by the proposed method? Are these models of higher/lower variance than the Rashomon set?
.

**Relation To Broader Scientific Literature:**

This work is a very significant contribution in the work on optimal (or near optimal) decision trees.

**Theoretical Claims:**

The main analytical results are somewhat light, as (in my estimation) this is largely an applied and practical paper.

---

> ### Author Rebuttal · Authors · 2025-03-30
>
> **Question 1 (How does the distribution of data affect the runtime of the method)**
>
> We’ve given standard worst-case runtime analysis in our theory section, which gives the worst case runtime even for adversarial data distributions. Even for an adversarial dataset that requires high tree complexity (i.e. a noiseless sine wave over a single continuous feature), our method is a substantial improvement to existing optimal tree methods that struggle to scale to larger depths.
>
> It seems like this is a question about decision trees’ theoretical benefits as a model class under adversarial distributions. This question is out of scope for our work, which is about improving performance for real-world datasets relative to other decision tree approaches. We’re happy to discuss related work on trees’ ability to achieve performance comparable to other model classes (such as [1], which shows trees match other model classes’ performance when there is noise in labels), but our work’s primary contribution is an improvement targeting the specific model class of trees.
>
> **Question 2 (random retraining vs RESPLIT)**
>
> Our understanding is that you’re asking about how RESPLIT compares to an alternative approach that uses random resampling of data, then runs decision tree learning methods on these samples. Is that correct? If so, please see below.
>
> We draw inspiration from the original paper on enumerating the entire Rashomon set for decision trees [2]. Figure 1 of that paper shows a comparison with this proposed resampling method, and demonstrates that it rarely creates models in the Rashomon set. It also leaves much of the Rashomon set unexplored. We show that the models we find are consistently within the true Rashomon set (see for example the precision result in table 2).
>
> Based on your question, it also seems like you may be asking how close the models we find are to optimal, relative to that resampling approach. Certainly the best models in our approximate Rashomon set are almost exactly the same as the best models in the true Rashomon set - since RESPLIT always includes the SPLIT tree in its set of models, and we’ve shown earlier in the paper that the SPLIT tree is regularly comparable to the objective of the best possible tree (see also the training objective results in our response to reviewer KaCM).
>
> If there’s another baseline you had in mind when you mention RETRAIN, please do let us know!
>
> **Question 3 (Predictive Multiplicity)**
>
> Here are some results on predictive multiplicity: https://rb.gy/axzek9. For each example in the training set, we’ve computed the variance in predictions across models in the Rashomon set. The distribution of this variance over training examples is shown as a box plot for each dataset. We see that RESPLIT exhibits similar predictive multiplicity as models in the original Rashomon set. This is yet another metric showing the approximation ability of RESPLIT.
>
> > Aspects that are typical in the Rashomon analysis are also relevant in a heuristic Rashomon search: predictive multiplicity and other variance measures, etc.
>
> We evaluate predictive multiplicity above, but to address your comment about other Rashomon set evaluations, we’d like to draw your attention to the evaluation we’ve done in Table 1, which shows our Rashomon Set approximation’s ability to help in downstream variable importance tasks. The Rashomon Importance Distribution computed on the full Rashomon set gives almost identical variable importance rankings to the Rashomon Importance Distribution computed on the RESPLIT Rashomon set sample (with rank correlation near or matching the best possible result of 1.0). The full results can be seen in Figure 16 in the Appendix.
>
>
> [1] Semenova, L., Chen, H., Parr, R., & Rudin, C. (2023). A path to simpler models starts with noise. Advances in neural information processing systems, 36, 3362-3401.
>
> [2] Xin, R., Zhong, C., Chen, Z., Takagi, T., Seltzer, M., & Rudin, C. (2022). Exploring the whole rashomon set of sparse decision trees. Advances in neural information processing systems, 35, 14071-14084.

---

> > ### Comment · Reviewer_2Fk5 · 2025-04-02
> >
> > Thank you for your thoughtful responses. This addressed all of my concerns with the paper. This is good work that I hope to see at the conference. Best of luck on the submission.

---

### Official Review · Reviewer_KacM · 2025-03-19

**Overall Recommendation:** 3

**Summary:**

Authors propose a decision tree learning method fast like greedy trees and precise like optimal ones. For that they call an optimal solvers only for some subtrees during construction of the overall tree. The paper is well-explained and well-written. Topic is important. The claims are not supported properly by experiment which we discuss next.

## Update after rebuttal
While I still believe SPLIT will not be used in practice to do machine learning, i.e, by industry or education to compute trees that generalize well to unseen data, the authors did a good job to answer my questions during the rebuttal. Indeed, I am not conviced that SPLIT will scale better than purely optimal decision trees like claimed by the authors as no significant benchmarking of SPLIT against CART was performed on big datasets and or deep trees. But the paper might be of interest for the optimal trees community, albeit smaller than the whole ICML community. Hence I would not mind this paper to be accepted and I will still praise the authors for the clarity of the work and the hard work during the rebuttal. Let objectivity triumph over subjectivity and make ICML better.

**Claims And Evidence:**

You claim near optimality which makes sense from how you construct your tree. However you don't show it in practice. I do not see any figure with (regularized) train loss in the main paper.

You claim speed/scalability (lower complexity than fully optimal approaches hence can be used for bigger problems). It is hard for me to say you support that claim. First the train time is measured in seconds. Baselines that you compare have different implementations, e.g. the greedy tree is in cython while murtree is in C. It would be better for you to measure the speed as a function of calls to a greedy solver during construction of the tree. Hence you could fairly compare speed.
Furthermore, looking at the red squared figures in Figure 3 and 4; I only see 2 datasets (Bank and Netherlands) where both (Lickety) SPLIT is the fastest non-greedy algorithm and the greedy tree is not in the red square.
Overall, adding the fact that SPLIT requires binary features, I am not convinced by your experiments that SPLIT is a useful algorithm. As I dont see a clear trend of SPLIT have both a better loss than CART and a better speed than Optimal trees.

I would also recommend that you do multiplte train/test split repetitions in your experiment to make them more significant and that you write what the error bars are.

I think claiming interpretability does not bring much to your contribution I like more the fact that you attempt to brdige the gap between greedy and optimal trees.

I did not check any claims regarding rashomon sets.

**Essential References Not Discussed:**

No.

**Experimental Designs Or Analyses:**

I spent most of my time after understanding SPLIT to review the experiment sections. I read all of them in the main paper and in the Appendix, and in the code.

The experiments are weak:
No multiple repetitions of experiments, no multiple train/test splits no explanation of error bars no cross validation.

When proposing a new tree algorithm, I recommend using the benchmark from Grinsztajn et al., 2022: Why do tree-based models still outperform deep learning on tabular data?

**Methods And Evaluation Criteria:**

I checked the proposed method and evaluation criteria.Method is sound and somewhat original (optimal splits up to depth d<D the greedy from d to D). Datasets chosen for evaluation are big enough. A weak point is that the method is limited to binary features.
I don't think those red squared boxes bring anything. Just make a plot of single points (one for each algo) on the loss-speed frontier.

**Other Comments Or Suggestions:**

In the contribs you write:
"We theoretically prove that our algorithms scale exponentially faster in the number of features than optimal decision
tree methods" I guess you wanted to say "slower".

**Other Strengths And Weaknesses:**

N/A

**Questions For Authors:**

Could you do more straight to the point plots with multiple train/test splits ? Just plot the loss - speed trade-off of the different tree algorithms. You can do five subplots, each one with a different sparsity value.

Could you redo figure 2 with train accuracy ? If you claim "optimality" it is essential.

Why or when would someone use SPLIT over CART ?

Could you compute optimal splits close to the root with some other optimal algortihm than GOSDT ?

I am unfamiliar with rashomon sets so I did not review most of the related contribution. I hope I did not miss something key. Could you please explain why rashomon sets are useful for decision tree algorithms ?
In the meantime I believe you did not perform serious enough experiments and the lack of compatibility with continuous features is a strong limitation for SPLIT to ever be used in practice.

The work is still good and original but more straightforward experiments focusing on train/test loss with respect to speed with multiple repetitions are for me necessary for this paper to be accepted

**Relation To Broader Scientific Literature:**

This works bridges the gap between optimal and greedy trees. They cover well both literatures. They even mention a very similar work (Top N from Blanc. 2024) which also develops a new heuristic tree algorithm with better accuracy than greedy trees and better speed than optimal trees. Hence this work is perfectly situated in the current literature.

I think the authors did an amazing job situating their work. And I liked the little teaser figure 1 a lot.

**Theoretical Claims:**

I did not check theroretical claim.

---

> ### Author Rebuttal · Authors · 2025-03-30
>
> Your core concerns seem to focus on handling continuous features, number of repetitions, and showing results with training objective. We address those points in detail below. In brief, our method is fully compatible with continuous features, our results are robust to (even more) multiple repetitions, and we have an excellent tradeoff between training objective and runtime.
>
> Requested figures/results can be found here: https://rb.gy/5w970v
>
> **Continuous features**
>
> Our method is fully compatible with continuous features. In this work, we start off with datasets with continuous features, and binarize scalably, using a method with provable guarantees on the performance of a downstream model and demonstrated practical benefit ([1]). Separating a decision tree algorithm into two steps, a binarization step and an optimization step, is a standard approach for non-greedy decision tree optimization (see, for example, [2,3]).
>
> **Further Repetitions**
>
> We originally ran experiments with 3 trials in Figures 3, 4, 5, and 6. The error bars correspond to the 95% CI obtained (i.e., 1.96x standard error). In the linked figures above, we’ve added results over 5 seeded trials. We do a random 80/20 train-test split for each trial.
>
> **Regularized Train Loss**
>
> We’ve redone Figure 2 of our paper with regularized train loss - showing the same benefits as the test loss (see Figure 3 in https://rb.gy/5w970v). Tables 1-3 in that link present some of same results in tabular form for clarity, showing our method is very close to optimal (GOSDT) in regularized training loss while being much faster. Figure 1 in the link also shows training loss for many methods.
>
> **Greedy vs (Lickety)SPLIT**
>
> Note from figures 1 and 2 in https://rb.gy/5w970v that we see a consistent benefit in runtime and performance for both training and test losses. These results do hold up in our main paper figures - greedy is sometimes within the red square, but rarely, if ever, performing equivalently on loss.
>
> You asked us to describe when we would use SPLIT instead of CART - our answer, based on these plots, is "Always!". We’re substantially outperforming greedy methods. Even in the worst case on these real world datasets, we achieve accuracy comparable to CART with a runtime of about a second, so there is minimal cost to using our approach – no disadvantage at all.
>
> **Scalability**
>
> These distinct implementations are not quite as distinct as they may seem - MurTree, MAPTree, GOSDT, and DL8.5 all use python wrappers of C++ implementations. Cython is a python wrapper around C.
> We also note that there should be no concerns about the comparison between GOSDT and SPLIT, since SPLIT uses a modified version of GOSDT’s code.
>
> Note that our scalability claims are thoroughly theoretically justified, with further details in section 5 of the main paper as well as the Appendix. For example, we prove LicketySPLIT is a polynomial-time algorithm, whereas optimal trees are NP-hard.
>
> The number of calls to a greedy solver would be a metric unique only to SPLIT and LicketySPLIT, and none of the other algorithms we compared against. We’ve never seen a paper use that metric since it doesn’t actually measure run time fairly across algorithms. It is unfortunately not a suitable metric to facilitate comparison with other algorithms.
>
> **Additional Points**
>
> >Could you compute optimal splits close to the root with some other optimal algorithm than GOSDT?
>
> We can! One of our approach’s advantages is that you can indeed compute optimal splits up to the lookahead depth using a different optimal algorithm than GOSDT. The adjustments to SPLIT remain quite simple for any dynamic programming based approach.
>
> >Could you please explain why rashomon sets are useful for decision tree algorithms ?
>
> Rashomon sets are useful whenever we want to understand a set of many alternative models that are all almost identically performant. Please see [4] for an approachable review of some of the many benefits of this approach.
>
> >I recommend using the benchmark from Grinsztajn et al., 2022
>
> Figure 4 in https://rb.gy/5w970v shows results for the largest classification dataset in that benchmark. We are happy to add even further experiments on more datasets.
>
> [1] McTavish, H., Zhong, C., Achermann, R., Karimalis, I., Chen, J., Rudin, C., & Seltzer, M. (2022). Fast Sparse Decision Tree Optimization via Reference Ensembles. AAAI, 36(9), 9604-9613
>
> [2] Lin, J., Zhong, C., Hu, D., Rudin, C., & Seltzer, M. (2020). Generalized and Scalable Optimal Sparse Decision Trees. ICML, 119, 6150–6160.
>
> [3] Demirović, E., Lukina, A., Hebrard, E., Chan, J., Bailey, J., Leckie, C., Ramamohanarao, K., & Stuckey, P. J. (2022). MurTree: Optimal decision trees via dynamic programming and search. JMLR, 23(26), 1–47.
>
> [4] Rudin, C., Zhong, C., Semenova, L., Seltzer, M., Parr, R., Liu, J., Katta, S., Donnelly, J., Chen, H., & Boner, Z. (2024). Position: Amazing things come from having many good models. ICML, 1742, 1–13.

---

> > ### Comment · Reviewer_KacM · 2025-04-02
> >
> > Hello,
> >
> > Thank you for the new figures and the answer.
> >
> > **Conitnuous Features**
> > I know that most optimal trees work with binarized datasets which in my opinion does not mean they are compatible with continuous features as the returned trees are partitioning the binarized dataset and not the *continuous* features one. For example there exists optimal trees that are natively compatible with continuous features: [Quant-BnB: A Scalable Branch-and-Bound Method for Optimal Decision Trees with Continuous Features] . Similarly, the CART algorithm [Breiman 1984] is natively compatible with continuous features.
> >
> > **Further repetitions and CIs**
> > I believe your methodology is flawed. When you compute the 95% CIs with 1.96 x std it assumes that the std there is the *true* std of the distribution of your test loss. However you only *estimate* the latter with 5 seeds, hence you should use bootstrapped confidence intervals. The scipy package for python has a built-in method for that.
> >
> > **Regularized Training Loss**
> > Thank you for the updated figure. I raised my score 2
> >
> > **Generalization and when you should use SPLIT over CART**
> > I would not overclaim that SPLIT should always be used over CART. 1) You don't account for binarization in the runtimes. 2) You did not tune hyperparameters for CART. 3) Out of 27 "red squares", 21 contain greedy trees. 4) On your plots (from both the rebuttal and the main paper) it seems that SPLIT cannot really compute trees with more than 14 leaves whereas CART and topK can; and it seems those more complex trees have better test losses than (Lickety)SPLIT most of the time.
> >
> > Hence, I am still not convinced that one should use SPLIT over CART. I would say the use case of SPLIT is when one want to compute a fast approximation of a small optimal tree.
> >
> > If you provide a more convincing comparison of SPLIT with CART I will raise my score again !
> >
> > Thank you for the hard work.

---

> > > ### Author Response · Authors · 2025-04-07
> > >
> > > Thank you for considering our responses and updating your score. We’ve added a more comprehensive comparison between SPLIT and CART, which we hope you’ll find compelling.
> > >
> > > You can view the updated results here: [https://shorturl.at/7zV7a](https://shorturl.at/7zV7a). These include bootstrapped confidence intervals (CIs), as you recommended.
> > >
> > > ---
> > >
> > > ### **Key Updates**
> > >
> > > 1. **Binarization**
> > >    - Binarization is now included in runtimes. We now binarize such that all datasets have between 30–35 binary features. We perform a search over lambda between 0.001 and 0.02, with 40 values in between, to get a range of SPLIT/LicketySPLIT trees with different leaves. From Tables 1, 2, and 3, it typically takes only a **couple seconds** (consistent with observations in Figures 3 and 12 in [1]), and both variants of SPLIT usually outperform, or rarely match, CART in test loss even when the latter is trained on the continuous dataset.
> > >
> > >    - Because the optimal decision tree methods tested in our paper also require binarization, this cost also needs to be added to their runtime, so this **doesn’t change our paper’s conclusions** with respect to other methods.
> > >
> > >    - As you mention, CART is natively compatible with continuous features. However, we show in Tables 5, 6, and 7 in the updated results that **CART's test performance is largely unaffected by using binarized vs. continuous data**. This implies that binarization is not unfairly disadvantageous to CART in our paper.
> > >
> > > 2. **Expanded CART Hyperparameter Search**
> > >    - We now perform grid search over `min_samples_leaf`, `min_samples_split`, `max_features`, `splitter`, and `class_weight` using scikit-learn’s `DecisionTreeClassifier`. All results in all tables in https://shorturl.at/7zV7a show the best performing configuration for that sparsity level.
> > >      - `min_samples_leaf` and `min_samples_split` are chosen from `np.logspace(0, 5, 10)`
> > >      - `max_features` is chosen from `['sqrt', 'log2', None]`
> > >      - `splitter` is chosen from `['best', 'random']`
> > >      - `class_weight` is chosen from `[None, 'balanced']`
> > >
> > >
> > >    - Even with these broader configurations, CART **underperforms** compared to SPLIT/LicketySPLIT across sparsity levels.
> > >
> > > 3. **Red Squares**
> > >    - We’ve removed the "red squares" from plots in our rebuttal — they were meant for illustrative purposes to show the parts of the tradeoff that we’re interested in. They often cover a wide range of losses (e.g., Bike in Figure 3 in the paper), so two trees both being in the red square does not mean they are equivalently performant — they just warrant further comparison on accuracy vs. runtime.
> > >
> > > 4. **Support for Non-Sparse Trees**
> > >    - SPLIT/LicketySPLIT can absolutely generate trees with >14 leaves. We reduce regularization (`λ=1e-5`) to show the best LicketySPLIT or SPLIT tree with up to **25+ leaves** in Table 4 in [https://shorturl.at/7zV7a](https://shorturl.at/7zV7a).
> > >
> > >    - Even with low regularization, some datasets (e.g., Hypothyroid, Covertype) have optimal or near-optimal trees with 10–12 leaves. This is why they are absent from this table. However, we note that there is no drop in test performance relative to CART / top-K with 25+ leaves when this happens — see Hypothyroid and Covertype in Figure 4.
> > >
> > > ---
> > >
> > > ### **Clarifications**
> > >
> > > To avoid your concern about overclaiming in our prior response, we would say that SPLIT/LicketySPLIT is best used when one desires fast, high-performing, and sparse decision trees.
> > >
> > > ---
> > >
> > > ### **On Bootstrapped CIs**
> > >
> > > We now use bootstrapped CIs rather than `1.96 * std / sqrt(N)`, as shown in the updated results.
> > >
> > > ---
> > >
> > > ### **On Continuous Features**
> > >
> > > If your concern is regarding comparisons with CART trained directly on the continuous data, [https://shorturl.at/7zV7a](https://shorturl.at/7zV7a) shows this comparison in Tables 1, 2, and 3. As mentioned above, we note that **SPLIT/LicketySPLIT still outperform CART**.
> > >
> > > If your concern is more broadly that binarization breaks compatibility with continuous features, we respectfully disagree. Even after binarization, splits still represent thresholds on original features (e.g., `age ≤ 30`), so our trees correspond to partitions in the continuous space, not just the binary space.
> > >
> > > Regarding your point about QuantBnB — it does handle continuous features natively, but it is **only practical for shallow depths**. We mention this limitation in the related work section of our paper. Their paper quotes: "*we recommend using our procedure for fitting optimal decision trees with d ≤ 3*". Our method **scales to deeper trees** (e.g., depth 6), with results in our paper demonstrating this (e.g., Figure 6 in the Appendix).
> > >
> > > ---
> > >
> > > **[1]** McTavish, H., Zhong, C., Achermann, R., Karimalis, I., Chen, J., Rudin, C., & Seltzer, M. (2022). Fast Sparse Decision Tree Optimization via Reference Ensembles. *Proceedings of the AAAI Conference on Artificial Intelligence, 36*(9), 9604–9613.

---

### Decision · Program_Chairs · 2025-05-01

**Decision:**

Accept (oral)

**Comment:**

The paper introduces methods for finding decision trees that enjoys high accuracy and scalability often found in greedy algorithms. The proposed algorithms are tested against benchmarks and to characterize the Rashomon Set of decision trees.

Overall, this is an excellent paper that significantly moves the needle in finding models that are accurate and interpretable. The reviewers were unanimous in accepting the paper. Decision trees are widely used in a variety of applications, can be very accurate, and are interpretable. This paper makes an important contribution to this area.

I encourage the authors to take the reviewers' comments into account when polishing their camera-ready version.